# Robust and Computation-Aware Gaussian Processes

**Marshal Sinaga**
ELLIS Institute Finland, Aalto University
`marshal.sinaga@aalto.fi`

**Julien Martinelli**
ELLIS Institute Finland, Aalto University
`julien.martinelli@aalto.fi`

**Samuel Kaski**
ELLIS Institute Finland, Aalto University, University of Manchester
`samuel.kaski@aalto.fi`

## Abstract

Gaussian processes (GPs) are widely used for regression and optimization tasks such as Bayesian optimization (BO) due to their expressiveness and principled uncertainty estimates. However, in settings with large datasets corrupted by outliers, standard GPs and their sparse approximations struggle with computational tractability and robustness. We introduce Robust Computation-aware Gaussian Process (RCaGP), a novel GP model that jointly addresses these challenges by combining a principled treatment of approximation-induced uncertainty with robust generalized Bayesian updating. The key insight is that robustness and approximation-awareness are not orthogonal but intertwined: approximations can exacerbate the impact of outliers, and mitigating one without the other is insufficient. Unlike previous work that focuses narrowly on either robustness or approximation quality, RCaGP combines both in a principled and scalable framework, thus effectively managing both outliers and computational uncertainties introduced by approximations such as low-rank matrix multiplications. Our model ensures more conservative and reliable uncertainty estimates, a property we rigorously demonstrate. Additionally, we establish a robustness property and show that the mean function is key to preserving it, motivating a tailored model selection scheme for robust mean functions. Empirical results confirm that solving these challenges jointly leads to superior performance across both clean and outlier-contaminated settings, both on regression and high-throughput Bayesian optimization benchmarks.

## 1 Introduction

Gaussian Processes (GPs) are a foundational tool in probabilistic machine learning, offering non-parametric modeling with principled uncertainty estimates [36]. Their use spans diverse domains such as time series forecasting, spatial modeling, and regression on structured scientific data. One particularly impactful area where GPs have become integral is Bayesian Optimization (BO), a sample-efficient framework for optimizing expensive black-box functions, where the surrogate model's predictive accuracy and uncertainty calibration directly influence decision-making [11]. Many scientific and industrial tasks are high-throughput optimization problems in which numerous experiments or simulations are performed, possibly in parallel. These can be effectively tackled by BO and include material design [15], drug discovery with extensive chemical space search [18], and fine-tuning gene expression in synthetic biology [6].

Yet, two major challenges limit the reliability of GPs in both regression and BO settings: scalability to large datasets and robustness to corrupted or outlier-contaminated observations. Sparse Variational Gaussian Processes (SVGPs, [17, 41, 42]) address scalability via inducing point approximations, yielding a low-rank approximation that allows large-scale regression and high-throughput BO.

39th Conference on Neural Information Processing Systems (NeurIPS 2025).

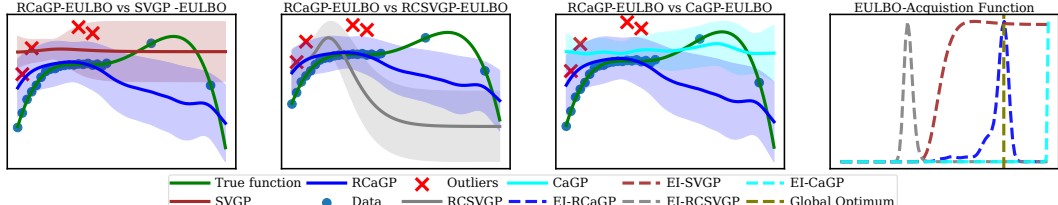

Figure 1: **Overview of our proposed RCaGP against concurrent baselines on a 1D example.**
(**Left**) SVGP fails to fit observed data contaminated by outliers, whereas RCaGP successfully fits
high-density data regions while preserving a higher variance due to the presence of outliers. (**Middle-left**) While enhancing robustness compared to SVGP, RCSVGP deviates more significantly from the
true function than RCaGP. (**Middle-right**) Even if CaGP displays increased posterior variance near
outliers, RCaGP provides superior posterior mean prediction. (**Right**) As a result, the acquisition
function landscape for RCaGP better prioritizes the true global optima.

Despite their computational efficiency, significant challenges hinder the adoption of SVGPs for
high-throughput BO tasks, due to the impact of surrogate posterior quality on data acquisition. For
starters, SVGP is not robust against outliers: upon contamination of the observed data, the posterior
mean can deviate significantly from the true latent function. Such deviation will misguide most
acquisition strategies, resulting in suboptimal performance. Outliers are an almost unavoidable
aspect of many real-world applications and datasets, specifically in high-throughput settings. They
can arise from various factors, such as faulty measurements, malfunctioning sensors, or genetic
anomalies. Next, SVGPs suffer from overconfident predictions, a well-known pathology caused by
the error incurred by the posterior approximation. Overconfident predictions occur in regions where
the inducing points are far from observed data. Accurate uncertainty quantification is essential for
sequentially identifying the optimal data points in BO [45].

**Contributions.** To address these challenges, we propose a novel approach that combines robust-conjugate SVGP [2] with computation-aware GP [47]. This method, referred to as the Robust
Computation-Aware GP (RCaGP), jointly tackles outlier robustness and approximation-induced
uncertainty, two challenges that are deeply intertwined in sparse GP models. Computational approximations such as low-rank matrix factorizations can exacerbate the effects of outliers, and mitigating
one without addressing the other leads to suboptimal performance. RCaGP can be used as a surrogate
in high-throughput BO, and its design is motivated by the need to address these two problems in
tandem. This can be seen in Figure 1. In summary, this work

1. (**Methodological**) introduces a novel GP model (RCaGP) that jointly addresses approximation-awareness and outlier robustness through a principled probabilistic framework, going beyond
   previous work that tackles these issues in isolation.

2. (**Theoretical**) establishes RCaGP's reliability through conservative uncertainty estimates and a
   formal robustness property. Our analysis reveals the central role of the mean function in preserving
   robustness, motivating a tailored model selection scheme.

3. (**Empirical**) demonstrates that jointly solving these two problems yields superior performance:
   RCaGP consistently outperforms existing state-of-the-art methods such as CaGP and RCSVGP on
   UCI regression tasks and BO settings, with and without outliers.

## 2   Background

**Robust conjugate GP.** The Gaussian noise assumption in GPs often leads to misspecification,
e.g., due to outliers. To address this problem, recent work proposed robust conjugate GP (RCGP),
which employs a robust loss function through generalized Bayesian inference [2]. Consider a
noisy dataset $\mathcal{D}_n = \{(\mathbf{x}_j, y_j)\}_{j=1}^n = (\mathbf{X}, \mathbf{y})$. We specify a GP prior for the finite function
evaluations $\mathbf{f} = [f(\mathbf{x}_1), \ldots, f(\mathbf{x}_n)]$: $p(\mathbf{f}) = \mathcal{N}(\mathbf{f}; \mathbf{m}, \mathbf{K})$, with $\mathbf{m} = [m(\mathbf{x}_1), \ldots, m(\mathbf{x}_n)]^\top$ and
$\mathbf{K} = [k(\mathbf{x}_j, \mathbf{x}_l)]_{1 \le j, l \le n}$. Both $m : \mathcal{X} \to \mathbb{R}$ and $k : \mathcal{X} \times \mathcal{X} \to \mathbb{R}$ denote the mean and kernel
functions, encoding prior beliefs about the function's structure and smoothness [36]. Conditioning on

observations, the posterior predictive distribution is Gaussian with mean $\mu_*(\mathbf{x})$ and variance $k_*(\mathbf{x},\mathbf{x})$.

$$\mu_*(\mathbf{x}) = m(\mathbf{x}) + \mathbf{k}_{\mathbf{x}}^\top (\mathbf{K} + \sigma_{\text{noise}}^2 \mathbf{J_w})^{-1}(\mathbf{y} - \mathbf{m_w}), \tag{1}$$

$$k_*(\mathbf{x},\mathbf{x}) = k(\mathbf{x},\mathbf{x}) - \mathbf{k}_{\mathbf{x}}^\top (\mathbf{K} + \sigma_{\text{noise}}^2 \mathbf{J_w})^{-1}\mathbf{k}_{\mathbf{x}}, \tag{2}$$

where $\mathbf{k}_{\mathbf{x}} = [k(\mathbf{x},\mathbf{x}_1), \dots k(\mathbf{x},\mathbf{x}_n)]^\top \in \mathbb{R}^n$. The key difference with vanilla GPs is the introduction of a weight function $w$, used by RCGP to down-weight the influence of potential outliers. Concretely, RCGP induces a corrected mean and posterior variance by replacing $(\mathbf{y} - \mathbf{m})$ with the shrinkage term $(\mathbf{y} - \mathbf{m_w})$, with $\mathbf{m_w} = \mathbf{m} + \sigma_{\text{noise}}^2 \nabla_y \log(\mathbf{w}^2)$ and the identity matrix $\mathbf{I}$ with a diagonal matrix $\mathbf{J_w} = \text{diag}\left(\frac{\sigma_{\text{noise}}^2}{2}\mathbf{w}^{-2}\right)$, for $\mathbf{w} = [w(\mathbf{x}_1,y_1), \dots, w(\mathbf{x}_n,y_n)]^\top$.

Nevertheless, RCGP inherits the cubic inference complexity of vanilla GPs, prohibiting its usage on large-scale datasets like those encountered in high-throughput BO. To remedy this issue, RCSVGP were introduced, enabling outlier-robust regression on large-scale data [2]. This model introduces inducing points $\mathbf{u} = \{\mathbf{u}_l\}_{l=1}^r$ for $r \ll n$, with inducing locations $\mathbf{z} = \{\mathbf{z}_l\}_{l=1}^r$ such that $\mathbf{u}_l = f(\mathbf{z}_l)$. Given $\mathbf{u}$, RCSVGP constructs an approximate posterior $q(\mathbf{u}) = \mathcal{N}(\boldsymbol{\mu}_{\mathbf{u}}, \boldsymbol{\Sigma}_{\mathbf{u}})$ with variational parameters $\boldsymbol{\mu}_{\mathbf{u}}$ and $\boldsymbol{\Sigma}_{\mathbf{u}}$, obtained by maximizing the evidence lower bound (ELBO) criterion:

$$\ell_{\text{ELBO}}^{\text{RCSVGP}} = \mathbb{E}_{q(\mathbf{f})}[\log \Psi^w(\mathbf{y}, \mathbf{f})] - \text{KL}[q(\mathbf{u})|p(\mathbf{u})], \tag{3}$$

where $\Psi^w(\mathbf{y}, \mathbf{f})$ denotes the robust loss function and $q(\mathbf{f}) = p(\mathbf{f}|\mathbf{u})q(\mathbf{u})$ is the approximate posterior over the function values conditioned on $\mathbf{u}$.

While RCSVGP reduces the time complexity to $\mathcal{O}(nr^2)$, like other approximate GP methods, it also exhibits overconfidence in regions where inducing points are far from the observations. This overconfidence can harm the exploration/exploitation balance in BO, leading to suboptimal results.

**Computation-aware GP.** [46, 47] proposed computation-aware GP (CaGP) to address the SVGP overconfidence. CaGP mitigates this issue by ensuring its posterior variance exceeds the vanilla GP variance. This corrected variance is obtained by adding *computational uncertainty*, associated with the approximation error in the representer weights $\mathbf{v}_* = (\mathbf{K} + \sigma_{\text{noise}}^2 \mathbf{I})^{-1}(\mathbf{y} - \mathbf{m})$. To obtain computational uncertainty, CaGP places a prior $p(\mathbf{v}_*) = \mathcal{N}(\mathbf{v}_0, \boldsymbol{\Sigma}_0)$ and updates its posterior distribution $\mathcal{N}(\mathbf{v}_i, \boldsymbol{\Sigma}_i)$ using a probabilistic linear solver based on linear Gaussian identities:

$$\mathbf{v}_i = \mathbf{C}_i (\mathbf{y} - \mathbf{m}), \tag{4}$$

$$\boldsymbol{\Sigma}_i = \mathbf{K}^{-1} - \mathbf{C}_i, \tag{5}$$

where $\mathbf{C}_i = \mathbf{S}_i(\mathbf{S}_i^\top \hat{\mathbf{K}}^{-1} \mathbf{S}_i)\mathbf{S}_i^\top \approx \mathbf{K}^{-1}$ is a low-rank approximation and $\mathbf{S}_i \in \mathbb{R}^{n \times i}$ is the action matrix associated with the chosen approximation technique. In the case of inducing points, $\mathbf{S}_i = k(\mathbf{X}, \mathbf{z}_i)$. Given a new data point $\mathbf{x}$, the predictive distribution of $f(\mathbf{x})$ is Gaussian, characterized by the mean $\mu_i(\mathbf{x})$ and variance $k_i(\mathbf{x}, \mathbf{x})$, given by:

$$\mu_i(\mathbf{x}) = m(\mathbf{x}) + \mathbf{k}_{\mathbf{x}}^\top \mathbf{v}_i, \tag{6}$$

$$k_i(\mathbf{x},\mathbf{x}) = k(\mathbf{x},\mathbf{x}) - \mathbf{k}_{\mathbf{x}}^\top \hat{\mathbf{K}}^{-1} \mathbf{k}_{\mathbf{x}} + \mathbf{k}_{\mathbf{x}}^\top \boldsymbol{\Sigma}_i \mathbf{k}_{\mathbf{x}} = k(\mathbf{x},\mathbf{x}) - \mathbf{k}_{\mathbf{x}}^\top \mathbf{C}_i \mathbf{k}_{\mathbf{x}}, \tag{7}$$

where $\hat{\mathbf{K}} = \mathbf{K} + \sigma_{\text{noise}}^2 \mathbf{I}$ and $\sigma_i(\mathbf{x}) = \mathbf{k}_{\mathbf{x}}^\top \boldsymbol{\Sigma}_i \mathbf{k}_{\mathbf{x}}$ denote the computational uncertainty. The posterior can be obtained in $\mathcal{O}(n^2 i)$ time complexity. As $i \to n$, CaGP recovers both the predictive mean and variance of the exact GP.

# 3 Robust and Computation-Aware Gaussian Processes

To tackle the intertwined challenges of outliers and approximation-induced overconfidence, we introduce RCaGP—a principled integration of robust-conjugate GPs with computation-aware approximations. These issues must be addressed jointly: approximations can amplify outlier effects, while robustness alone does not correct the biases they introduce. RCaGP unifies both aspects to improve predictive reliability and decision-making. Section 3.1 details its composite inference scheme, Section 3.2 derives a tailored optimization criterion, and Section 3.3 extends the framework to BO for joint model and query selection.

## 3.1 Robust Computation-aware Inference

Inspired by CaGP, we apply a probabilistic treatment to the representer weights $\hat{\mathbf{v}} = (\mathbf{K} + \sigma_{\text{noise}}^2 \mathbf{J_w})^{-1}(\mathbf{y} - \mathbf{m_w})$ of RCGP in Equation (1). This enables us to quantify uncertainty in approximating $\hat{\mathbf{v}}$—an aspect most approximate GPs ignore. RCaGP later incorporates this uncertainty into its predictive variance. We begin by placing a prior on $\hat{\mathbf{v}}$, i.e., $\hat{\mathbf{v}} \sim \mathcal{N}(\tilde{\mathbf{v}}_0 = \mathbf{0}, \tilde{\mathbf{\Sigma}}_0 = \tilde{\mathbf{K}}^{-1})$, where $\tilde{\mathbf{K}} = \mathbf{K} + \sigma_{\text{noise}}^2 \mathbf{J_w}$. We then update this belief via the linear Gaussian identity [46], yielding a Gaussian posterior with mean $\tilde{\mathbf{v}}_i$ and variance $\tilde{\mathbf{\Sigma}}_i$:

$$\tilde{\mathbf{v}}_i = \tilde{\mathbf{C}}_i \,(\mathbf{y} - \mathbf{m_w}), \tag{8}$$

$$\tilde{\mathbf{\Sigma}}_i = \tilde{\mathbf{K}}^{-1} - \tilde{\mathbf{C}}_i, \tag{9}$$

where $\tilde{\mathbf{C}}_i = \mathbf{S}_i (\mathbf{S}_i^\top \tilde{\mathbf{K}} \mathbf{S}_i)^{-1} \mathbf{S}_i^\top$ and $\mathbf{S}_i \in \mathbb{R}^{n \times i}$ are low-rank approximations of $\tilde{\mathbf{K}}^{-1}$ and the actions, respectively. The updates can be done sequentially [46, Algorithm 1] or in batches [47, Algorithm S2].

Recent work generalized the action matrix $\mathbf{S}_i \in \mathbb{R}^{n \times i}$ by treating it as a variable that can be optimized during model selection [47]. Given a new data point $\mathbf{x} \in \mathcal{X}$, the predictive distribution of $f(\mathbf{x})$ follows a normal distribution with mean $\hat{\mu}_i(\mathbf{x})$ and variance $\hat{k}_i(\mathbf{x}, \mathbf{x})$:

$$\hat{\mu}_i(\mathbf{x}) = m(\mathbf{x}) + \mathbf{k_x}^\top \tilde{\mathbf{v}}_i, \tag{10}$$

$$\hat{k}_i(\mathbf{x}, \mathbf{x}) = k(\mathbf{x}, \mathbf{x}) - \mathbf{k_x}^\top \tilde{\mathbf{K}}_i \mathbf{k_x} + \mathbf{k_x}^\top \tilde{\mathbf{\Sigma}}_i \mathbf{k_x} = k(\mathbf{x}, \mathbf{x}) - \mathbf{k_x}^\top \tilde{\mathbf{C}}_i \mathbf{k_x}. \tag{11}$$

In Equation (11), RCaGP incorporates the computational uncertainty $\mathbf{k_x}^\top \tilde{\mathbf{\Sigma}}_i \mathbf{k_x}$, corresponding to the variance of the representer weights $\tilde{\mathbf{\Sigma}}_i$, resulting in a combined uncertainty $\hat{k}_i(\mathbf{x}, \mathbf{x})$. This computational uncertainty guarantees that the predictive posterior variance of RCaGP is larger than RCGP, preventing RCaGP from overconfident predictions. Such behavior is referred to as a conservative uncertainty estimate. The computational complexity of RCaGP can be found in Appendix B.

The robustness of RCaGP hinges on the weight function $w$, which we choose to incorporate into the representer weights $\tilde{\mathbf{v}}$. To that end, we follow [2] and let:

$$w(\mathbf{x}, y) = \beta \left(1 + (y - m(\mathbf{x}))^2/c^2\right)^{-1/2}, \tag{12}$$

for $\beta$ and $c > 0$ the learning rate and soft threshold.

The weight function assigns smaller values to observations $y$ that deviate significantly from the mean prior $m(\mathbf{x})$, treating them as potential outliers. Lower weights reduce the influence of such points during inference, leading to a more robust posterior. Additionally, the weight enters the noise term $\mathbf{J_w} = \text{diag}([\sigma_{\text{noise}}^2/2\mathbf{w}^{-2}])$, effectively inflating the noise for outliers and further limiting their impact. $w(\mathbf{x}, y)$ depends critically on the mean prior: for an outlier $\hat{y}$, if $m(\hat{\mathbf{x}})$ is close to $\hat{y}$, the weight increases. However, choosing an informative mean prior is difficult without domain knowledge.

## 3.2 Model hyperparameters optimization

We use the evidence lower bound (ELBO) as a loss function to optimize the kernel hyperparameters $\boldsymbol{\theta} \in \mathbb{R}^p$. Following [47], this enables RCaGP to scale to large datasets while avoiding overconfidence. The variational family is defined using the RCaGP posterior $q_i(\mathbf{f}|\mathbf{y}, \boldsymbol{\theta})$, i.e., $\{q_i(\mathbf{f}) = \mathcal{N}(\hat{\mu}_i(\mathbf{X}), \hat{k}_i(\mathbf{X}, \mathbf{X}))|\mathbf{S}_i \in \mathbb{R}^{n \times i}\}$, parameterized by action $\mathbf{S}_i$. We replace the evidence term $\log p(\mathbf{y}|\boldsymbol{\theta})$ with the robust loss $\log \Psi^w(\mathbf{y}, \mathbf{f})$ from [2]. We then formulate ELBO as

$$\ell_{\text{ELBO}}^{\text{RCaGP}} = \mathbb{E}_{q_i(\mathbf{f})}[\log \Psi^w(\mathbf{y}, \mathbf{f})] - \text{KL}[q_i(\mathbf{f})\|p(\mathbf{f})]. \tag{13}$$

This loss learns the hyperparameters as if maximizing $\mathbb{E}_{p(\mathbf{f}|\mathbf{y}, \boldsymbol{\theta})}[\log \Psi^w(\mathbf{y}, \mathbf{f})]$ while minimizing the computational uncertainty associated with the approximation error. The derivation of the expected loss $\mathbb{E}_{q(\mathbf{f})}[\log \Psi^w(\mathbf{y}, \mathbf{f})]$ follows [2], while the derivation of the KL term $\text{KL}[q(\mathbf{f})\|p(\mathbf{f})]$ draws an analogy to CaGP. For the closed form and detailed derivation of ELBO, we refer the readers to Proposition C.1. We obtain the optimal hyperparameters $\boldsymbol{\theta}^*$ by minimizing the negative ELBO.

## 3.3 Joint model parameters and design selection strategy

To fully take advantage of RCaGP in BO, we leverage the expected lower bound utility (EULBO) framework introduced by [29]. At acquisition time, instead of optimizing the surrogate's inducing

points, variational parameters, and hyperparameters, only then to find a design maximizing the acquisition function, a global criterion is maximized in an end-to-end manner:

$$\ell_{\text{EULBO}}^{\text{RCaGP}} = \ell_{\text{ELBO}}^{\text{RCaGP}} + \mathbb{E}_{q_i(\mathbf{f})}[\log u(\mathbf{x}, f; \mathcal{D}_t)], \tag{14}$$

where $u : \mathcal{X} \times \mathcal{F} \to \mathbb{R}$ is the utility function. Here, we consider the expected improvement (EI) AF, reformulated as an expectation of the following utility:

$$u_{\text{EI}}(\mathbf{x}, f; \mathcal{D}_t) = \text{softplus}(f(\mathbf{x}) - y_t^*), \tag{15}$$

where $y_t^*$ denotes the best evaluation observed so far and $\text{softplus} : \mathbf{x} \mapsto \log(1 + \exp(\mathbf{x}))$. This reformulation replaces the commonly used ReLU function with a softplus function, guaranteeing the utility function remains strictly positive whenever $f(\mathbf{x}) \geq y^*$. Moreover, EULBO can be extended to support batch BO using Monte Carlo batch mode [5, 48]. Given a set of candidates $\mathbf{X} = \{\mathbf{x}_1, \ldots, \mathbf{x}_q\}$, the expected utility function corresponding to the $q-$improvement utility is:

$$\mathbb{E}[\log u_{\text{EI}}(\mathbf{X}, f; \mathcal{D}_t)] \approx \frac{1}{S} \sum_{s=1}^{S} \max_{s=1,\ldots,q} \text{softplus}(r_s), \tag{16}$$

with $r_s = f(\mathbf{x}_s) + \epsilon_s - y_t^*$ and $\epsilon_s \sim \mathcal{N}(0, 1)$. Finally, when considering RCaGP, the action matrices $\mathbf{S}_i$ are optimized, leading to the joint maximization problem

$$\mathbf{x}_1^*, \ldots, \mathbf{x}_q^*, \mathbf{S}_i^*, \boldsymbol{\theta}_t^* = \underset{\mathbf{x}_1, \ldots, \mathbf{x}_q, \mathbf{S}, \boldsymbol{\theta}}{\text{argmax}} \ \ell_{\text{EULBO}}^{\text{RCaGP}}. \tag{17}$$

For efficiency, we impose the sparse block structure on the action matrices $\mathbf{S}_i \in \mathbb{R}^{n \times i}$, following [47]. Specifically, we enforce each block to be column vector $\mathbf{s}_j \in \mathbb{R}^{k \times 1}$, with $k = n/i$ entries, so that the number of trainable parameters is $k \times i = n$ (training data size). It is worth noticing that unlike EULBO with SVGP, which maximizes variational parameters and inducing locations, our approach directly optimizes the action matrices $\mathbf{S}_i$.

Maximizing the ELBO independently of the posterior-expected utility function can result in suboptimal data acquisition decisions, as the ELBO is primarily designed to model observed data [28]. In contrast, EULBO considers how the surrogate performs when selecting the next query, effectively guiding the solution of ELBO optimization towards high utility regions [29]. Our RCaGP model further balances such aggressive behavior by incorporating the combined uncertainty, preventing overconfidence during candidate query selection. Moreover, RCaGP enhances robustness in scenarios where function evaluations occasionally produce outliers.

## 4 Theoretical Analysis

This section presents a theoretical analysis of RCaGP. Proposition 4.1 establishes its robustness to outliers, while Proposition 4.2 links its uncertainty estimates to worst-case error, ensuring conservative predictions, thus addressing the challenges outlined in Section 1.

We demonstrate the robustness of RCaGP through the posterior influence function (PIF), similarly to [2, Proposition 3.2]. The PIF is a criterion proving robustness to misspecification in observation error [12, 27]. For this purpose, we define the dataset $\mathcal{D} = \{(\mathbf{x}_j, y_j)\}_{j=1}^n$ and the corresponding contaminated dataset $\mathcal{D}_m^c = (\mathcal{D} \setminus (\mathbf{x}_m, y_m)) \cup (\mathbf{x}_m, y_m^c)$, indexed by $m \in \{1, \ldots, n\}$. The impact of $y_m^c$ on inference is measured through PIF, expressed as Kullback-Leibler (KL) divergence between RCaGP's contaminated posterior $p(\mathbf{f}|\mathcal{D}_m^c)$ and its uncontaminated counterpart $p(\mathbf{f}|\mathcal{D})$:

$$\text{PIF}(y_m^c, \mathcal{D}) = \text{KL}(p(\mathbf{f}|\mathcal{D})\|p(\mathbf{f}|\mathcal{D}_m^c)). \tag{18}$$

PIF views the KL-divergence as a function of $|y_m^c - y_m|$. In principle, we can consider any divergence that is not uniformly bounded. Here, we choose KL-divergence since it provides a closed-form expression for multivariate Gaussian. A posterior is robust if $\sup_{y \in \mathcal{Y}} |\text{PIF}(y_m^c, \mathcal{D})| < \infty$ [2]. We then show that RCaGP's PIF is bounded through the following proposition:

**Proposition 4.1.** *Let* $\mathbf{f} \sim \mathcal{GP}(m, k)$ *denote the RCaGP prior, and let* $i \in 0, \ldots, n$ *represent the number of actions in RCaGP. Define constants* $C_1'$ *and* $C_2'$, *which are independent of* $y_m^c$. *For any* $i$ *and assuming* $\sup_{\mathbf{x}, y} w(\mathbf{x}, y) < \infty$, *the PIF of RCaGP is given by:*

$$\text{PIF}_{\text{RCaGP}}(y_m^c, \mathcal{D}) = C_1'(w(\mathbf{x}_m, y_m^c)^2 y_m^c)^2 + C_2'. \tag{19}$$

*Thus, if* $\sup_{\mathbf{x}, y} y \, w(\mathbf{x}, y)^2 < \infty$, *then RCaGP is robust since* $\sup_{y_m^c} |\text{PIF}_{\text{RCaGP}}(y_m^c, \mathcal{D})| < \infty$.

The constraint $\sup_{\mathbf{x},y} w(\mathbf{x}, y) < \infty$ ensures no observation has infinite weight, and $\sup_{\mathbf{x},y} y, w^2(\mathbf{x}, y) < \infty$ guarantees that $w$ down-weights observations at least at rate $1/y$. The proof sketch shows that each term in Equation (18) is bounded. We leverage the pseudo-inverse property of positive semi-definite matrices and matrix norm bounds on the low-rank approximation to relate RCGP and CaGP. The full proof is in Appendix D. Crucially, the PIF of CaGP is unbounded, indicating a lack of robustness to outliers (see Appendix F.1).

Next, we show that RCaGP's uncertainty captures the worst-case error over all latent functions, paralleling [46, Theorem 2]. We assume the difference between the latent function $g$ and the shrinkage term $m_w$ lies in an RKHS. This holds when $\|\nabla_y \log w^2(\mathbf{x}, y)\|_{\mathcal{H}_k}^2 < \infty$. In the case of our robust weight function, the correction term is a nonlinear rational function. Generally, such functions do not lie in the RKHS associated with common kernels (e.g., RBF, Matérn), unless specific and uncommon conditions are met—such as the RKHS being closed under composition with the given nonlinearity. A practical approach to ensure it belongs to the RKHS is to project the correction term onto the RKHS. Alternatively, one could use or design a kernel whose RKHS explicitly accommodates this class of nonlinear transformations. Under this assumption, we establish the link between RCaGP's uncertainty estimates and the worst-case error through the following proposition:

**Proposition 4.2.** *Let* $\hat{k}_i(\cdot, \cdot) = k_*(\cdot, \cdot) + \hat{\sigma}_i(\cdot, \cdot)$ *be the combined uncertainty of RCaGP with zero-mean prior* $m$. *Then, for any new* $\mathbf{x} \in \mathcal{X}$ *we have that*

$$\sup_{\|h\|_{\mathcal{H}_{k^w}} \leq 1} (h(\mathbf{x}) - \hat{\mu}_i^g(\mathbf{x}))^2 = \hat{k}_i(\mathbf{x}, \mathbf{x}) + \sigma_{\text{noise}}^2 \tag{20}$$

$$\sup_{\|h\|_{\mathcal{H}_{k^w}} \leq 1} (\mu_*^g(\mathbf{x}) - \hat{\mu}_i^g(\mathbf{x}))^2 = \hat{\sigma}_i(\mathbf{x}, \mathbf{x}) \tag{21}$$

*where* $\mu_*^g(\cdot) = k(\cdot, \mathbf{X})\tilde{\mathbf{K}}^{-1}h(\mathbf{X})$ *is the RCGP's posterior and* $\hat{\mu}_i^g(\cdot) = k(\cdot, \mathbf{X})\tilde{\mathbf{C}}_i(g(\mathbf{X}) - m_w(\mathbf{X}))$ *RCaGP's posterior mean for a function* $h \in \mathcal{H}_{k^w} = g(\mathbf{x}) - m_w(\mathbf{x})$ *with a latent function* $g : \mathcal{X} \to \mathbb{R}$ *and the shrinkage function* $m_w : \mathcal{X} \times \mathcal{Y} \to \mathbb{R}$.

The first equation in Proposition 4.2 shows that RCaGP's uncertainty $\hat{k}_i(.,.)$ captures the worst-case error between the latent function $g$ and the posterior mean $\hat{\mu}_i(.)$. Instead of pretending the model is fully correct, we allow a structured form of misspecification, namely, the part captured by $\mathbf{m}_w$. Proposition 4.2 then guarantees that any remaining discrepancy is still bounded by our variance. Even when classical Bayesian inference may fail to offer meaningful uncertainty due to model misspecification or outliers, RCaGP maintains a rigorous worst-case interpretation of its predictive variance, so long as the residual is sufficiently regular. This mirrors the CaGP assumption in spirit: just as CaGP analyzes worst-case error over functions in the RKHS, RCaGP does the same for the residual, which can be viewed as the portion of the signal that the robust update leaves uncorrected. The second shows that the computational uncertainty $\hat{\sigma}_i^2$ captures the worst-case error between the RCGP and RCaGP posterior means, induced by approximation actions such as inducing points. The proof follows [46], leveraging [21, Proposition 3.9] and the RKHS associated with RCGP. Full details are in Appendix E. Lastly, the convergence of RCaGP in terms of mean function in RKHS norm is established in Appendix F.2.

## 5 Expert-guided robust mean prior definition

Our proposed RCaGP enforces outlier robustness through the weight function $w$, which—as discussed in Section 3.1 and Proposition 4.2—critically depends on the prior mean $m$. Motivated by these insights, we propose defining a robust and informative mean prior using domain expert feedback.

We assume the expert can identify a subset of outliers in $\mathcal{D}_n$ and provide corrections. Instead of discarding these outliers, we use them to inform the prior: removing them would increase evaluation cost, while RCaGP can still benefit from them. Since expert corrections are imperfect, we model them probabilistically and define the mean via the inferred posterior.

Let $\mathbf{o} = \{o_j\}_{j=1}^n$ be binary outlier labels for dataset $\mathcal{D}_n = \{(\mathbf{x}_j, y_j)\}_{j=1}^n$, where $o_j = 1$ indicates an outlier. Each $o_j$ follows a Bernoulli likelihood $p(o_j|\delta_j)$ with latent probability $\delta_j \sim \mathcal{B}(\alpha_j, \beta_j)$. We infer $p(\delta_j|o_j)$ via Bayes' rule. Next, let $\bar{\mathbf{o}} = \{\bar{o}_o\}_{o=1}^{\mathring{o}}$ and $\hat{\mathbf{y}} = \{\hat{y}_o\}_{o=1}^{\mathring{o}}$ denote the identified outliers, and $\bar{\mathbf{y}} = \{\bar{y}_o\}_{o=1}^{\mathring{o}}$ the expert-provided corrections. Each correction $\bar{y}_o$ is drawn from $\mathcal{N}(\bar{\mu}_o, \sigma_{\text{corr.}}^2)$

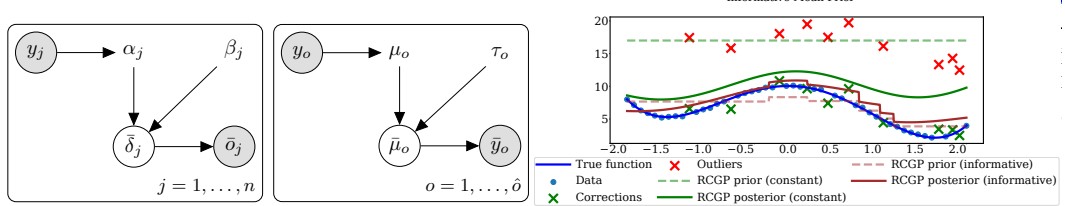

Figure 2: **Expert-guided robust mean prior.** **(Left)** Graphical model of expert's feedback generative process for identified outliers. **(Middle)** User graphical model for expert outlier corrections. **(Right)** Toy example: with expert corrections, the expert-guided prior better captures the true function than a constant mean prior under contamination.

with latent $\bar{\mu}_o \sim \mathcal{N}(\mu_o, \tau_o)$. Posterior inference yields $p(\bar{\mu}_o|\bar{y}_o)$. We define the mean prior by combining the expected labels and corrected values using a kernel-weighted average:

$$m(\mathbf{x}) = \frac{1}{\hat{n}} \sum_{\hat{j}=1}^{\hat{n}} \mathbb{E}_{p(\delta_{\hat{j}}|\bar{o}_{\hat{j}})}[\delta_{\hat{j}}] \mathbb{E}_{p(\bar{\mu}_{\hat{j}}|\bar{y}_{\hat{j}})}[\bar{\mu}_{\hat{j}}] l(\mathbf{x}, \mathbf{x}_j), \tag{22}$$

This corresponds to averaging over the $\hat{n}$ nearest identified outliers, weighted by a kernel $l$ and the expected confidence in each correction. In practice, we average over all identified outliers and use only posterior expectations, prioritizing computational efficiency over full Bayesian inference. The full generative process and toy illustration appear in Figure 2 (left and middle). Additional inference details are provided in Appendix G.

## 6 Related Work

**Outlier-robust Gaussian processes** fall into three main categories. The first replaces Gaussian noise with heavy-tailed alternatives—e.g., Student-$t$[20, 35], Huber[1], Laplace [23], data-dependent [13], or mixture models [8, 25]. These approaches break conjugacy and require approximate inference. The second class removes suspected outliers before applying standard inference [24, 32, 4], but this is often unreliable and computationally expensive in high dimensions. The third approach, RCGP [2], builds on generalized Bayesian inference [7, 19, 22] to down-weight outliers without discarding data. We adopt RCGP as the foundation for RCaGP due to its strong performance in regression with outliers, its preserved conjugacy, and greater computational efficiency—enabling more accurate posterior inference and better predictive performance.

**SVGPs** are the standard approach for scaling Gaussian processes to large-scale Bayesian optimization [14, 38, 44, 43]. [31] improved inducing point selection by optimizing their weighted Gram matrix via a determinantal point process. [29] formulated acquisition as expected utility, enabling joint optimization of queries, variational parameters, hyperparameters, and inducing points in an end-to-end fashion. [26] proposed an online conditioning strategy that updates SVGPs with new data without reoptimizing variational parameters, enabling look-ahead acquisition in BO.

**Computation-aware Gaussian processes** are a novel class of GPs that accounts for *computational uncertainty*, due to approximations in matrix-vector multiplications [46]. These methods fall under the domain of probabilistic numerics, which quantify additional uncertainties arising from approximation errors [33, 40, 34]. In that respect, [47] introduced a batch update algorithm for CaGP and proposed an ELBO objective to facilitate model selection. [16] extended CaGPs by proposing a calibrated probabilistic linear solver, resulting in a calibrated computation-aware GP model.

## 7 Experimental results

We evaluate the performance of our proposed RCaGP against concurrent baselines on a range of regression datasets, including a real-world dataset, followed by several high-throughput Bayesian Optimization tasks. We conclude by presenting insights gained after carrying out dedicated ablation studies. Our implementation is available at https://github.com/MarshalArijona/RCaGP.

**GP baselines.** We compare RCaGP against several baseline methods: RCSVGP [2], SVGP [17], and CaGP [47]. For the UCI regression benchmarks, we further include comparisons with SVGP models using a Student-$t$ likelihood [20] and SVGP with relevance pursuit (RRP), adapted from [3]

and formally derived in Section F.3. We set the number of actions (for RCaGP and CaGP) or inducing points (for SVGPs and RCSVGP) to 25. All baselines employ a Matérn-$5/2$ kernel and are implemented in `GPyTorch` [10]. Model hyperparameters are optimized by maximizing the ELBO (Equation 13).

**Outlier contamination protocol.** In our experiments, we follow the settings described by [2]. Unless stated otherwise in dedicated ablation studies, for regression datasets, we sample uniformly at random 10% of the training dataset input-output pairs $(\mathbf{x}_i, y_i)$, and replace the $y_i$'s by asymmetric outliers, i.e., *via* subtraction of noise sampled from a uniform distribution $\mathcal{U}(3\bar{\sigma}, 9\bar{\sigma})$, with $\bar{\sigma}$ being the standard deviation of the original observations. For BO tasks, while running the optimization loop, each evaluation has a $25\%$ chance of returning an asymmetric outlier, obtained by addition of noise sampled from a uniform distribution $\mathcal{U}(1\bar{\sigma}, 2\bar{\sigma})$, with $\bar{\sigma}$ being is the estimated standard deviation of the normalized evaluation function. Finally, for the weight function $w$ (Equation 12), we set the soft threshold $c = Q_n(|\mathbf{y} - \mathbf{m}|, 1 - \epsilon)$ with $\epsilon = 0.2$, with $Q_n$ the $(1 - \epsilon)$-quantile of $|\mathbf{y} - \mathbf{m}|$.

**Bayesian Optimization settings.** As described during Section 3.3, we integrate the baselines into EULBO, building on the implementation of [29]. We initialize the optimization process for all baselines with 250 data points sampled uniformly across the search space. All surrogates employ expected improvement (EI) as the acquisition function. Standard BO is applied across all tasks, while trust-region BO (TuRBO [9]) is additionally employed for high-dimensional tasks.

## 7.1 Regression on UCI datasets

We evaluate GP regression on four UCI datasets with asymmetric outliers (details in Section H.1); results are shown in the left part of Table 1. RCaGP delivers the overall best performance, combining strong predictive accuracy and well-calibrated uncertainty with acceptable runtimes, and being only surpassed in the Yacht dataset by RRP, a much slower baseline. Notably, it significantly outperforms both CaGP and RCSVGP in this setting, indicating that the integration of robustness and approximation-awareness is effective.

To test robustness across contamination types, we further consider two additional outlier scenarios, uniform and focused outliers, inspired by [2] (definitions in Section H.3). The corresponding results are shown in Table S1 and Table S2. RCaGP remains the top performer overall, particularly in uniform outliers for MAE, and focused outliers for NLL. These findings support its robustness across contamination types. More importantly, because outlier presence is not always known in advance, we also evaluate performance in the absence of outliers (right side of Table 1). RCaGP remains superior, except for the Energy dataset, where CaGP marginally outperforms RCaGP. This suggests that even in the absence of outliers, the weight function $w$ used by RCaGP (Equation 12) proves useful. An alternative explanation would be that these datasets already contain outliers from the start.

Overall, these results demonstrate that RCaGP is more than the sum of its parts: the combination of RCGP and CaGP is not merely additive, but synergistic, consistently outperforming either component alone, both in outlier-rich and clean regimes.

Table 1: **UCI Regression datasets results with asymmetric outliers and without outliers.** Average test set mean absolute error, negative log-likelihood, and clock-time (in seconds), with 1 std, for 20 train-test splits. Bolded results refer to the best baseline. Lower is better.

| | | Asymmetric outliers | | | | No outliers | | | |
|---|---|---|---|---|---|---|---|---|---|
| | | **Boston** | **Energy** | **Yacht** | **Parkinsons** | **Boston** | **Energy** | **Yacht** | **Parkinsons** |
| SVGP | MAE | 0.749 ± 0.062 | 0.607 ± 0.066 | 0.837 ± 0.103 | 0.766 ± 0.090 | 0.506 ± 0.048 | 0.407 ± 0.030 | 0.672 ± 0.059 | 0.701 ± 0.078 |
| | NLL | 1.442 ± 0.037 | 1.298 ± 0.036 | 1.460 ± 0.086 | 1.475 ± 0.062 | 1.335 ± 0.036 | 1.207 ± 0.017 | 1.358 ± 0.042 | 1.407 ± 0.063 |
| | clock-time | **1.480 ± 0.260** | **2.770 ± 0.070** | 0.960 ± 0.070 | **0.730 ± 0.170** | 1.570 ± 0.370 | **2.930 ± 0.310** | 0.950 ± 0.030 | **0.520 ± 0.030** |
| CaGP | MAE | 0.738 ± 0.059 | 0.562 ± 0.055 | 0.775 ± 0.078 | 0.798 ± 0.084 | 0.488 ± 0.042 | **0.334 ± 0.024** | 0.512 ± 0.041 | 0.674 ± 0.077 |
| | NLL | 1.3863 ± 0.0357 | 1.253 ± 0.027 | 1.395 ± 0.064 | 1.440 ± 0.068 | 1.278 ± 0.040 | **1.106 ± 0.011** | 1.263 ± 0.042 | 1.362 ± 0.067 |
| | clock-time | 2.410 ± 0.050 | 3.680 ± 0.080 | **0.720 ± 0.040** | 1.220 ± 0.080 | 2.470 ± 0.040 | 3.710 ± 0.080 | **0.730 ± 0.060** | 0.840 ± 0.010 |
| RCSVGP | MAE | 0.532 ± 0.050 | 0.566 ± 0.060 | 0.477 ± 0.049 | 0.627 ± 0.089 | 0.574 ± 0.056 | 0.612 ± 0.034 | 0.646 ± 0.076 | 0.648 ± 0.071 |
| | NLL | 1.298 ± 0.038 | 1.280 ± 0.034 | 1.241 ± 0.035 | 1.346 ± 0.070 | 1.332 ± 0.048 | 1.324 ± 0.033 | 1.322 ± 0.047 | 1.403 ± 0.076 |
| | clock-time | 1.500 ± 0.070 | 2.860 ± 0.060 | 1.010 ± 0.030 | 0.750 ± 0.020 | **1.560 ± 0.040** | 2.970 ± 0.120 | 1.010 ± 0.030 | 0.570 ± 0.010 |
| RCaGP | MAE | **0.477 ± 0.046** | **0.380 ± 0.038** | 0.435 ± 0.052 | **0.586 ± 0.084** | **0.428 ± 0.053** | 0.393 ± 0.024 | **0.436 ± 0.046** | **0.538 ± 0.078** |
| | NLL | **1.272 ± 0.036** | **1.162 ± 0.012** | 1.253 ± 0.043 | **1.317 ± 0.067** | **1.261 ± 0.049** | 1.160 ± 0.008 | **1.251 ± 0.040** | **1.311 ± 0.070** |
| | clock-time | 2.450 ± 0.110 | 3.790 ± 0.130 | 0.870 ± 0.060 | 1.250 ± 0.090 | 2.580 ± 0.150 | 3.780 ± 0.190 | 0.910 ± 0.070 | 0.920 ± 0.020 |
| Student-t | MAE | 0.962 ± 0.037 | 1.015 ± 0.028 | 0.907 ± 0.061 | 1.025 ± 0.079 | 0.896 ± 0.041 | 0.970 ± 0.025 | 0.815 ± 0.071 | 0.965 ± 0.074 |
| | NLL | 1.619 ± 0.039 | 1.606 ± 0.024 | 1.564 ± 0.067 | 1.658 ± 0.085 | 1.593 ± 0.063 | 1.572 ± 0.026 | 1.559 ± 0.100 | 1.627 ± 0.092 |
| | clock-time | 1.210 ± 0.240 | 1.800 ± 0.400 | 1.350 ± 0.260 | 0.990 ± 0.090 | 1.330 ± 0.050 | 1.330 ± 0.080 | 0.890 ± 0.150 | 0.730 ± 0.020 |
| RRP | MAE | 0.550 ± 0.050 | 0.597 ± 0.112 | **0.356 ± 0.051** | 0.644 ± 0.106 | 0.815 ± 0.103 | 0.775 ± 0.118 | 0.708 ± 0.035 | 1.052 ± 0.100 |
| | NLL | 1.147 ± 0.050 | 1.240 ± 0.117 | **0.896 ± 0.030** | 1.306 ± 0.121 | 1.527 ± 0.194 | 1.547 ± 0.214 | 1.179 ± 0.045 / | 1.884 ± 0.179 |
| | clock-time | 5.120 ± 0.100 | 3.840 ± 0.090 | 7.200 ± 0.160 | 3.010 ± 0.080 | 5.260 ± 0.120 | 3.730 ± 0.100 | 6.980 ± 0.110 | 3.010 ± 0.060 |

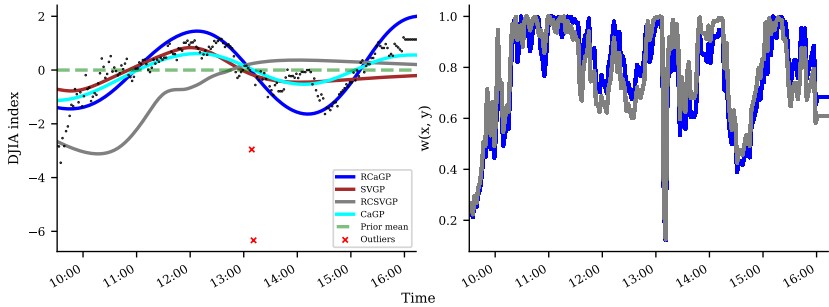

Figure 3: Left: Predictive mean of various approximate GP models on the DJIA index with a constant mean prior. Outliers affect the baselines, whereas RCaGP maintains robustness. Right: learned weight functions $w(\mathbf{x}, y)$ for RCaGP and RCSVGP, illustrating their influence on the regression.

## 7.2 Twitter Flash Crash

We illustrate the real-world applicability of RCaGP by evaluating its performance and other baselines on the Dow Jones Industrial Average (DJIA) index for April 17, 2013. On this day, the Associated Press Twitter account was hacked, posting a false report of an explosion at the White House and an injury to the U.S. President. The resulting panic triggered a sudden market sell-off, followed by a rapid rebound, creating a brief interval during which the DJIA did not reflect the true economic state of the U.S. stock market. This scenario provides a unique stress test for evaluating model robustness under sudden, anomalous market conditions. Observations during this period can reasonably be treated as outliers. Figure 3 (left) shows the DJIA index alongside the predictive means of RCaGP and other baselines. The predictive means of SVGP, RCSVGP, and CaGP exhibit noticeable deviations before or after 1 PM, while RCaGP yields a more reliable estimate, with only minor discrepancies around 2 PM. These results suggest that insufficient robustness or computational uncertainty in the baseline models contributes to their degraded performance under sudden market shocks. In the right panel of Figure 3, we plot the weight functions of RCaGP and RCSVGP. Although their weight functions exhibit similar profiles, they yield markedly different mean predictions. This supports our claim that robustness alone, without an appropriate treatment of computational uncertainty, can degrade the performance of approximate Gaussian process models.

## 7.3 High-Throughput Bayesian Optimization

We consider 4 tasks: Hartmann6D, Lunar12D, Rover60D and Lasso-DNA180D (Section H.2). Figure 4 reports the best value found during BO, averaged over 20 trials, with ±1 standard deviation. To ensure fair comparisons despite outlier contamination, we report the uncontaminated mean best value. In the top row (columns 1–4), using EULBO-EI and under asymmetric outliers, RCaGP clearly outperforms all baselines on Lasso DNA 180D, and ranks first or on par with CaGP on Hartmann 6D, Lunar12D, and Rover60D. SVGP and RCSVGP display the same performance. The bottom row (columns 1–4) shows results with Determinantal Point Process (DPP)-based inducing point selection [31], still in the case of asymmetric outliers. RCaGP remains the top performer on DNA and performs similarly to CaGP on other tasks, with RCSVGP slightly ahead on Hartmann. Substituting EI with Thompson Sampling in DPP-BO leads to equal performance across baselines (Figure S1).

Next, Figure S2 presents results with TurBO, a BO variant tailored to high dimensions [9]. RCaGP exhibits strong average performance, and notably, it is the only method that never ranks last in a statistically significant manner. Lastly, as one might not know beforehand whether outliers will affect the trial, column 5 in Figure 4 shows results for EULBO EI on Hartmann and Lunar without outliers. RCaGP maintains a clear advantage on Hartmann, while all baselines perform similarly on Lunar.

Together, these results highlight RCaGP's versatility and robustness across inducing point allocation strategies (EULBO, DPP), acquisition functions (EI, TS), BO behavior (global BO, trust-region BO), and noise regimes (with or without outliers). Its consistent superiority supports the central claim: combining robustness from RCGP with the uncertainty calibration of CaGP leads to a synergistic improvement, not just an additive one, across a wide range of BO challenges.

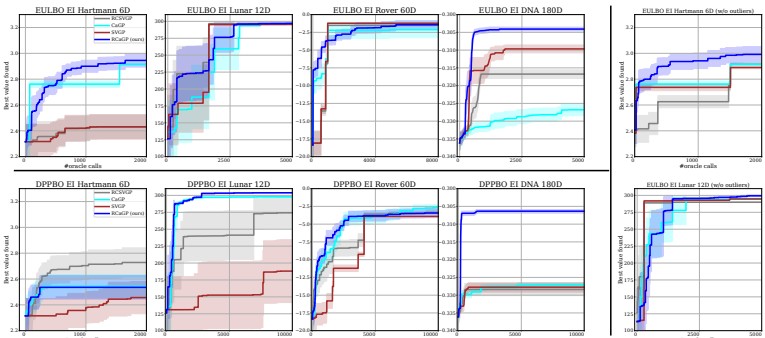

Figure 4: **High-throughput Bayesian Optimization task.** Each panel shows the best value found each iteration found so far, averaged across 20 repetitions $\pm$ 1 std. Columns 1-4 report results under asymmetric outliers contamination, with the $1^{st}$ row using the EULBO-EI acquisition function, the $2^{nd}$ row using DPPBO-EI. Column 5 features results without outliers using EULBO-EI.

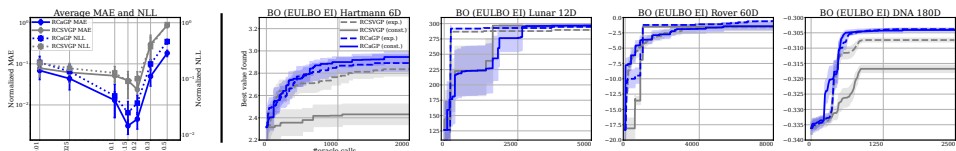

Figure 5: **Ablation studies. (Left)** Varying $c$ in weight function $w$ (Equation 12) on UCI regression datasets. MAE and NLL have been normalized for each dataset; the metrics displayed represent the average. **(Right)** BO results for RCaGP and RCSVGP using expert-driven prior mean function $m$ in $w$. Results are averaged over across 20 repetitions, with $\pm$ 1 standard deviation being shown.

## 7.4 Ablation studies

**Effect of the soft threshold $c$ in the weight function.** We evaluate RCaGP's sensitivity to the threshold $c = Q_n(|\mathbf{y} - \mathbf{m}|, 1 - \epsilon)$, defined as the $(1 - \epsilon)$-quantile of $|\mathbf{y} - \mathbf{m}|$ and controlling how many values are treated as outliers, by varying $\epsilon$. Figure 5 (left) shows MAE and NLL averaged over 4 UCI datasets (normalized; per-dataset results in Figure S3). RCaGP consistently achieves lower error across all $\epsilon$, with best results at $0.15$, suggesting treating roughly 15% of the data as potential outliers is optimal. This aligns well with the naturally present and 10% injected asymmetric outliers.

**Effect of the mean $m$ in the weight function.** Given the central role of the mean function $m$ in $w$ (Equation 12), we compare a constant prior to the expert-driven one from Section 5, assuming perfect outlier correction. This yields four variants: RCaGP/RCSVGP with either constant (const.) or expert (exp.) mean. As shown in Figure 5 (right), the informed prior greatly boosts RCSVGP across BO tasks, and modestly accelerates RCaGP on Lunar12D. See Appendix G for further analysis.

## 8 Conclusion

This paper introduced Robust Computation-Aware Gaussian Processes (RCaGP), a principled framework unifying robust inference and approximation-aware modeling coherently rather than tackling these challenges independently. Our theoretical and empirical results show that RCaGP yields conservative uncertainty estimates, is provably robust to outliers, and consistently outperforms methods addressing only one challenge. Additionally, we proposed an expert-guided prior mean to further enhance robustness in practice. RCaGP represents a meaningful step forward in probabilistic numerics, enabling robust and trustworthy inference for complex, large-scale regression and optimization tasks.

**Limitations.** While RCaGP demonstrates strong performance across diverse regression and BO tasks, it inherits certain practical limitations. Like CaGP, it requires full batching, leading to increased GPU memory demands compared to fully stochastic approaches like SVGP. Its performance also depends on manually selecting the projection dimensionality, which cannot be tuned via standard GP model selection techniques. Lastly, broader evaluations across more complex domains remain an avenue for future work.

**Acknowledgments.** This research was supported by the Research Council of Finland (flagship programme: Finnish Center for Artificial Intelligence, FCAI grants 358958, 345604, and 341763), and the UKRI Turing AI World-Leading Researcher Fellowship, EP/W002973/1. We also acknowledge the computational resources provided by the Aalto Science-IT Project.

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

# Supplementary Material - Robust and Computation-Aware Gaussian Processes

The appendix is organized as follows:

- Appendix A gives the results of additional experiments mentioned in the main text. These include experiments on UCI regression datasets with uniform and focused outliers (Table S1 and Table S2), BO results for DPP-BO-TS (Figure S1), TuRBO (Figure S2), and dataset-specific results for the ablation study conducted in Section 7.4, where $c$ is varied.

- Appendix B gives the computational complexity of RCaGP.

- Appendix C formally derives the ELBO for RCaGP introduced in Section 3.2.

- Appendix D contains the proof for the robustness property presented in Proposition 4.1.

- Appendix E proves Proposition 4.2, showing that RCaGP's uncertainty estimates capture the worse-case error over all latent functions.

- Appendix F contains additional theoretical results: a proposition showing the convergence in mean of RCaGP in RKHS norm (Section F.2), and a proposition showing the lack of robustness of CaGP (Section F.1).

- Appendix G provides further details regarding the expert-driven prior mean introduced in Section 5 and employed during the ablation study in Section 7.4.

- Appendix H provides additional details about the UCI regression datasets (Section H.1), the test functions employed in BO (Section H.2), and the outlier-contamination protocols studied in this work (Section H.3).

- Appendix I provides tables that report the hyperparmeters used for UCI regression and high-throughput BO experiments (Section I.2) and the description of the computing resources used for our experiments (Section I.1).

## A   Additional experiments

The results of the UCI regression experiments in the presence of uniform and focused outliers are presented in Table S1 and Table S2, respectively. The MAE and NLL metrics indicate that RCaGP consistently outperforms the baseline methods across most datasets. These findings demonstrate the robustness and versatility of RCaGP in handling different types of outliers. We further conducted experiments using the radial basis function (RBF) kernel on both clean and contaminated datasets. While RCaGP achieves competitive MAE and NLL performance compared to RCSVGP on clean data, it substantially outperforms all baselines on contaminated data. However, this performance improvement comes at the cost of increased computational time compared to the baselines.

The results of Bayesian optimization (BO) experiments using the Thompson Sampling (TS) acquisition function (AF) and Trust-Region BO (TuRBO) are presented in Figure S1 and Figure S2, respectively. In experiments utilizing the TF acquisition function, no model demonstrates a statistically significant performance advantage. Furthermore, the best values identified by these models are generally lower than those obtained with the BO-EI method. We hypothesize that this performance degradation arises from the reliance of the TS approach on posterior sampling, which may reduce the effectiveness of the robustness properties inherent in RCaGP. Subsequent BO experiments using the TuRBO framework show improved performance across all models and tasks—except for the Lunar task. In this case, TuRBO yields lower results for all models. This suggests that the presence of outliers in the Lunar task may impede TuRBO's ability to effectively explore and identify the global optimum.

The MAE and NLL plots for varying values of $\epsilon$ in the weight function $w$ are shown in Figure S3. Across all datasets, RCaGP consistently yields lower MAE and NLL values than RCSVGP, indicating superior predictive performance. Furthermore, both metrics improve when $\epsilon$ is set to 0.1 or 0.15, but begin to deteriorate once $\epsilon$ exceeds 0.2. This trend suggests that aligning $\epsilon$ with the true proportion of outliers in the data enhances model robustness and overall performance.

Table S1: **UCI Regression datasets results in the presence of uniform outliers with $\epsilon = 0.1$.** Average test set mean absolute error, negative log-likelihood, and clock-time (in seconds), with 1 std, for 20 train-test splits. Bolded results refer to the best baseline. Lower is better.

|  |  | **Boston** | **Energy** | **Yacht** | **Parkinsons** |
|---|---|---|---|---|---|
| SVGP | MAE | 0.799 ± 0.054 | 0.918 ± 0.070 | 0.899 ± 0.099 | 0.837 ± 0.093 |
|  | NLL | 1.511 ± 0.047 | 1.486 ± 0.042 | 1.558 ± 0.095 | 1.523 ± 0.057 |
|  | clock-time | **1.580 ± 0.050** | **2.680 ± 0.100** | 1.000 ± 0.080 | **0.800 ± 0.010** |
| CaGP | MAE | 0.782 ± 0.042 | **0.908 ± 0.046** | 0.861 ± 0.094 | 0.826 ± 0.087 |
|  | NLL | 1.483 ± 0.047 | **1.477 ± 0.029** | 1.542 ± 0.098 | 1.479 ± 0.073 |
|  | clock-time | 2.570 ± 0.040 | 3.640 ± 0.220 | **0.810 ± 0.280** | 0.960 ± 0.040 |
| RCSVGP (const.) | MAE | 0.763 ± 0.065 | 0.921 ± 0.082 | 0.799 ± 0.095 | 0.8567 ± 0.1017 |
|  | NLL | 1.482 ± 0.066 | 1.484 ± 0.059 | **1.496 ± 0.089** | 1.5494 ± 0.0753 |
|  | clock-time | 1.620 ± 0.040 | 2.700 ± 0.060 | 1.050 ± 0.030 | 0.830 ± 0.020 |
| RCaGP (const.) | MAE | **0.743 ± 0.053** | 0.910 ± 0.046 | **0.762 ± 0.097** | **0.811 ± 0.077** |
|  | NLL | **1.475 ± 0.062** | 1.490 ± 0.036 | 1.533 ± 0.099 | **1.469 ± 0.075** |
|  | clock-time | 2.630 ± 0.040 | 3.770 ± 0.110 | 0.960 ± 0.050 | 1.080 ± 0.020 |

Table S2: **UCI Regression datasets results in the presence of focused outliers with $\epsilon = 0.1$.** Average test set mean absolute error, negative log-likelihood, and clock-time (in seconds), with 1 std, for 20 train-test splits. Bolded results refer to the best baseline. Lower is better.

|  |  | **Boston** | **Energy** | **Yacht** | **Parkinsons** |
|---|---|---|---|---|---|
| SVGP | MAE | 0.627 ± 0.051 | **0.337 ± 0.021** | 0.630 ± 0.053 | 0.672 ± 0.078 |
|  | NLL | 1.405 ± 0.031 | 1.239 ± 0.009 | **1.255 ± 0.057** | 1.433 ± 0.053 |
|  | clock-time | **1.500 ± 0.050** | **2.800 ± 0.080** | 1.000 ± 0.060 | **0.770 ± 0.020** |
| CaGP | MAE | 0.677 ± 0.061 | 0.436 ± 0.051 | 0.525 ± 0.050 | 0.750 ± 0.073 |
|  | NLL | 1.425 ± 0.055 | 1.181 ± 0.03 | 1.277 ± 0.048 | 1.462 ± 0.064 |
|  | clock-time | 2.460 ± 0.060 | 3.670 ± 0.070 | **0.830 ± 0.290** | 0.930 ± 0.010 |
| RCSVGP (const.) | MAE | 0.709 ± 0.100 | 0.634 ± 0.070 | 0.666 ± 0.072 | **0.551 ± 0.079** |
|  | NLL | 1.419 ± 0.088 | 1.436 ± 0.096 | 1.321 ± 0.087 | 1.392 ± 0.068 |
|  | clock-time | 1.550 ± 0.040 | 2.850 ± 0.070 | 1.040 ± 0.040 | 0.790 ± 0.030 |
| RCaGP (const.) | MAE | **0.505 ± 0.069** | 0.390 ± 0.023 | **0.437 ± 0.053** | 0.600 ± 0.082 |
|  | NLL | **1.323 ± 0.073** | **1.163 ± 0.009** | 1.268 ± 0.051 | **1.374 ± 0.082** |
|  | clock-time | 2.570 ± 0.050 | 3.780 ± 0.110 | 0.930 ± 0.060 | 1.010 ± 0.040 |

Table S3: **UCI Regression datasets results with RBF kernel.** Average test set mean absolute error, negative log-likelihood, and clock-time (in seconds), with 1 std, for 20 train-test splits. Bolded results refer to the best baseline. Lower is better.

| | | Asymmetric outliers | | | | No outliers | | | |
|---|---|---|---|---|---|---|---|---|---|
| | | **Boston** | **Energy** | **Yacht** | **Parkinsons** | **Boston** | **Energy** | **Yacht** | **Parkinsons** |
| SVGP | MAE | 1.145 ± 0.052 | 1.138 ± 0.040 | 1.235 ± 0.044 | 1.002 ± 0.100 | 0.731 ± 0.052 | 0.908 ± 0.026 | 0.767 ± 0.063 | 0.811 ± 0.078 |
| | NLL | 1.691 ± 0.043 | 1.736 ± 0.037 | 1.732 ± 0.041 | 1.590 ± 0.079 | 1.433 ± 0.053 | 1.430 ± 0.023 | 1.420 ± 0.078 | 1.448 ± 0.070 |
| | clock-time | **0.930 ± 0.280** | **1.170 ± 0.070** | 0.650 ± 0.060 | **0.720 ± 0.260** | **0.940 ± 0.030** | **1.230 ± 0.150** | 0.870 ± 0.240 | **0.670 ± 0.130** |
| CaGP | MAE | 1.060 ± 0.095 | 0.971 ± 0.090 | 1.133 ± 0.092 | 0.955 ± 0.110 | 0.488 ± 0.042 | 0.334 ± 0.024 | 0.512 ± 0.042 | 0.674 ± 0.077 |
| | NLL | 1.611 ± 0.086 | 1.535 ± 0.071 | 1.679 ± 0.088 | 1.556 ± 0.080 | 1.278 ± 0.040 | **1.106 ± 0.011** | 1.263 ± 0.042 | 1.362 ± 0.067 |
| | clock-time | 2.470 ± 0.060 | 3.540 ± 0.080 | **0.640 ± 0.100** | 0.870 ± 0.080 | 2.410 ± 0.060 | 3.540 ± 0.160 | **0.770 ± 0.060** | 0.830 ± 0.100 |
| RCSVGP | MAE | **0.680 ± 0.122** | 0.456 ± 0.041 | 0.733 ± 0.114 | **0.840 ± 0.187** | 0.512 ± 0.074 | 0.558 ± 0.035 | 0.640 ± 0.074 | 0.540 ± 0.083 |
| | NLL | **1.364 ± 0.080** | 1.200 ± 0.026 | **1.518 ± 0.116** | 1.381 ± 0.084 | 1.302 ± 0.063 | 1.300 ± 0.034 | 1.318 ± 0.047 | 1.417 ± 0.065 |
| | clock-time | 1.500 ± 0.090 | 2.560 ± 0.100 | 0.970 ± 0.090 | 0.970 ± 0.090 | 1.530 ± 0.040 | 2.780 ± 0.120 | 1.130 ± 0.080 | 0.720 ± 0.060 |
| RCaGP | MAE | 0.733 ± 0.156 | **0.353 ± 0.053** | **0.651 ± 0.095** | 0.913 ± 0.198 | **0.379 ± 0.050** | **0.326 ± 0.023** | **0.365 ± 0.035** | **0.534 ± 0.078** |
| | NLL | 1.405 ± 0.118 | **1.173 ± 0.024** | 1.531 ± 0.152 | **1.241 ± 0.046** | 1.144 ± 0.007 | **1.226 ± 0.034** | **1.307 ± 0.068** |
| | clock-time | 2.550 ± 0.060 | 3.590 ± 0.090 | 0.790 ± 0.060 | 0.900 ± 0.030 | 2.470 ± 0.030 | 3.740 ± 0.170 | 0.940 ± 0.070 | 1.070 ± 0.240 |
| Student-t | MAE | 0.962 ± 0.037 | 1.015 ± 0.028 | 0.907 ± 0.061 | 1.025 ± 0.079 | 0.896 ± 0.041 | 0.970 ± 0.025 | 0.815 ± 0.071 | 0.965 ± 0.074 |
| | NLL | 1.619 ± 0.039 | 1.606 ± 0.024 | 1.564 ± 0.067 | 1.658 ± 0.085 | 1.593 ± 0.063 | 1.572 ± 0.026 | 1.559 ± 0.100 | 1.627 ± 0.092 |
| | clock-time | 1.030 ± 0.040 | 1.350 ± 0.070 | 0.790 ± 0.060 | 0.730 ± 0.040 | 1.090 ± 0.030 | 1.460 ± 0.060 | 0.920 ± 0.050 | 0.730 ± 0.010 |

# B   Computational Complexity

Assuming a constant mean prior, RCaGP exhibits the same time and memory complexity as CaGP, as both rely on the same batch update algorithm. Note that the weight function requires $\mathcal{O}(n)$ for time

and space complexity, thus it does not affect the complexity of the algorithm. The time complexity of RCaGP is $\mathcal{O}(n\,i\,\max(i,k))$, where $k = n/i$ is the number of non-zero entries for each column $\mathbf{S}_i$. In addition, RCaGP and CaGP have linear memory complexity $\mathcal{O}(ni)$.

## C   Evidence Lower Bound for RCaGP

**Proposition C.1.** *Define the variational family*

$$\mathcal{Q} \triangleq \{q_i(\mathbf{f}) = \mathcal{N}(\mathbf{f}; \hat{\mu}_i(\mathbf{X}), \hat{k}_i(\mathbf{X}, \mathbf{X})) | \mathbf{S}_i \in \mathbb{R}^{n \times i}\} \tag{S1}$$

*and robust loss function*

$$L_n^w(\mathbf{f}, \mathbf{y}, \mathbf{X}) = \frac{1}{2n}\left(\mathbf{f}^\top \sigma_{\text{noise}}^{-2} \mathbf{J}_\mathbf{w}^{-1} \mathbf{f} - 2\mathbf{f}^\top \nu + C(\mathbf{x}, \mathbf{y}, \sigma_{\text{noise}}^2)\right), \tag{S2}$$

*where $\nu = \sigma_{\text{noise}}^{-2} \mathbf{J}_\mathbf{w}^{-1}(\mathbf{y} - \mathbf{m_w})$ and $C(\mathbf{x}, \mathbf{y}, \sigma_{\text{noise}}^2) = \mathbf{y}^\top \sigma_{\text{noise}}^{-2}\text{diag}(\mathbf{w}^2)\mathbf{y} - 2\nabla_y \mathbf{y}^\top \mathbf{w}^2$. Then, the evidence lower bound of RCaGP is given by*

$$\ell_{\text{ELBO}}^{\text{RCaGP}} = \mathbb{E}_{q_i(\mathbf{f})}[\log p^w(\mathbf{y}|\mathbf{f})] - \text{KL}[q_i(\mathbf{f})\|p(\mathbf{f})] \tag{S3}$$

$$= -\frac{1}{2}\text{tr}(\sigma_{\text{noise}}^{-2}\mathbf{J}_\mathbf{w}^{-1/2}\hat{k}_i(\mathbf{X}, \mathbf{X})\mathbf{J}_\mathbf{w}^{-1/2}) - \frac{1}{2}\text{tr}(\sigma_{\text{noise}}^{-2}\mathbf{J}_\mathbf{w}^{-1/2}\hat{\mu}_i(\mathbf{X})^\top \mathbf{J}_\mathbf{w}^{-1/2}\hat{\mu}_i(\mathbf{X})) + \hat{\mu}_i(\mathbf{X})^\top \nu$$

$$- \frac{1}{2}C(\mathbf{x}, \mathbf{y}, \sigma_{\text{noise}}^2) - \frac{1}{2}(\bar{\mathbf{v}}_i^\top \mathbf{S}_i^\top \mathbf{K}\mathbf{S}_i\bar{\mathbf{v}}_i + \log\det(\mathbf{S}_i^\top \tilde{\mathbf{K}}\mathbf{S}_i) - i\log(\sigma_{\text{noise}}^2)$$

$$- \log\det(\mathbf{S}_i^\top \mathbf{S}_i) - \log\det(\mathbf{J}_\mathbf{w}) - \text{tr}((\mathbf{S}_i^\top \tilde{\mathbf{K}}\mathbf{S}_i)^{-1}\mathbf{S}_i^\top \mathbf{K}\mathbf{S}_i)) \tag{S4}$$

*where $\bar{\mathbf{v}}_i = (\mathbf{S}^\top \tilde{\mathbf{K}}\mathbf{S})^{-1}\mathbf{S}^\top(\mathbf{y} - \mathbf{m_w})$ and $p^w(\mathbf{y}|\mathbf{f}) = \exp(-nL_n^w(\mathbf{f}, \mathbf{y}, \mathbf{X}))$ are the projected representer weights and the pseudo-likelihood, respectively.*

*Proof:*

The ELBO is given by

$$\ell_{\text{ELBO}}^{\text{RCaGP}} = \mathbb{E}_{q_i(\mathbf{f})}[\log p^w(\mathbf{y}|\mathbf{f})] - \text{KL}[q_i(\mathbf{f})\|p(\mathbf{f})] \tag{S5}$$

We first compute the expected loss function term:

$$\mathbb{E}_{q_i(\mathbf{f})}[\log p^w(\mathbf{y}|\mathbf{f})] = \int \log\exp(-nL_n^w(\mathbf{f}, \mathbf{y}, \mathbf{X}))q_i(\mathbf{f})d\mathbf{f} \tag{S6}$$

$$= \int -\frac{1}{2}(\mathbf{f}^\top \sigma_{\text{noise}}^{-2}\mathbf{J}_\mathbf{w}^{-1}\mathbf{f} - 2\mathbf{f}^\top \nu + C(\mathbf{x}, \mathbf{y}, \sigma_{\text{noise}}^2))q_i(\mathbf{f})d\mathbf{f} \tag{S7}$$

$$= \int -\frac{1}{2}\mathbf{f}^\top \sigma_{\text{noise}}^{-2}\mathbf{J}_\mathbf{w}^{-1}\mathbf{f}\, q_i(\mathbf{f})\, d\mathbf{f} + \int \mathbf{f}^\top \nu\, q_i(\mathbf{f})\, d\mathbf{f}$$

$$- \int \frac{1}{2}C(\mathbf{x}, \mathbf{y}, \sigma_{\text{noise}}^2)\, q_i(\mathbf{f})d\mathbf{f} \tag{S8}$$

$$= -\frac{1}{2}\int \mathbf{f}^\top \sigma_{\text{noise}}^{-2}\mathbf{J}_\mathbf{w}^{-1}\mathbf{f}\, q_i(\mathbf{f})\, d\mathbf{f} + \mathbb{E}_{q_i(\mathbf{f})}[\mathbf{f}^\top \nu] - \frac{1}{2}C(\mathbf{x}, \mathbf{y}, \sigma_{\text{noise}}^2), \tag{S9}$$

Next, we apply the identity $\mathbf{f}^\top \sigma_{\text{noise}}^{-2}\mathbf{J}_\mathbf{w}^{-1}\mathbf{f} = \text{tr}(\sigma_{\text{noise}}^{-1}\mathbf{J}_\mathbf{w}^{-1/2}\mathbf{f}\mathbf{f}^\top \sigma_{\text{noise}}^{-1}\mathbf{J}_\mathbf{w}^{-1/2})$ (see [2]):

$$\mathbb{E}_{q_i(\mathbf{f})}[\log p^w(\mathbf{y}|\mathbf{f})] = -\frac{1}{2}\mathbb{E}_{q_i(\mathbf{f})}[\text{tr}(\sigma_{\text{noise}}^{-1}\mathbf{J}_\mathbf{w}^{-1/2}\mathbf{f}\mathbf{f}^\top \sigma_{\text{noise}}^{-1}\mathbf{J}_\mathbf{w}^{-1/2})] + \mathbb{E}_{q_i(\mathbf{f})}[\mathbf{f}^\top \nu] - \frac{1}{2}C(\mathbf{x}, \mathbf{y}, \sigma_{\text{noise}}^2) \tag{S10}$$

Since the expectation of the trace is equal to the trace of the expectation, we find that

$$\mathbb{E}_{q_i(\mathbf{f})}[\log p^w(\mathbf{y}|\mathbf{f})] = -\frac{1}{2}\text{tr}(\mathbb{E}_{q_i(\mathbf{f})}[\sigma_{\text{noise}}^{-1}\mathbf{J}_\mathbf{w}^{-1/2}\mathbf{f}\mathbf{f}^\top \sigma_{\text{noise}}^{-1}\mathbf{J}_\mathbf{w}^{-1/2}]) + \mathbb{E}_{q_i(\mathbf{f})}[\mathbf{f}^\top \nu] - \frac{1}{2}C(\mathbf{x}, \mathbf{y}, \sigma_{\text{noise}}^2) \tag{S11}$$

$$= -\frac{1}{2}\text{tr}(\mathbb{E}_{q_i(\mathbf{f})}[\sigma_{\text{noise}}^{-1}\mathbf{J}_\mathbf{w}^{-1/2}\mathbf{f}\mathbf{f}^\top \sigma_{\text{noise}}^{-1}\mathbf{J}_\mathbf{w}^{-1/2}]) + \mathbb{E}_{q_i(\mathbf{f})}[\mathbf{f}]^\top \nu - \frac{1}{2}C(\mathbf{x}, \mathbf{y}, \sigma_{\text{noise}}^2) \tag{S12}$$

By definition of mean and variance, we have that $\mathbb{E}_{p(\mathbf{x})}[\mathbf{x}\mathbf{x}^\top] = \mathbb{V}_{p(\mathbf{x})}[\mathbf{x}] + \mathbb{E}_{p(\mathbf{x})}[\mathbf{x}]\,\mathbb{E}_{p(\mathbf{x})}[\mathbf{x}]^\top$. Applying this identity, we obtain

$$\mathbb{E}_{q_i(\mathbf{f})}[\log p^w(\mathbf{y}|\mathbf{f})] = -\frac{1}{2}\mathrm{tr}(\sigma_{\mathrm{noise}}^{-1}\mathbf{J}_{\mathbf{w}}^{-1/2}\mathbb{E}_{q_i(\mathbf{f})}[\mathbf{f}\mathbf{f}^\top]\sigma_{\mathrm{noise}}^{-1}\mathbf{J}_{\mathbf{w}}^{-1/2}) + \mathbb{E}_{q_i(\mathbf{f})}[\mathbf{f}]^\top\nu - \frac{1}{2}C(\mathbf{x},\mathbf{y},\sigma_{\mathrm{noise}}^2) \tag{S13}$$

$$= -\frac{1}{2}\mathrm{tr}(\sigma_{\mathrm{noise}}^{-1}\mathbf{J}_{\mathbf{w}}^{-1/2}(\mathbb{V}_{q_i(\mathbf{f})}[\mathbf{f}] + \mathbb{E}_{q_i(\mathbf{f})}[\mathbf{f}]\mathbb{E}_{q_i(\mathbf{f})}[\mathbf{f}]^\top)\sigma_{\mathrm{noise}}^{-1}\mathbf{J}_{\mathbf{w}}^{-1/2})$$
$$+ \mathbb{E}_{q_i(\mathbf{f})}[\mathbf{f}]^\top\nu - \frac{1}{2}C(\mathbf{x},\mathbf{y},\sigma_{\mathrm{noise}}^2) \tag{S14}$$

$$= -\frac{1}{2}\mathrm{tr}(\sigma_{\mathrm{noise}}^{-1}\mathbf{J}_{\mathbf{w}}^{-1/2}(\hat{k}_i(\mathbf{X},\mathbf{X}) + \hat{\mu}_i(\mathbf{X})\hat{\mu}_i(\mathbf{X})^\top)\sigma_{\mathrm{noise}}^{-1}\mathbf{J}_{\mathbf{w}}^{-1/2}) + \hat{\mu}_i(\mathbf{X})^\top\nu$$
$$- \frac{1}{2}C(\mathbf{x},\mathbf{y},\sigma_{\mathrm{noise}}^2) \tag{S15}$$

$$= -\frac{1}{2}\mathrm{tr}(\sigma_{\mathrm{noise}}^{-2}\mathbf{J}_{\mathbf{w}}^{-1/2}\hat{k}_i(\mathbf{X},\mathbf{X})\mathbf{J}_{\mathbf{w}}^{-1/2}) - \frac{1}{2}\mathrm{tr}(\sigma_{\mathrm{noise}}^{-2}\mathbf{J}_{\mathbf{w}}^{-1/2}\hat{\mu}_i(\mathbf{X})\hat{\mu}_i(\mathbf{X})^\top\mathbf{J}_{\mathbf{w}}^{-1/2})$$
$$+ \hat{\mu}_i(\mathbf{X})^\top\nu - \frac{1}{2}C(\mathbf{x},\mathbf{y},\sigma_{\mathrm{noise}}^2) \tag{S16}$$

$$= -\frac{1}{2}\mathrm{tr}(\sigma_{\mathrm{noise}}^{-2}\mathbf{J}_{\mathbf{w}}^{-1/2}\hat{k}_i(\mathbf{X},\mathbf{X})\mathbf{J}_{\mathbf{w}}^{-1/2}) - \frac{1}{2}\mathrm{tr}(\sigma_{\mathrm{noise}}^{-2}\hat{\mu}_i(\mathbf{X})^\top\mathbf{J}_{\mathbf{w}}^{-1}\hat{\mu}_i(\mathbf{X}))$$
$$+ \hat{\mu}_i(\mathbf{X})^\top\nu - \frac{1}{2}C(\mathbf{x},\mathbf{y},\sigma_{\mathrm{noise}}^2) \tag{S17}$$

Next, we compute the KL term, where both $q(\mathbf{f})$ and $p(\mathbf{f})$ are multivariate Gaussian:

$$\mathrm{KL}[q_i(\mathbf{f}) \,\|\, p(\mathbf{f})] = \frac{1}{2}((\hat{\mu}_i(\mathbf{X}) - m(\mathbf{X}))^\top\mathbf{K}^{-1}(\hat{\mu}_i(\mathbf{X}) - m(\mathbf{X})) + \log\left(\frac{\det(\mathbf{K})}{\det(\hat{k}_i(\mathbf{X},\mathbf{X}))}\right)$$
$$+ \mathrm{tr}(\mathbf{K}^{-1}\hat{k}_i(\mathbf{X},\mathbf{X})) - n) \tag{S18}$$

$$= \frac{1}{2}((\mathbf{K}\tilde{\mathbf{C}}_i(\mathbf{y} - m_w(\mathbf{X})))^\top\mathbf{K}^{-1}(\mathbf{K}\tilde{\mathbf{C}}_i(\mathbf{y} - m_w(\mathbf{X}))) - \log\det(\mathbf{K}^{-1}\hat{k}_i(\mathbf{X},\mathbf{X}))$$
$$+ \mathrm{tr}(\mathbf{I}_{n\times n} - \tilde{\mathbf{C}}_i\mathbf{K}) - n) \tag{S19}$$

$$= \frac{1}{2}((\mathbf{y} - m_w(\mathbf{X}))^\top\tilde{\mathbf{C}}_i\mathbf{K}\tilde{\mathbf{C}}_i(\mathbf{y} - m_w(\mathbf{X})) - \log\det(\mathbf{I}_{n\times n} - \tilde{\mathbf{C}}_i\mathbf{K})$$
$$+ \mathrm{tr}(\mathbf{I}_{n\times n} - \tilde{\mathbf{C}}_i\mathbf{K}) - n) \tag{S20}$$

$$= \frac{1}{2}(\bar{\mathbf{v}}_i^\top\mathbf{S}_i^\top\mathbf{K}\mathbf{S}_i\bar{\mathbf{v}}_i - \log\det(\mathbf{I}_{n\times n} - \tilde{\mathbf{C}}_i\mathbf{K}) + \mathrm{tr}(\mathbf{I}_{n\times n} - \tilde{\mathbf{C}}_i\mathbf{K}) - n) \tag{S21}$$

$$= \frac{1}{2}\left(\bar{\mathbf{v}}_i^\top\mathbf{S}_i^\top\mathbf{K}\mathbf{S}_i\bar{\mathbf{v}}_i - \log\det(\mathbf{I}_{n\times n} - \tilde{\mathbf{C}}_i\mathbf{K}) - \mathrm{tr}((\mathbf{S}_i^\top\tilde{\mathbf{K}}\mathbf{S}_i)^{-1}\mathbf{S}_i^\top\mathbf{K}\mathbf{S}_i)\right) \tag{S22}$$

Here, we apply the Weinstein–Aronszajn identity:

$$= \frac{1}{2}\left(\bar{\mathbf{v}}_i^\top\mathbf{S}_i^\top\mathbf{K}\mathbf{S}_i\bar{\mathbf{v}}_i - \log\det(\mathbf{I}_{i\times i} - (\mathbf{S}_i^\top\tilde{\mathbf{K}}\mathbf{S}_i)^{-1}\mathbf{S}_i^\top\mathbf{K}\mathbf{S}_i) - \mathrm{tr}((\mathbf{S}_i^\top\tilde{\mathbf{K}}\mathbf{S}_i)^{-1}\mathbf{S}_i^\top\mathbf{K}\mathbf{S}_i)\right) \tag{S23}$$

$$= \frac{1}{2}\left(\bar{\mathbf{v}}_i^\top\mathbf{S}_i^\top\mathbf{K}\mathbf{S}_i\bar{\mathbf{v}}_i - \log\det((\mathbf{S}_i^\top\tilde{\mathbf{K}}\mathbf{S}_i)^{-1}(\mathbf{S}_i^\top\tilde{\mathbf{K}}\mathbf{S}_i - \mathbf{S}_i^\top\mathbf{K}\mathbf{S}_i)) - \mathrm{tr}((\mathbf{S}_i^\top\tilde{\mathbf{K}}\mathbf{S}_i)^{-1}\mathbf{S}_i^\top\mathbf{K}\mathbf{S}_i)\right) \tag{S24}$$

$$= \frac{1}{2}\left(\bar{\mathbf{v}}_i^\top\mathbf{S}_i^\top\mathbf{K}\mathbf{S}_i\bar{\mathbf{v}}_i - \log\det((\mathbf{S}_i^\top\tilde{\mathbf{K}}\mathbf{S}_i)^{-1}\sigma_{\mathrm{noise}}^2\mathbf{S}_i^\top\mathbf{J}_{\mathbf{w}}\mathbf{S}_i) - \mathrm{tr}((\mathbf{S}_i^\top\tilde{\mathbf{K}}\mathbf{S}_i)^{-1}\mathbf{S}_i^\top\mathbf{K}\mathbf{S}_i)\right) \tag{S25}$$

$$= \frac{1}{2}\left(\bar{\mathbf{v}}_i^\top\mathbf{S}_i^\top\mathbf{K}\mathbf{S}_i\bar{\mathbf{v}}_i + \log\det(\mathbf{S}_i^\top\tilde{\mathbf{K}}\mathbf{S}_i) - \log\det(\sigma_{\mathrm{noise}}^2\mathbf{S}_i^\top\mathbf{J}_{\mathbf{w}}\mathbf{S}_i) - \mathrm{tr}((\mathbf{S}_i^\top\tilde{\mathbf{K}}\mathbf{S}_i)^{-1}\mathbf{S}_i^\top\mathbf{K}\mathbf{S}_i)\right) \tag{S26}$$

$$= \frac{1}{2}(\bar{\mathbf{v}}_i^\top\mathbf{S}_i^\top\mathbf{K}\mathbf{S}_i\bar{\mathbf{v}}_i + \log\det(\mathbf{S}_i^\top\tilde{\mathbf{K}}\mathbf{S}_i) - i\log(\sigma_{\mathrm{noise}}^2) - \log\det(\mathbf{J}_{\mathbf{w}}) - \log\det(\mathbf{S}_i^\top\mathbf{S}_i)$$
$$- \mathrm{tr}((\mathbf{S}_i^\top\tilde{\mathbf{K}}\mathbf{S}_i)^{-1}\mathbf{S}_i^\top\mathbf{K}\mathbf{S}_i)) \tag{S27}$$

## D   Robustness Property

The following lemma contributes to the proof of Proposition 4.1.

**Lemma D.1.** *For an arbitrary matrice $\hat{\mathbf{S}} \in \mathbb{R}^{m \times n}$ and positive semidefinite matrice $\hat{\mathbf{B}} \in \mathbb{R}^{n \times n}$, we have that*

$$(\hat{\mathbf{S}}\hat{\mathbf{B}}\hat{\mathbf{S}}^\top)^{-1} = \hat{\mathbf{S}}^{+\top}\hat{\mathbf{B}}^{-1/2}\hat{\mathbf{G}}\hat{\mathbf{B}}^{-1/2}\hat{\mathbf{S}}^+ \tag{S28}$$

*where we define $\hat{\mathbf{G}} = \mathbf{I} - \hat{\mathbf{B}}^{-1/2}(\mathbf{I} - \hat{\mathbf{S}}^+\hat{\mathbf{S}})(\hat{\mathbf{B}}^{-1/2}(\mathbf{I} - \hat{\mathbf{S}}^+\hat{\mathbf{S}}))^+$ and $^+$ denotes the Moore-Penrose inverse.*

*Proof:*

The whole proof is derived from an answer to a question posted on the , which we write here for conciseness.

Denote $\hat{\mathbf{O}} = \mathbf{I} - \hat{\mathbf{S}}^+\hat{\mathbf{S}}$ and $\mathbf{H}(\alpha) = (\hat{\mathbf{S}}(\alpha\mathbf{I} + \hat{\mathbf{B}}^{-1})^{-1}\hat{\mathbf{S}}^\top)^{-1}$. We also note that

$$(\hat{\mathbf{S}}\hat{\mathbf{B}}\hat{\mathbf{S}}^\top)^{-1} = \lim_{\alpha \to 0} \mathbf{H}(\alpha) \tag{S29}$$

By applying Woodbury matrix identity, we can rewrite $\mathbf{H}(\alpha)$ as follows:

$$\mathbf{H}(\alpha) = \left( \frac{1}{\alpha}\hat{\mathbf{S}}\hat{\mathbf{S}}^\top - \frac{1}{\alpha}\hat{\mathbf{S}}\hat{\mathbf{B}}^{-1/2}\left(\mathbf{I} + \frac{1}{\alpha}\hat{\mathbf{B}}^{-1}\right)^{-1}\frac{1}{\alpha}\hat{\mathbf{B}}^{-1/2}\hat{\mathbf{S}}^\top \right)^{-1} \tag{S30}$$

Using the fact that $\hat{\mathbf{S}}\hat{\mathbf{S}}^\top$ is invertible and applying the Woodbury matrix identity for the second time, we obtain

$$\mathbf{H}(\alpha) = \alpha(\hat{\mathbf{S}}\hat{\mathbf{S}}^\top)^{-1} - (\hat{\mathbf{S}}\hat{\mathbf{S}}^\top)^{-1}\hat{\mathbf{S}}\hat{\mathbf{B}}^{-1/2}$$
$$(-(\mathbf{I} + \frac{1}{\alpha}\hat{\mathbf{B}}^{-1}) + \frac{1}{\alpha}\hat{\mathbf{B}}^{-1/2}\hat{\mathbf{S}}^\top(\hat{\mathbf{S}}\hat{\mathbf{S}}^\top)^{-1}\hat{\mathbf{S}}\hat{\mathbf{B}}^{-1/2})^{-1}\hat{\mathbf{B}}^{-1/2}\hat{\mathbf{S}}^\top(\hat{\mathbf{S}}\hat{\mathbf{S}}^\top)^{-1} \tag{S31}$$
$$= \alpha(\hat{\mathbf{S}}\hat{\mathbf{S}}^\top)^{-1} + (\hat{\mathbf{S}}\hat{\mathbf{S}}^\top)^{-1}\hat{\mathbf{S}}\hat{\mathbf{B}}^{-1/2}(\mathbf{I} + \frac{1}{\alpha}\hat{\mathbf{B}}^{-1/2}(\mathbf{I} - \hat{\mathbf{S}}^\top(\hat{\mathbf{S}}\hat{\mathbf{S}}^\top)^{-1}\hat{\mathbf{S}})\hat{\mathbf{B}}^{-1/2})^{-1}$$
$$\hat{\mathbf{B}}^{-1/2}\hat{\mathbf{S}}^\top(\hat{\mathbf{S}}\hat{\mathbf{S}}^\top)^{-1} \tag{S32}$$

We note that

$$\hat{\mathbf{S}}^\top(\hat{\mathbf{S}}\hat{\mathbf{S}}^\top)^{-1} = \hat{\mathbf{S}}^+ \tag{S33}$$
$$\mathbf{I} - \hat{\mathbf{S}}^\top(\hat{\mathbf{S}}\hat{\mathbf{S}}^\top)^{-1}\hat{\mathbf{S}} = \hat{\mathbf{O}} \tag{S34}$$

Then, we rewrite $\mathbf{H}(\alpha)$ as follows:

$$\mathbf{H}(\alpha) = \alpha(\hat{\mathbf{S}}\hat{\mathbf{S}}^\top)^{-1} + \hat{\mathbf{S}}^{+\top}\hat{\mathbf{B}}^{-1/2}\left(\mathbf{I} + \frac{1}{\alpha}\hat{\mathbf{B}}^{-1/2}\hat{\mathbf{O}}\hat{\mathbf{O}}\hat{\mathbf{B}}^{-1/2}\right)^{-1}\hat{\mathbf{B}}^{-1/2}\hat{\mathbf{S}}^+ \tag{S35}$$

Applying the Woodbury matrix identity for the third time gives us

$$\mathbf{H}(\alpha) = \alpha(\hat{\mathbf{S}}\hat{\mathbf{S}}^\top)^{-1} + \hat{\mathbf{S}}^{+\top}\hat{\mathbf{B}}^{-1/2}(\mathbf{I} - \hat{\mathbf{B}}^{-1/2}\hat{\mathbf{O}}(\alpha\mathbf{I} + \hat{\mathbf{O}}\hat{\mathbf{B}}^{-1}\hat{\mathbf{O}})^{-1}\hat{\mathbf{O}}\hat{\mathbf{B}}^{-1/2})\hat{\mathbf{B}}^{-1/2}\hat{\mathbf{S}}^+ \tag{S36}$$

Since the Moore-Penrose inverse of a matrix $\mathbf{A}$ is a limit:

$$\mathbf{A}^+ = \lim_{\alpha \to 0}(\mathbf{A}^\top\mathbf{A} + \alpha\mathbf{I})^{-1}\mathbf{A}^\top = \lim_{\alpha \to 0}\mathbf{A}^\top(\mathbf{A}\mathbf{A}^\top + \alpha\mathbf{I})^{-1} \tag{S37}$$

We can take the limit of $\mathbf{H}(\alpha)$ as $\alpha \to 0$ and apply the limit relation above to obtain the following result:

$$(\hat{\mathbf{S}}\hat{\mathbf{B}}\hat{\mathbf{S}}^\top)^{-1} = \hat{\mathbf{S}}^{+\top}\hat{\mathbf{B}}^{-1/2}\underbrace{(\mathbf{I} - \hat{\mathbf{B}}^{-1/2}\hat{\mathbf{O}}(\hat{\mathbf{B}}^{-1/2}\hat{\mathbf{O}})^+)}_{\hat{\mathbf{G}}}\hat{\mathbf{B}}^{-1/2}\hat{\mathbf{S}}^+ \tag{S38}$$

**PIF of RCaGP** We now prove Proposition 4.1: Let $\mathbf{f} \sim \mathcal{GP}(m, k)$ denote the RCaGP prior, and let $i \in 0, \ldots, n$ represent the number of actions in RCaGP. Assume the observation noise is

$\varepsilon \sim \mathcal{N}(\mathbf{0}, \sigma_{\text{noise}}^2 \mathbf{I})$. Define constants $C_k' \in \mathbb{R}; k = 1, 2, 3$, which are independent of $y_m^c$. For any $i$ and assuming $\sup_{\mathbf{x}, y} w(\mathbf{x}, y) < \infty$, the PIF of RCaGP is given by:

$$\text{PIF}_{\text{RCaGP}}(y_m^c, \mathcal{D}) = C_2'(w(\mathbf{x}_m, y_m^c)^2 y_m^c)^2 + C_3'. \tag{S39}$$

Therefore, if $\sup_{\mathbf{x}, y} y\, w(\mathbf{x}, y)^2 < \infty$, RCaGP regression is robust since $\sup_{y_m^c} |\text{PIF}_{\text{RCaGP}}(y_m^c, \mathcal{D})| < \infty$.

*Proof:*

Without loss of generality, we aim to prove the bound for $m = n$. We can extend the proof for an arbitrary $m \in \{1, \ldots, n\}$. Let $p^w(\mathbf{f}|\mathcal{D}) = \mathcal{N}(\mathbf{f}; \hat{\boldsymbol{\mu}}_i, \hat{\mathbf{K}}_i)$ and $p^w(\mathbf{f}|\mathcal{D}_m^c) = \mathcal{N}(\mathbf{f}; \hat{\boldsymbol{\mu}}_i^c, \hat{\mathbf{K}}_i^c)$ be the uncontaminated and contaminated computation-aware RCGP, respectively. Here,

$$\hat{\boldsymbol{\mu}}_i = \mathbf{m} + \mathbf{K}\tilde{\mathbf{C}}_i\tilde{\mathbf{v}}_i \tag{S40}$$

$$\hat{\mathbf{K}}_i = \mathbf{K}\tilde{\mathbf{C}}_i\sigma_{\text{noise}}^2\mathbf{J}_{\mathbf{w}} \tag{S41}$$

$$\hat{\boldsymbol{\mu}}_i^c = \mathbf{m} + \mathbf{K}\tilde{\mathbf{C}}_i^c\tilde{\mathbf{v}}_i^c \tag{S42}$$

$$\hat{\mathbf{K}}_i^c = \mathbf{K}\tilde{\mathbf{C}}_i^c\sigma_{\text{noise}}^2\mathbf{J}_{\mathbf{w}^c} \tag{S43}$$

where $\mathbf{w}^c = [w(\mathbf{x}_1, y_1), \ldots, w(\mathbf{x}_n, y_n^c)]^\top$. The PIF has the following form

$$\text{PIF}_{\text{RCaGP}}(y_m^c, \mathcal{D}, i) = \frac{1}{2}\left( \underbrace{\text{Tr}((\hat{\mathbf{K}}_i^c)^{-1}\hat{\mathbf{K}}_i) - n}_{(1)} + \underbrace{(\hat{\boldsymbol{\mu}}_i^c - \hat{\boldsymbol{\mu}}_i)^\top(\hat{\mathbf{K}}_i^c)^{-1}(\hat{\boldsymbol{\mu}}_i^c - \hat{\boldsymbol{\mu}}_i)}_{(2)} + \underbrace{\ln\left(\frac{\det(\hat{\mathbf{K}}_i^c)}{\det(\hat{\mathbf{K}}_i)}\right)}_{(3)} \right) \tag{S44}$$

We first derive the bound for (1):

$$(1) = \text{Tr}((\hat{\mathbf{K}}_i^c)^{-1}\hat{\mathbf{K}}_i) - n \tag{S45}$$

$$= \text{Tr}\left((\mathbf{K}\tilde{\mathbf{C}}_i^c\sigma_{\text{noise}}^2\mathbf{J}_{\mathbf{w}^c})^{-1}\mathbf{K}\tilde{\mathbf{C}}_i\sigma_{\text{noise}}^2\mathbf{J}_{\mathbf{w}}\right) - n \tag{S46}$$

$$= \text{Tr}(\sigma_{\text{noise}}^{-2}\mathbf{J}_{\mathbf{w}^c}^{-1}(\tilde{\mathbf{C}}_i^c)^{-1}\tilde{\mathbf{C}}_i\sigma_{\text{noise}}^2\mathbf{J}_{\mathbf{w}}) - n \tag{S47}$$

$$\leq \text{Tr}(\sigma_{\text{noise}}^{-2}\mathbf{J}_{\mathbf{w}^c}^{-1}(\tilde{\mathbf{C}}_i^c)^{-1})\text{Tr}(\tilde{\mathbf{C}}_i\sigma_{\text{noise}}^2\mathbf{J}_{\mathbf{w}}) - n \tag{S48}$$

$$\leq \text{Tr}(\sigma_{\text{noise}}^{-2}\mathbf{J}_{\mathbf{w}^c}^{-1})\text{Tr}((\tilde{\mathbf{C}}_i^c)^{-1})\text{Tr}(\tilde{\mathbf{C}}_i\sigma_{\text{noise}}^2\mathbf{J}_{\mathbf{w}}) - n \tag{S49}$$

The first and second inequality come from the fact that $\text{Tr}(\mathbf{A}\mathbf{F}) \leq \text{Tr}(\mathbf{A})\text{Tr}(\mathbf{F})$ for two positive semidefinite matrices $\mathbf{A}$ and $\mathbf{F}$. Since $\text{Tr}(\tilde{\mathbf{C}}_i\sigma_{\text{noise}}^2\mathbf{J}_{\mathbf{w}})$ does not contain the contamination term, we can write $\bar{C}_1 = \text{Tr}(\tilde{\mathbf{C}}_i\sigma_{\text{noise}}^2\mathbf{J}_{\mathbf{w}})$. Let $\mathbf{B} = (\mathbf{S}_i^\top\tilde{\mathbf{K}}^c\mathbf{S}_i)^{-1}$ such that $\mathbf{C}_i^c = \mathbf{S}_i^\top\mathbf{B}\mathbf{S}_i^\top$. Observe that matrice $\mathbf{B}$ is positive semidefinite. Thus, we can apply Lemma D.1 to obtain the bound of $\text{Tr}((\tilde{\mathbf{C}}_i^c)^{-1})$:

$$\text{Tr}((\tilde{\mathbf{C}}_i^c)^{-1}) = \text{Tr}((\mathbf{S}_i^\top\mathbf{B}\mathbf{S}_i^\top)^{-1}) \tag{S50}$$

$$= \text{Tr}(\mathbf{S}_i^{+\top}\mathbf{B}^{-1/2}\mathbf{G}\mathbf{B}^{-1/2}\mathbf{S}_i^+) \tag{S51}$$

$$\leq \text{Tr}(\mathbf{S}_i^+\mathbf{S}_i^{+\top})\text{Tr}(\mathbf{B}^{-1/2}\mathbf{B}^{-1/2})\text{Tr}(\mathbf{G}) \tag{S52}$$

Furthermore, we derive the bound of $\text{Tr}(\mathbf{G})$ as follows:

$$\text{Tr}(\mathbf{G}) = \text{Tr}(\mathbf{I} - \mathbf{B}^{-1/2}(\mathbf{I} - \mathbf{S}_i^+\mathbf{S}_i)(\mathbf{B}^{-1/2}(\mathbf{I} - \mathbf{S}_i^+\mathbf{S}_i))^+) \tag{S53}$$

$$= n - \text{Tr}(\mathbf{B}^{-1/2}(\mathbf{I} - \mathbf{S}_i^+\mathbf{S}_i)(\mathbf{I} - \mathbf{S}_i^+\mathbf{S}_i)^+\mathbf{B}^{-1/2+}) \tag{S54}$$

$$\leq n - \text{Tr}(\mathbf{B}^{-1/2+}\mathbf{B}^{-1/2})\text{Tr}((\mathbf{I} - \mathbf{S}_i^+\mathbf{S}_i)(\mathbf{I} - \mathbf{S}_i^+\mathbf{S}_i)^+) \tag{S55}$$

The inequality S52 stems from the trace circular property and the properties of the product of two positive semidefinite matrices. Notably, we observe that $\text{Tr}(\mathbf{G}) \leq n$ since $\mathbf{B}^{-1/2+}\mathbf{B}^{-1/2}$ and $(\mathbf{I} - \mathbf{S}_i^+\mathbf{S}_i)(\mathbf{I} - \mathbf{S}_i^+\mathbf{S}_i)^+$ in S55 are positive semidefinite matrices. By definition, the trace of a

positive semidefinite matrix is non-negative. We then apply the trace circular property for the second time to obtain

$$\mathrm{Tr}((\tilde{\mathbf{C}}_i^c)^{-1}) \leq n\mathrm{Tr}(\mathbf{S}_i^+\mathbf{S}_i^{+\top})\mathrm{Tr}(\mathbf{B}^{-1}) \tag{S56}$$

$$\leq n\mathrm{Tr}(\mathbf{S}_i^+\mathbf{S}_i^{+\top})\mathrm{Tr}(\mathbf{S}_i\mathbf{S}_i^\top)\mathrm{Tr}(\tilde{\mathbf{K}}^c) \tag{S57}$$

$$= \bar{C}_2\mathrm{Tr}(\mathbf{K} + \sigma_{\mathrm{noise}}^2\mathbf{J}_{\mathbf{w}^c}) \tag{S58}$$

where we define $\bar{C}_2 = n\mathrm{Tr}(\mathbf{S}_i^+\mathbf{S}_i^{+\top})\mathrm{Tr}(\mathbf{S}_i\mathbf{S}_i^\top)$. We then plug S58 into S49 to obtain

$$(1) \leq \mathrm{Tr}(\sigma_{\mathrm{noise}}^{-2}\mathbf{J}_{\mathbf{w}^c}^{-1})\mathrm{Tr}(\mathbf{K} + \sigma_{\mathrm{noise}}^2\mathbf{J}_{\mathbf{w}^c})\bar{C}_1\bar{C}_2 - n \tag{S59}$$

$$= \left(\sum_{j=1}^n \left(4\sigma_{\mathrm{noise}}^{-2}w^2(\mathbf{x}_j, y_j)\right)\sum_{k=1}^n \left(\mathbf{K}_{kk} + \sigma_{\mathrm{noise}}^4/2\ w^{-2}(\mathbf{x}_k, y_k)\right)\right)\bar{C}_1\bar{C}_2 - n \tag{S60}$$

$$\leq \left(n\sup_{\mathbf{x},y} 4\sigma_{\mathrm{noise}}^{-2}\ w^2(\mathbf{x}, y)\left(n\sup_{\hat{\mathbf{x}},\hat{y}} \sigma_{\mathrm{noise}}^4/2\ w^{-2}(\hat{\mathbf{x}}, \hat{y}) + \sum_{k=1}^n \mathbf{K}_{kk}\right)\right)\bar{C}_1\bar{C}_2 - n = \bar{C}_3 \tag{S61}$$

Next, we derive the bound for the term (2). Following [2], we have that

$$(2) \leq \lambda_{\max}((\hat{\mathbf{K}}_i^c)^{-1})\|\hat{\boldsymbol{\mu}}_i^c - \hat{\boldsymbol{\mu}}_i\|_1^2, \tag{S62}$$

where $\lambda_{\max}((\hat{\mathbf{K}}_i^c)^{-1})$ is the maximum eigenvalue of $(\hat{\mathbf{K}}_i^c)^{-1}$. We expand $\lambda_{\max}((\hat{\mathbf{K}}_i^c)^{-1})$ to derive the following bound:

$$\lambda_{\max}((\hat{\mathbf{K}}_i^c)^{-1}) = \lambda_{\max}(\sigma^{-2}\mathbf{J}_{\mathbf{w}^c}^{-1}(\tilde{\mathbf{C}}_i^c)^{-1}\mathbf{K}^{-1}) \tag{S63}$$

$$\leq \lambda_{\max}(\sigma_{\mathrm{noise}}^{-2}\mathbf{J}_{\mathbf{w}^c}^{-1})\lambda_{\max}((\tilde{\mathbf{C}}_i^c)^{-1})\lambda_{\max}(\mathbf{K}^{-1}) \tag{S64}$$

$$= \lambda_{\max}(\sigma_{\mathrm{noise}}^{-2}\mathbf{J}_{\mathbf{w}^c}^{-1})\lambda_{\min}(\tilde{\mathbf{C}}_i^c)\lambda_{\max}(\mathbf{K}^{-1}) \tag{S65}$$

$$\leq \lambda_{\max}(\sigma_{\mathrm{noise}}^{-2}\mathbf{J}_{\mathbf{w}^c}^{-1})\lambda_{\min}((\tilde{\mathbf{K}}^c)^{-1})\lambda_{\max}(\mathbf{K}^{-1}) \tag{S66}$$

$$\leq \lambda_{\max}(\sigma_{\mathrm{noise}}^{-2}\mathbf{J}_{\mathbf{w}^c}^{-1})(\lambda_{\max}(\mathbf{K}) + \lambda_{\max}(\sigma_{\mathrm{noise}}^2\mathbf{J}_{\mathbf{w}^c}))\lambda_{\max}(\mathbf{K}^{-1}) \tag{S67}$$

The first inequality follows from the property of the maximum eigenvalue of the product of two positive semidefinite matrices. Subsequently, the second equality is derived from the fact that a matrice's maximum eigenvalue equals its inverse's minimum eigenvalue. By the definition $\tilde{\mathbf{C}}_i^c = (\tilde{\mathbf{K}}^c)^{-1} - \tilde{\boldsymbol{\Sigma}}_i^c$ and the fact that $\tilde{\mathbf{C}}_i^c$, $(\tilde{\mathbf{K}}^c)^{-1}$ and $\boldsymbol{\Sigma}_i$ are positive semidefinite matrices, the minimum eigenvalue of $\tilde{\mathbf{K}}^c$ is less or equal than of $\tilde{\mathbf{C}}_i^c$, resulting in second inequality. The last inequality follows from the equivalence of the maximum eigenvalue and the addition property of the maximum eigenvalue of two positive semidefinite matrices.

Since $\mathbf{J}_{\mathbf{w}^c}^{-1} = \mathrm{diag}(\frac{\sigma_{\mathrm{noise}}^2}{2}(\mathbf{w}^c)^2)$, and $\sup_{\mathbf{x},y} w(\mathbf{x}, y) < \infty$, it holds that $\lambda_{\max}(\sigma_{\mathrm{noise}}^{-2}\mathbf{J}_{\mathbf{w}^c}^{-1}) = \bar{C}_4 < +\infty$ and $\lambda_{\max}(\sigma_{\mathrm{noise}}^2\mathbf{J}_{\mathbf{w}^c}) = \bar{C}_5 < +\infty$, such that

$$\lambda_{\max}((\hat{\mathbf{K}}_i^c)^{-1}) \leq \bar{C}_4(\lambda_{\max}(\mathbf{K}) + \bar{C}_5)\lambda_{\max}(\mathbf{K}^{-1}) = \bar{C}_6 \tag{S68}$$

We substitute $\bar{C}_6$ into (2) to obtain

$$(2) \leq \bar{C}_6\|\hat{\boldsymbol{\mu}}_i^c - \hat{\boldsymbol{\mu}}_i\|_1^2 \tag{S69}$$

$$= \bar{C}_6\|(\mathbf{m} + \mathbf{K}\tilde{\mathbf{v}}_i^c) - (\mathbf{m} + \mathbf{K}\tilde{\mathbf{v}}_i)\|_1^2 \tag{S70}$$

$$= \bar{C}_6\|\mathbf{K}(\tilde{\mathbf{C}}_i^c(\mathbf{y}^c - \mathbf{m}_{\mathbf{w}^c}) - \tilde{\mathbf{C}}_i(\mathbf{y} - \mathbf{m}_{\mathbf{w}}))\|_1^2 \tag{S71}$$

$$\leq \bar{C}_6\|\mathbf{K}\|_F\|\tilde{\mathbf{C}}_i^c(\mathbf{y}^c - \mathbf{m}_{\mathbf{w}^c}) - \tilde{\mathbf{C}}_i(\mathbf{y} - \mathbf{m}_{\mathbf{w}})\|_1^2 \tag{S72}$$

$$\leq q\,\bar{C}_6\|\mathbf{K}\|_F\|(\tilde{\mathbf{K}}^c)^{-1}(\mathbf{y}^c - \mathbf{m}_{\mathbf{w}^c}) - (\tilde{\mathbf{K}})^{-1}(\mathbf{y} - \mathbf{m}_{\mathbf{w}})\|_1^2 \tag{S73}$$

$$= q\,\bar{C}_6\|\mathbf{K}\|_F\|(\mathbf{K} + \sigma_{\mathrm{noise}}^2\mathbf{J}_{\mathbf{w}^c})^{-1}(\mathbf{y}^c - \mathbf{m}_{\mathbf{w}^c}) - (\mathbf{K} + \sigma_{\mathrm{noise}}^2\mathbf{J}_{\mathbf{w}})^{-1}(\mathbf{y} - \mathbf{m}_{\mathbf{w}})\|_1^2 \tag{S74}$$

for a constant $q < +\infty$. The second equality comes from the definition of $\tilde{\mathbf{v}}_i$, and the fourth row follows the Cauchy-Schwarz inequality. Finally, the last inequality holds since $\tilde{\mathbf{K}}_i = (\tilde{\mathbf{C}}_i^{-1} - \tilde{\boldsymbol{\Sigma}}_i)$.

Applying results from [2], we obtain

$$(2) \leq q\,\bar{C}_6\,\|\mathbf{K}\|_F\,\|(\mathbf{K} + \sigma_{\text{noise}}^2\mathbf{J}_{\mathbf{w}^c})^{-1}(\mathbf{y} - \mathbf{m}_{\mathbf{w}^c}) - (\mathbf{K} + \sigma_{\text{noise}}^2\mathbf{J}_{\mathbf{w}})(\mathbf{y} - \mathbf{m}_{\mathbf{w}})\|_1^2 \tag{S75}$$

$$\leq q\,\bar{C}_6\,\|\mathbf{K}\|_F\,2((\bar{C}_7 + \bar{C}_8)^2 + (\bar{C}_9 + \bar{C}_{10})^2(w(x_n, y_n^c)^2 y_n^c)^2) \tag{S76}$$

$$\leq \bar{C}_{11} + \bar{C}_{12}(w(x_n, y_n^c)^2 y_n^c)^2 \tag{S77}$$

where $\bar{C}_{11} = q\bar{C}_6\|\mathbf{K}\|_F 2(\bar{C}_7 + \bar{C}_8)^2$ and $\bar{C}_{12} = q\bar{C}_6\|\mathbf{K}\|_F 2(\bar{C}_9 + \bar{C}_{10})^2$. The terms $\bar{C}_7, \bar{C}_8, \bar{C}_9, \bar{C}_{10}$ correspond to $\tilde{C}_6, \tilde{C}_8, \tilde{C}_7, \tilde{C}_9$ in [2].

The term (3) can be written as follows:

$$(3) = \ln\left(\frac{\det(\hat{\mathbf{K}}_i^c)}{\det(\hat{\mathbf{K}}_i)}\right) \tag{S78}$$

$$= \ln\left(\frac{\det(\tilde{\mathbf{C}}_i^c \sigma_{\text{noise}}^2 \mathbf{J}_{\mathbf{w}^c})}{\det(\tilde{\mathbf{C}}_i \sigma_{\text{noise}}^2 \mathbf{J}_{\mathbf{w}})}\right) \tag{S79}$$

$$= \ln(\det(\sigma_{\text{noise}}^{-2}\mathbf{J}_{\mathbf{w}}^{-1}\tilde{\mathbf{C}}_i^{-1})\det(\tilde{\mathbf{C}}_i^c)\det(\sigma_{\text{noise}}^2\mathbf{J}_{\mathbf{w}^c})) \tag{S80}$$

Observe that we can write $\bar{C}_{13} = \ln(\det(\sigma_{\text{noise}}^{-2}\mathbf{J}_{\mathbf{w}}^{-1}\tilde{\mathbf{C}}_i^{-1}))$ since it does not contain the contimation term. Furthermore, we obtain

$$(3) = \ln(\bar{C}_{13}\det(\tilde{\mathbf{C}}_i^c)\det(\sigma_{\text{noise}}^2\mathbf{J}_{\mathbf{w}^c})) \tag{S81}$$

$$\leq \ln(\bar{C}_{13}\det((\tilde{\mathbf{K}}^c)^{-1})\det(\sigma_{\text{noise}}^2\mathbf{J}_{\mathbf{w}^c})) \tag{S82}$$

$$= \ln\left(\bar{C}_{13}\frac{\det(\sigma_{\text{noise}}^2\mathbf{J}_{\mathbf{w}^c})}{\det(\mathbf{K} + \sigma_{\text{noise}}^2\mathbf{J}_{\mathbf{w}^c})}\right) \tag{S83}$$

$$\leq \ln\left(\bar{C}_{13}\frac{\det(\sigma_{\text{noise}}^2\mathbf{J}_{\mathbf{w}^c})}{\det(\mathbf{K}) + \det(\sigma_{\text{noise}}^2\mathbf{J}_{\mathbf{w}^c})}\right) \tag{S84}$$

The first inequality follows from the determinant property of positive semidefinite matrices. The last inequality leverages the fact that $\det(\mathbf{A} + \mathbf{F}) \geq \det(\mathbf{A}) + \det(\mathbf{F})$ for $\mathbf{A}$ and $\mathbf{F}$ are positive semidefinite matrices. Since $\det(\mathbf{K}), \det(\sigma_{\text{noise}}^2\mathbf{J}_{\mathbf{w}^c}) \geq 0$, we find that

$$\ln\left(\frac{\det(\sigma_{\text{noise}}^2\mathbf{J}_{\mathbf{w}^c})}{\det(\mathbf{K}) + \det(\sigma_{\text{noise}}^2\mathbf{J}_{\mathbf{w}^c})}\right) \leq 1, \tag{S85}$$

leading to the following inequality:

$$(3) \leq \ln(\bar{C}_{13}) = \bar{C}_{14}. \tag{S86}$$

Finally, putting the three terms together, we obtain the following bound:

$$\text{PIF}_{\text{RCaGP}}(y_m^c, \mathcal{D}, i) \leq \bar{C}_3 + \bar{C}_{11} + \bar{C}_{12}(w(x_n, y_n^c)^2 y_n^c)^2 + \bar{C}_{14} \tag{S87}$$

$$= C_1'(w(x_n, y_n^c)^2 y_n^c)^2 + C_2' \tag{S88}$$

where $C_1' = \bar{C}_{12}$ and $C_2' = \bar{C}_3 + \bar{C}_{11} + \bar{C}_{14}$.

# E  RCaGP Capture the Worst-case Errors

**Proposition 4.2.** *Let $\hat{k}_i(\cdot, \cdot) = k_*(\cdot, \cdot) + \hat{\sigma}_i(\cdot, \cdot)$ be the combined uncertainty of RCaGP with zero-mean prior $m$. Then, for any new $\mathbf{x} \in \mathcal{X}$ we have that*

$$\sup_{\|h\|_{\mathcal{H}_{k^w}} \leq 1} (h(\mathbf{x}) - \hat{\mu}_i^g(\mathbf{x}))^2 = \hat{k}_i(\mathbf{x}, \mathbf{x}) + \sigma_{\text{noise}}^2 \tag{20}$$

$$\sup_{\|h\|_{\mathcal{H}_{k^w}} \leq 1} (\mu_*^g(\mathbf{x}) - \hat{\mu}_i^g(\mathbf{x}))^2 = \hat{\sigma}_i(\mathbf{x}, \mathbf{x}) \tag{21}$$

*where $\mu_*^g(\cdot) = k(\cdot, \mathbf{X})\tilde{\mathbf{K}}^{-1}h(\mathbf{X})$ is the RCGP's posterior and $\hat{\mu}_i^g(\cdot) = k(\cdot, \mathbf{X})\tilde{\mathbf{C}}_i(g(\mathbf{X}) - m_w(\mathbf{X}))$ RCaGP's posterior mean for a function $h \in \mathcal{H}_{k^w} = g(\mathbf{x}) - m_w(\mathbf{x})$ with a latent function $g : \mathcal{X} \to \mathbb{R}$ and the shrinkage function $m_w : \mathcal{X} \times \mathcal{Y} \to \mathbb{R}$.*

*Proof:*

Let $c_j = [\tilde{\mathbf{C}}_i k^w(\mathbf{X}, \mathbf{x})]_j$ for $j = 1, \ldots, n$, where we define $k^w(.,.) = k(.,.) + \frac{\sigma_{\text{noise}}^2}{2}\delta_w(.,.)$:

$$\delta_w(\mathbf{x}, \mathbf{x}') = \begin{cases} w^{-2}(\mathbf{x}, y) & \mathbf{x} = \mathbf{x}' \text{ and } \mathbf{x} \in \mathcal{D} \\ 2 & \mathbf{x} = \mathbf{x}' \text{ and } \mathbf{x} \notin \mathcal{D} \\ 0 & \mathbf{x} \neq \mathbf{x}' \end{cases}, \tag{S89}$$

where $\mathcal{D}$ denotes the observation set. Then, applying [21, Lemma 3.9] provides

$$\left(\sup_{\|h\|_{\mathcal{H}_{k^w}} \leq 1} h(\mathbf{x}) - \hat{\mu}_i^g(\mathbf{x})\right)^2 = \left(\sup_{\|h\|_{\mathcal{H}_{k^w}} \leq 1} h(\mathbf{x}) - \sum_{j=1}^n c_j h(\mathbf{x}_j)\right)^2 \tag{S90}$$

$$= \|k^w(.,\mathbf{x}) - k(\mathbf{x}, \mathbf{X})\tilde{\mathbf{C}}_i k^w(\mathbf{X},.)\|_{\mathcal{H}_{k^w}}^2 \tag{S91}$$

$$= \langle k^w(.,\mathbf{x}), k^w(.,\mathbf{x})\rangle_{\mathcal{H}_k^w} - 2\langle k^w(.,\mathbf{x}), k(\mathbf{x}, \mathbf{X})\tilde{\mathbf{C}}_i k^w(\mathbf{X},.)\rangle_{\mathcal{H}_k^w}$$
$$+ \langle k(\mathbf{x}, \mathbf{X})\tilde{\mathbf{C}}_i k^w(\mathbf{X},.), k(\mathbf{x}, \mathbf{X})\tilde{\mathbf{C}}_i k^w(\mathbf{X},.)\rangle_{\mathcal{H}_k^w} \tag{S92}$$

By reproducing property, we have

$$\left(\sup_{\|h\|_{\mathcal{H}_{k^w}} \leq 1} h(\mathbf{x}) - \hat{\mu}_i^g(\mathbf{x})\right)^2 = k^w(\mathbf{x}, \mathbf{x}) - 2k^w(\mathbf{x}, \mathbf{X})\tilde{\mathbf{C}}_i k^w(\mathbf{X}, \mathbf{x}) + k^w(\mathbf{x}, \mathbf{X})\tilde{\mathbf{C}}_i k^w(\mathbf{X}, \mathbf{X})\tilde{\mathbf{C}}_i k^w(\mathbf{X}, \mathbf{x}) \tag{S93}$$

since $\mathbf{x} \neq \mathbf{x}_j$, it holds that $k^w(\mathbf{x}, \mathbf{X}) = k(\mathbf{x}, \mathbf{X})$. By definition, we have $k^w(\mathbf{X}, \mathbf{X}) = \tilde{\mathbf{K}} = \mathbf{K} + \frac{\sigma_{\text{noise}}^2}{2}\mathbf{J_w}$, and following in [46, Eq. (S42)], it holds that $\tilde{\mathbf{C}}_i\tilde{\mathbf{K}}\tilde{\mathbf{C}}_i = \tilde{\mathbf{C}}_i$. Therefore, we obtain

$$\left(\sup_{\|h\|_{\mathcal{H}_{k^w}} \leq 1} h(\mathbf{x}) - \hat{\mu}_i^g(\mathbf{x})\right)^2 = k(\mathbf{x}, \mathbf{x}) + \sigma_{\text{noise}}^2 - 2k(\mathbf{x}, \mathbf{X})\tilde{\mathbf{C}}_i k(\mathbf{X}, \mathbf{x}) + k(\mathbf{x}, \mathbf{X})\tilde{\mathbf{C}}_i\tilde{\mathbf{K}}\tilde{\mathbf{C}}_i k(\mathbf{X}, \mathbf{x}) \tag{S94}$$

$$= k(\mathbf{x}, \mathbf{x}) + \sigma_{\text{noise}}^2 - k(\mathbf{x}, \mathbf{X})\tilde{\mathbf{C}}_i k(\mathbf{X}, \mathbf{x}) \tag{S95}$$

$$= \hat{k}_i(\mathbf{x}, \mathbf{x}) + \sigma_{\text{noise}}^2 \tag{S96}$$

For the last result, we analogously choose $c_j = [(\tilde{\mathbf{K}}^{-1} - \tilde{\mathbf{C}}_i)k^w(\mathbf{X}, \mathbf{x})]_j$. Then, we obtain

$$\left(\sup_{\|h\|_{\mathcal{H}_{k^w}} \leq 1} \hat{\mu}^g(\mathbf{x}) - \hat{\mu}_i^g(\mathbf{x})\right)^2 = \left(\sup_{\|h\|_{\mathcal{H}_{k^w}} \leq 1} \sum_{j=0}^n c_j h(\mathbf{x}_j)\right)^2 \tag{S97}$$

$$= \|k(\mathbf{x}, \mathbf{X})(\tilde{\mathbf{K}}^{-1} - \tilde{\mathbf{C}}_i)k^w(\mathbf{X},.)\|_{\mathcal{H}_{k^w}}^2 \tag{S98}$$

$$= k^w(\mathbf{x}, \mathbf{X})\tilde{\mathbf{K}}^{-1}\tilde{\mathbf{K}}\tilde{\mathbf{K}}^{-1}k^w(\mathbf{X}, \mathbf{x}) - 2k^w(\mathbf{x}, \mathbf{X})\tilde{\mathbf{K}}^{-1}\tilde{\mathbf{K}}\tilde{\mathbf{C}}_i k^w(\mathbf{X}, \mathbf{x}) +$$
$$k^w(\mathbf{x}, \mathbf{X})\tilde{\mathbf{C}}_i\tilde{\mathbf{K}}\tilde{\mathbf{C}}_i k^w(\mathbf{X}, \mathbf{x}) \tag{S99}$$

$$= k(\mathbf{x}, \mathbf{X})(\tilde{\mathbf{K}}^{-1} - \tilde{\mathbf{C}}_i)k(\mathbf{X}, \mathbf{x}) \tag{S100}$$

$$= \tilde{\sigma}_i^2(\mathbf{x}, \mathbf{x}) \tag{S101}$$

The consequence of Proposition 4.2 is then formalized through the following corollary:

**Corollary E.1.** *Assume the conditions of Proposition 4.2 hold. Let $\mu_*$ be the corresponding mathematical RCGP posterior mean and $\hat{\mu}_i$ the RCaGP posterior mean. Then, it holds that*

$$\frac{|h(\mathbf{x}) - \hat{\mu}_i(\mathbf{x})|}{\|h\|_{\mathcal{H}_{k^w}}} \leq \sqrt{\hat{k}_i(\mathbf{x}, \mathbf{x}) + \sigma_{\text{noise}}^2} \tag{S102}$$

$$\frac{\hat{\mu}(\mathbf{x}) - \hat{\mu}_i(\mathbf{x})}{\|h\|_{\mathcal{H}_{k^w}}} \leq \sqrt{\tilde{\sigma}_i^2(\mathbf{x}, \mathbf{x})} \tag{S103}$$

The proof follows immediately from Proposition 4.2 and notice that $h/\|h\|_{\mathcal{H}_{k^w}}$ has unit norm. Corollary E.1 implies that the computational uncertainty provides the pointwise bound relative to RCGP's mathematical posterior, and the combined uncertainty is the pointwise bound relative to the shifted latent function.

# F Additional Theoretical Results

## F.1 CaGP Lacks Robustness

**PIF for the CaGP.** We aim to prove the following statement: CaGP regression has the PIF for some constant $C_3' \in \mathbb{R}$.

$$\text{PIF}_{\text{CaGP}}(y_m^c, \mathcal{D}, i) = C_3'(y_m - y_m^c)^2 \tag{S104}$$

and is not robust: $\text{PIF}_{\text{CaGP}}(y_m^c, \mathcal{D}, i) \to \infty$ as $|y_m^c| \to \infty$.

*Proof:*

Let $p(\mathbf{f}|\mathcal{D}) = \mathcal{N}(\mathbf{f}; \boldsymbol{\mu}_i, \mathbf{K}_i)$ and $p(\mathbf{f}|\mathcal{D}_m^c) = \mathcal{N}(\mathbf{f}; \boldsymbol{\mu}_i^c, \mathbf{K}_i^c)$ be the uncontaminated and contaminated computation-aware GP, respectively. Here,

$$\boldsymbol{\mu}_i = \mathbf{m} + \mathbf{K}\mathbf{v}_i \tag{S105}$$

$$\mathbf{K}_i = \mathbf{K}\mathbf{C}_i \sigma_{\text{noise}}^2 \mathbf{I}_n \tag{S106}$$

$$\boldsymbol{\mu}_i^c = \mathbf{m} + \mathbf{K}\mathbf{v}_i^c \tag{S107}$$

$$\mathbf{K}_i^c = \mathbf{K}\mathbf{C}_i \sigma_{\text{noise}}^2 \mathbf{I}_n \tag{S108}$$

Note that both $\mathbf{K}_i$ and $\mathbf{K}_i^c$ share the same matrice $\mathbf{C}_i$. Then, the PIF has the following form:

$$\text{PIF}_{\text{CaGP}}(y_m^c, \mathcal{D}, i) = \frac{1}{2}\left(\text{Tr}(\mathbf{K}_i^c \mathbf{K}_i) - n + (\boldsymbol{\mu}_i^c - \boldsymbol{\mu}_i)^\top (\mathbf{K}_i^c)^{-1}(\boldsymbol{\mu}_i^c - \boldsymbol{\mu}_i) + \ln\left(\frac{\det(\mathbf{K}_i^c)}{\det(\mathbf{K}_i)}\right)\right) \tag{S109}$$

Based on [2], the PIF leads to the following form:

$$\text{PIF}_{\text{CaGP}}(y_m^c, \mathcal{D}, i) = \frac{1}{2}\left((\boldsymbol{\mu}_i^c - \boldsymbol{\mu}_i)^\top (\mathbf{K}_i^c)^{-1}(\boldsymbol{\mu}_i^c - \boldsymbol{\mu}_i)\right) \tag{S110}$$

Notice that the term $\boldsymbol{\mu}_i^c - \boldsymbol{\mu}_i$ can be written as

$$\boldsymbol{\mu}_i^c - \boldsymbol{\mu}_i = (\mathbf{m} + \mathbf{K}\mathbf{v}_i^c) - (\mathbf{m} + \mathbf{K}\mathbf{v}_i) \tag{S111}$$

$$= \mathbf{K}(\mathbf{v}_i^c - \mathbf{v}_i) \tag{S112}$$

$$= \mathbf{K}(\mathbf{C}_i(\mathbf{y}^c - \mathbf{m}) - \mathbf{C}_i(\mathbf{y} - \mathbf{m})) \tag{S113}$$

$$= \mathbf{K}(\mathbf{C}_i(\mathbf{y}^c - \mathbf{y})) \tag{S114}$$

Substituting the RHS of Eq. (S114) to $\boldsymbol{\mu}_i^c - \boldsymbol{\mu}_i$ in Eq. (S110), we obtain

$$\text{PIF}_{\text{CaGP}}(y_m^c, \mathcal{D}, i) = \frac{1}{2}(\mathbf{C}_i(\mathbf{y}^c - \mathbf{y}))^\top \mathbf{K}\left(\mathbf{K}\mathbf{C}_i \sigma_{\text{noise}}^2 \mathbf{I}\right)^{-1} \mathbf{K}(\mathbf{C}_i(\mathbf{y}^c - \mathbf{y})) \tag{S115}$$

$$= \frac{1}{2}\sigma_{\text{noise}}^{-2}(\mathbf{y}^c - \mathbf{y})^\top \mathbf{C}_i^\top \mathbf{K}(\mathbf{y}^c - \mathbf{y}) \tag{S116}$$

Note that $\mathbf{y}$ and $\mathbf{y}^c$ have only one exception for the $m-$th element. Thus, we have

$$\text{PIF}_{\text{CaGP}}(y_m^c, \mathcal{D}, i) = \frac{1}{2}[\mathbf{C}_i^\top \mathbf{K}\sigma^{-2}\mathbf{I}]_{mm}(y_m^c - y_m)^2 \tag{S117}$$

## F.2 Mean convergence of RCaGP

### F.2.1 Empirical-risk minimization problem of RCGP.

We first show the corresponding empirical-risk minimization problem of RCGP whenever $\mathbf{m} = \mathbf{0}$. Following [2, proof of Proposition 3.1], we can rewrite $L_n^w$ and formulate the RCGP objective as follows:

$$\hat{\mathbf{f}} = \text{argmin}_{\mathbf{f} \in \mathcal{H}_k} \frac{1}{2n}\left(\underbrace{\mathbf{f}^\top \lambda^{-1}\mathbf{J}_{\mathbf{w}}^{-1}\mathbf{f} - 2\mathbf{f}^\top \lambda^{-1}\mathbf{J}_{\mathbf{w}}^{-1}(\mathbf{y} - \mathbf{m}_{\mathbf{w}})}_{L_n^w}\right) + \frac{1}{2}\|\mathbf{f}\|_{\mathcal{H}_k}^2 \tag{S118}$$

for $\lambda > 0$. The regularization constant $\lambda$ controls the smoothness of the estimator to avoid overfitting, where the larger the $\lambda$ is, the smoother the resulting estimator $\hat{\mathbf{f}}$ becomes. Next, we show the unique solution to S118 through the following lemma:

**Lemma F.1.** *If $\lambda > 0$ and the kernel $k$ is invertible, the solution to S118 is unique, and is given by*

$$\hat{f}(\mathbf{x}) = \mathbf{k}_{\mathbf{x}}(\mathbf{K} + \lambda \mathbf{J}_{\mathbf{w}})^{-1}(\mathbf{y} - \mathbf{m}_{\mathbf{w}}) = \sum_{j=1}^{n} \hat{\alpha}_j k(\mathbf{x}, \mathbf{x}_j), \mathbf{x} \in \mathcal{X} \qquad (S119)$$

*where*

$$(\hat{\alpha}_i, \ldots, \hat{\alpha}_n) = (\mathbf{K} + \lambda \mathbf{J}_{\mathbf{w}})^{-1}(\mathbf{y} - \mathbf{m}_{\mathbf{w}}) \in \mathbb{R}^n \qquad (S120)$$

*Proof:*

The optimization problem in S118 allows us to apply the representer theorem [37]. It implies that the solution of S118 can be written as a weighted sum, i.e.,

$$\hat{\mathbf{f}} = \sum_{j=1}^{n} \hat{\alpha}_j k(., \mathbf{x}_j) \qquad (S121)$$

for $\hat{\alpha}_1, \ldots, \hat{\alpha}_n \in \mathbb{R}$. Let $\hat{\boldsymbol{\alpha}} = [\hat{\alpha}_1, \ldots, \hat{\alpha}_n]^\top \in \mathbb{R}^n$. Substituting S121 into S118 provides

$$\operatorname{argmin}_{\hat{\boldsymbol{\alpha}} \in \mathbb{R}^n} \frac{1}{2n}(\lambda^{-1}\hat{\boldsymbol{\alpha}}^\top \mathbf{K} \mathbf{J}_{\mathbf{w}}^{-1}\mathbf{K}\hat{\boldsymbol{\alpha}} - 2\lambda^{-1}\hat{\boldsymbol{\alpha}}^\top \mathbf{K} \mathbf{J}_{\mathbf{w}}^{-1}(\mathbf{y} - \mathbf{m}_{\mathbf{w}})) + \frac{1}{2}\hat{\boldsymbol{\alpha}}^\top \mathbf{K}\hat{\boldsymbol{\alpha}} \qquad (S122)$$

Taking the differentiation of the objective w.r.t. $\boldsymbol{\alpha}$, setting it equal to zero, and arranging the result yields the following equation:

$$\mathbf{K}(\mathbf{K} + n\,\lambda\,\mathbf{J}_{\mathbf{w}})\hat{\boldsymbol{\alpha}} = \mathbf{K}(\mathbf{y} - \mathbf{m}_{\mathbf{w}}) \qquad (S123)$$

Since the objective in S122 is a convex function of $\boldsymbol{\alpha}$, we find that $\hat{\boldsymbol{\alpha}} = (\mathbf{K} + \lambda \mathbf{J}_{\mathbf{w}})^{-1}(\mathbf{y} - \mathbf{m}_{\mathbf{w}})$ provides the minimum of the objective (S118 and S122). Furthermore, we can verify that $L_n^w$ is a convex function w.r.t. $\mathbf{f}$. Therefore, we conclude that $\hat{\boldsymbol{\alpha}} = (\mathbf{K} + \lambda \mathbf{J}_{\mathbf{w}})^{-1}(\mathbf{y} - \mathbf{m}_{\mathbf{w}})$ provides the unique solution to S118. As a remark, Proposition F.1 closely connects with [21, Theorem 3.4].

### F.2.2 Proof of Proposition F.3

**Relative bound errors.** We first provide the equivalence of [46, Proposition 2]:

**Proposition F.2.** *For any choice of actions a relative bound error $\hat{\rho}(i)$ s.t. $\|\hat{\mathbf{v}} - \tilde{\mathbf{v}}_i\|_{\tilde{\mathbf{K}}} \leq \hat{\rho}(i)\|\hat{\mathbf{v}}\|_{\tilde{\mathbf{K}}}$ is given by*

$$\hat{\rho}(i) = (\bar{\mathbf{v}}^\top(\mathbf{I} - \tilde{\mathbf{C}}_i\tilde{\mathbf{K}})\bar{\mathbf{v}})^{1/2} \leq \lambda_{\max}(\mathbf{I} - \tilde{\mathbf{C}}_i\tilde{\mathbf{K}}) \leq 1 \qquad (S124)$$

*where $\bar{\mathbf{v}} = \hat{\mathbf{v}}/\|\tilde{\mathbf{v}}\|_{\tilde{\mathbf{K}}}$.*

The proof is by directly substituting $\mathbf{C}_i, \hat{\mathbf{K}}, \mathbf{v}_*$ in [46] with $\tilde{\mathbf{C}}_i, \tilde{\mathbf{K}}, \hat{\mathbf{v}}$, respectively. Next, we establish the convergence theorem of RCaGP through the following proposition:

**Proposition F.3.** *Let $\mathcal{H}_k$ be the RKHS w.r.t. kernel $k$, $\sigma_{\mathrm{noise}}^2 > 0$ and let $\hat{\boldsymbol{\mu}}_* - \mathbf{m} \in \mathcal{H}_k$ be the unique solution to following empirical risk minimization problem*

$$\operatorname{argmin}_{f \in \mathcal{H}_k} \frac{1}{2n}\left(\underbrace{\mathbf{f}^\top \lambda^{-1}\mathbf{J}_{\mathbf{w}}^{-1}\mathbf{f} - 2\mathbf{f}^\top \lambda^{-1}\mathbf{J}_{\mathbf{w}}^{-1}(\mathbf{y} - \mathbf{m}_{\mathbf{w}})}_{L_n^w}\right) + \frac{1}{2}\|\mathbf{f}\|_{\mathcal{H}_k}^2 \qquad (S125)$$

*which is equivalent to the mathematical RCGP mean posterior shifted by prior mean $\mathbf{m}$. Then for the number of actions $i \in \{0, \ldots, n\}$ the RCaGP posterior mean $\hat{\boldsymbol{\mu}}_i$ satisfies:*

$$\|\hat{\boldsymbol{\mu}}_* - \hat{\boldsymbol{\mu}}_i\|_{\mathcal{H}_k} \leq \hat{\rho}(i)\,c(\mathbf{J}_{\mathbf{w}})\,\|\hat{\boldsymbol{\mu}}_* - \mathbf{m}\|_{\mathcal{H}_k} \qquad (S126)$$

*where $\hat{\rho}$ is the relative bound errors corresponding to the number of actions $i$ and the constant $c(\mathbf{J}_{\mathbf{w}}) = \sqrt{1 + \frac{\lambda_{\max}(\mathbf{J}_{\mathbf{w}})}{\lambda_{\min}(\mathbf{K})}} \to 1$ as $\lambda_{\max}(\mathbf{J}_{\mathbf{w}}) \to 0$.*

*Proof:*

Lemma F.1 implies there exists a unique solution to the corresponding RCGP risk minimization problem. Choosing $\hat{\rho}(i)$ as described in Proposition F.2, we have that $\|\hat{\mathbf{v}} - \tilde{\mathbf{v}}_i\|_{\tilde{\mathbf{K}}}^2 \leq \hat{\rho}(i)\|\hat{\mathbf{v}} - \tilde{\mathbf{v}}_0\|_{\tilde{\mathbf{K}}}$, where $\tilde{\mathbf{v}}_0 = \mathbf{0}$.

Then, for $i \in \{0, \dots, n\}$ we find that

$$\|\hat{\mathbf{v}} - \tilde{\mathbf{v}}_i\|_{\mathbf{K}}^2 \leq \|\hat{\mathbf{v}} - \tilde{\mathbf{v}}_i\|_{\tilde{\mathbf{K}}}^2 \leq \hat{\rho}^2(i)\|\hat{\mathbf{v}} - \tilde{\mathbf{v}}_0\|_{\tilde{\mathbf{K}}}^2 \tag{S127}$$

$$\leq \hat{\rho}(i)^2 \left( \|\hat{\mathbf{v}} - \tilde{\mathbf{v}}_0\|_{\mathbf{K}}^2 + \frac{\lambda_{\max}(\mathbf{J}_{\mathbf{w}})}{\lambda_{\min}(\mathbf{K})}\lambda_{\min}(\mathbf{K})\|\hat{\mathbf{v}} - \tilde{\mathbf{v}}_0\|_2^2 \right) \tag{S128}$$

$$\leq \hat{\rho}(i)^2 \left( \|\hat{\mathbf{v}} - \tilde{\mathbf{v}}_0\|_{\mathbf{K}}^2 + \frac{\lambda_{\max}(\mathbf{J}_{\mathbf{w}})}{\lambda_{\min}(\mathbf{K})}\|\hat{\mathbf{v}} - \tilde{\mathbf{v}}_0\|_{\mathbf{K}}^2 \right) \tag{S129}$$

$$\leq \hat{\rho}(i)^2 \left( 1 + \frac{\lambda_{\max}(\mathbf{J}_{\mathbf{w}})}{\lambda_{\min}(\mathbf{K})} \right) \|\hat{\mathbf{v}} - \tilde{\mathbf{v}}_0\|_{\mathbf{K}}^2 \tag{S130}$$

The third inequality stems from the definition of $\mathbf{J}_{\mathbf{w}}$ and the fact that the maximum eigenvalue of a diagonal matrix is the largest component of its diagonal. Assuming $\hat{\mu}_i(.) = m(.) + \sum_{j=1}^n (\tilde{\mathbf{v}}_i)_j k(., \mathbf{x}_j) = m(.) + k(., \mathbf{X})\tilde{\mathbf{C}}_i(\mathbf{y} - \mathbf{m}_{\mathbf{w}})$ and applying result from [46], we have that

$$\|\hat{\mathbf{v}} - \tilde{\mathbf{v}}_i\|_{\mathbf{K}}^2 = \|\hat{\boldsymbol{\mu}}_* - \hat{\boldsymbol{\mu}}_i\|_{\mathcal{H}_k}^2 \tag{S131}$$

Combining both results and defining $c(\mathbf{J}_{\mathbf{w}}) = \left( 1 + \frac{\lambda_{\max}(\mathbf{J}_{\mathbf{w}})}{\lambda_{\min}(\mathbf{K})} \right)$, we obtain

$$\|\hat{\boldsymbol{\mu}}_* - \hat{\boldsymbol{\mu}}_i\|_{\mathcal{H}_k} = \|\hat{\mathbf{v}} - \tilde{\mathbf{v}}_i\|_{\mathbf{K}} \leq \hat{\rho}(i)c(\mathbf{J}_{\mathbf{w}})\|\hat{\mathbf{v}} - \tilde{\mathbf{v}}_0\|_{\mathbf{K}} = \hat{\rho}(i)c(\mathbf{J}_{\mathbf{w}})\|\hat{\boldsymbol{\mu}}_* - \mathbf{m}\|_{\mathcal{H}_k} \tag{S132}$$

## F.3 SVGP with Relevance Pursuit

**Proposition F.4.** *Let $\mathcal{D}_{\setminus i} = \{(\mathbf{x}_j, y_j) : j \neq i\}, \mathbf{Z} = z_m$ is the inducing points, $\boldsymbol{\rho} = \boldsymbol{\rho}_{\setminus i} + \rho_i \mathbf{e}_i$, where $\boldsymbol{\rho}, \boldsymbol{\rho}_{\setminus i} \in \mathbb{R}_+^n, [\boldsymbol{\rho}_{\setminus i}]_i = 0$, and $\mathbf{e}_i$ is the ith canonical basis vector. Then keeping $\boldsymbol{\rho}_{\setminus i}$ fixed,*

$$\rho_i^* = \operatorname{argmax}_{\rho_i} \mathrm{ELBO}(\boldsymbol{\rho}_{\setminus i} + \rho_i \mathbf{e}_i) = \left[ \frac{1}{\sqrt{\mathbf{K}_{ii} - [\mathbf{Q}_{\mathbf{f}}]_{ii}}}(y_i - \mathbb{E}_q[y_i|\mathcal{D}_{\setminus i}]) - \mathbb{V}_q[y_i|\mathcal{D}_{\setminus i}] \right]_+ \tag{S133}$$

*where $y_i = f(\mathbf{x}_i) + \epsilon_i$. The quantities can be expressed as a function of $\mathbf{A}^{-1} = (\mathbf{Q}_{\mathbf{f}} + \mathbf{D}_{\sigma_{\mathrm{noise}}^2 + \boldsymbol{\rho}})^{-1}$ :*

$$\mathbb{E}_q[y_i|\mathcal{D}_{\setminus i}] = y_i - [\mathbf{A}^{-1}\mathbf{y}]_i/[\mathbf{A}^{-1}]_{ii} \quad and \quad \mathbb{V}_q[y_i|\mathcal{D}_{\setminus i}] = 1/[\mathbf{A}^{-1}]_{ii} \tag{S134}$$

*where $\mathbf{D}_{\sigma_{\mathrm{noise}}^2 + \boldsymbol{\rho}}$ is a diagonal matrix whose entries are $\sigma_{\mathrm{noise}}^2 + \boldsymbol{\rho}$.*

*Proof:* Following [41], the evidence lower bound (ELBO) of GP can be written as

$$-2\mathrm{ELBO}(\boldsymbol{\theta}) = n\log(2\pi) + \log\det(\mathbf{A}) + \mathbf{y}^\top \mathbf{A}^{-1}\mathbf{y} + \operatorname{tr}(\mathbf{D}_{\boldsymbol{\rho}+\sigma_{\mathrm{noise}}^2}^{-1}(\mathbf{K} - \mathbf{Q}_{\mathbf{f}})), \tag{S135}$$

where $\mathbf{A} = \mathbf{Q}_{\mathbf{f}} + \mathbf{D}_{\boldsymbol{\rho}+\sigma_{\mathrm{noise}}^2}$, and $\mathbf{Q}_{\mathbf{f}} = \mathbf{K}_{\mathbf{xz}}\mathbf{K}_{\mathbf{z}}^{-1}\mathbf{K}_{\mathbf{xz}}^\top$

Following [3], we partition matrix $\mathbf{A}$ to separate the effect of $\rho_i$ and use Schur's complement:

$$\mathbf{A}^{-1} = \begin{bmatrix} \mathbf{A}_{\setminus i}^{-1} + \hat{\mathbf{u}}\beta_i\hat{\mathbf{u}}^\top & -\hat{\mathbf{u}}\beta_i \\ -\hat{\mathbf{u}}^\top\beta_i & \beta_i \end{bmatrix}, \tag{S136}$$

where

$$\mathbf{A}_{\setminus i} = [\mathbf{Q}_{\mathbf{f}}]_{\setminus i} + \mathbf{D}_{\boldsymbol{\rho}_{\setminus i}+\sigma_{\mathrm{noise}}^2}, \tag{S137}$$

$$\hat{\mathbf{u}} = \mathbf{A}_{\setminus i}^{-1}\mathbf{K}_{\mathbf{xz}}\mathbf{K}_{\mathbf{z}}^{-1}k^\top(\mathbf{x}_i, \mathbf{Z}), \tag{S138}$$

$$\hat{\beta}_i = ([k(\mathbf{x}_i, \mathbf{x}_i) + \sigma_{\mathrm{noise}}^2 + \rho_i] - k(\mathbf{x}_i, \mathbf{Z})\mathbf{K}_{\mathbf{z}}^{-1}\mathbf{K}_{\mathbf{xz}}^\top\mathbf{A}_{\setminus i}^{-1}\mathbf{K}_{\mathbf{xz}}\mathbf{K}_{\mathbf{z}}^{-1}k^\top(\mathbf{x}_i, \mathbf{Z}))^{-1}. \tag{S139}$$

Quadratic term:

$$\mathbf{y}^\top(\mathbf{A} + \mathbf{D}_{\boldsymbol{\rho}+\sigma_{\mathrm{noise}}^2})^{-1}\mathbf{y} = \mathbf{y}_{\setminus i}^\top\mathbf{A}_{\setminus i}^{-1}\mathbf{y}_{\setminus i} + \beta_i(\mathbf{y}_{\setminus i}^\top\hat{\mathbf{u}} - y_i)^2. \tag{S140}$$

Determinant term:

$$\det(\mathbf{A}) = \det(\mathbf{Q}_{\mathbf{f}} + \mathbf{D}_{\boldsymbol{\rho}_i+\sigma_{\mathrm{noise}}^2}) = \det(\mathbf{A}_{\setminus i})\beta_i^{-1}. \tag{S141}$$

Following [3], we formulate the ELBO difference as

$$2(\mathrm{ELBO}(\boldsymbol{\rho}) - \mathrm{ELBO}(\boldsymbol{\rho}_{\setminus i})) = -\hat{\beta}_i(\mathbf{y}_{\setminus i}^\top\hat{\mathbf{u}} - y_i)^2 - \log(\hat{\beta}_i^{-1}) - \rho_i(\mathbf{K}_{ii} - [\mathbf{Q}_{\mathbf{f}}]_{ii}). \tag{S142}$$

The derivative of the difference in ELBO w.r.t. $\rho_i$, is

$$\partial_{\rho_i} 2(\mathrm{ELBO}(\boldsymbol{\rho}) - \mathrm{ELBO}(\boldsymbol{\rho}_{\setminus i})) = (\mathbf{y}_{\setminus i}^\top\hat{\mathbf{u}} - y_i)^2\hat{\beta}_i^2 - \hat{\beta}_i - (\mathbf{K}_{ii} - [\mathbf{Q}_{\mathbf{f}}]_{ii}). \tag{S143}$$

Using the quadratic formula, we obtain the solution $\hat{\beta}_{1,2} = \frac{1 \pm \sqrt{1 + 4(\mathbf{y}_{\backslash i}^\top \hat{\mathbf{u}} - y_i)^2 (\mathbf{K}_{ii} - [\mathbf{Q_f}]_{ii})}}{2(\mathbf{y}_{\backslash i}^\top \hat{\mathbf{u}} - y_i)^2}$. Note that $(\mathbf{y}_{\backslash i}^\top \hat{\mathbf{u}} - y_i)^2 > 0$ and $\mathbf{K}_{ii} - [\mathbf{Q_f}]_{ii} \geq 0$. Therefore, the smaller root $\rho_2$ is negative when $(\mathbf{K}_{ii} - [\mathbf{Q_f}]_{ii}) > 0$ and zero when $(\mathbf{K}_{ii} - [\mathbf{Q_f}]_{ii}) = 0$. Based on the $\hat{\beta}$ formulation (predictive variance cannot be zero), we reject $\hat{\beta}_2 < 0$. Similarly, we reject $\hat{\beta}_2 = 0$, following [3]. By ignoring the constant term in $\hat{\beta}_2$ and following the approach of [3], we solve $\hat{\beta}_2^{-1} = 1/\sqrt{(\mathbf{K}_{ii} - [\mathbf{Q_f}]_{ii})}(\mathbf{y}_{\backslash i}^\top \hat{\mathbf{u}} - y_i)$ for $\rho_i$ and projecting to non-negative half-line, we get

$$\rho_i = [\frac{1}{\sqrt{(\mathbf{K}_{ii} - [\mathbf{Q_f}]_{ii})}}(\mathbf{y}_{\backslash i}^\top \hat{\mathbf{u}} - y_i) - (k(\mathbf{x}_i, \mathbf{x}_i) + \sigma_{\text{noise}}^2 - k(\mathbf{x}_i, \mathbf{Z})\mathbf{K_z}^{-1}\mathbf{K_{xz}^\top}\mathbf{A}_{\backslash i}^{-1}\mathbf{K_{xz}}\mathbf{K_z}^{-1}k^\top(\mathbf{x}_i, \mathbf{Z}))]_+. \tag{S144}$$

Lastly, we note that

$$\mathbf{y}_{\backslash i}^\top \hat{\mathbf{u}} - y_i = \mathbf{y}_{\backslash i}^\top(\mathbf{A}_{\backslash i}^{-1}\mathbf{K_{xz}}\mathbf{K_z}^{-1}k^\top(\mathbf{x}_i, \mathbf{Z})) - y_i = \mathbb{E}_q[\mathbf{y}_i|\mathcal{D}_{\backslash i}] - y_i, \tag{S145}$$

$$k(\mathbf{x}_i, \mathbf{x}_i) + \sigma_{\text{noise}}^2 - k(\mathbf{x}_i, \mathbf{Z})\mathbf{K_z}^{-1}\mathbf{K_{xz}^\top}\mathbf{A}_{\backslash i}^{-1}\mathbf{K_{xz}}\mathbf{K_z}^{-1}k^\top(\mathbf{x}_i, \mathbf{Z}) = \mathbb{V}_q[\mathbf{y}_i|\mathcal{D}_{\backslash i}]. \tag{S146}$$

Following [36], we have that

$$\mathbb{E}_q[\mathbf{y}_i|\mathcal{D}_{\backslash i}] = y_i - [\mathbf{A}^{-1}\mathbf{y}]_i/[\mathbf{A}^{-1}]_i, \tag{S147}$$

$$\mathbb{V}_q[\mathbf{y}_i|\mathcal{D}_{\backslash i}] = 1/[\mathbf{A}^{-1}]_{ii}, \tag{S148}$$

which follow the LOO predictive values.

# G  Details of Expert-guided Robust Mean Prior

In this section, we provide the posterior inference details of our probabilistic user model for defining the informative mean prior. We first discuss the posterior inference of latent variables $\bar{\delta}_o$, governing human expert decisions in identifying outliers. For every outlier candidate $(\mathbf{x}_o, \hat{y}_o)$, we have that

$$\begin{aligned} p(\bar{\delta}_o) &= \mathcal{B}(\alpha = \alpha_o, \beta = \beta_o) && \text{prior,} && \text{(S149)}\\ p(\bar{o}_o \,|\, \bar{\delta}_o) &= \bar{\delta}_o \bar{o}_o + (1 - \bar{\delta}_o)(1 - \bar{o}_o) && \text{likelihood,} && \text{(S150)}\\ p(\bar{\delta}_o|\bar{o}_o) &\propto p(\bar{\delta}_o)\, p(\bar{o}_o|\bar{\delta}_o) && \text{posterior} \\ &= \mathcal{B}(\alpha = \alpha_o, \beta = \beta_o)\left(\bar{\delta}_o(\bar{o}_o) + (1 - \bar{\delta}_o)(1 - \bar{o}_o)\right) \\ &= \mathcal{B}(\alpha = \alpha_o + \bar{o}_o, \beta = \beta_o + 1 - \bar{o}_o), && && \text{(S151)} \end{aligned}$$

where we define the hyperparameters $\alpha_0$ and $\beta_0$ as follows:

$$\alpha_o = |z_o| = \left|\frac{\hat{y}_o - \bar{\mu}}{\bar{\sigma}}\right| \qquad \text{and} \qquad \beta_o \approx 0.$$

The posterior $p(\bar{\delta}_o|\bar{o}_o)$ exhibits conjugacy, allowing us to derive a closed-form expression, i.e., Beta distribution. In this framework, we define $\alpha_o$ using the z-score of the outlier candidates. The z-score information effectively recognizes outliers by comparing each data point to the sample mean and standard deviation. Next, we set $\beta_o$ close to zero, reflecting that the expert is confident that $\bar{y}_o$ is an outlier. Given the posterior $p(\bar{\delta}_o|\bar{o}_o)$, the expected value $\mathbb{E}_{p(\bar{\delta}_o|\bar{o}_o)}[\bar{\delta}_o]$ is expressed by:

$$\mathbb{E}_{p(\bar{\delta}_o|\bar{o}_o)}[\bar{\delta}_o] = \frac{\alpha}{\alpha + \beta} = \frac{\alpha_o + \bar{o}_o}{\alpha_o + \beta_o + 1} \tag{S152}$$

Next, we outline the posterior inference details for the outlier corrections provided by the human expert:

$$\begin{aligned} p(\bar{\mu}_o) &= \mathcal{N}(\mu = \mu_o, \sigma^2 = \tau_o^{-1}) && \text{prior,} && \text{(S153)}\\ p(\bar{y}_o|\bar{\mu}_o) &= \mathcal{N}(\mu = \bar{\mu}_o, \sigma^2 = \sigma_{\text{corr.}}^2) && \text{likelihood,} && \text{(S154)}\\ p(\bar{\mu}_o|\bar{y}_o) &\propto p(\bar{\mu}_o)\, p(\bar{y}_o|\bar{\mu}_o) && \text{posterior} && \text{(S155)}\\ &= \mathcal{N}(\mu = \mu_o, \sigma^2 = \tau_o^{-1})\mathcal{N}(\mu = \bar{\mu}_o, \sigma^2 = \sigma_{\text{corr.}}^2) && && \text{(S156)}\\ &= \mathcal{N}\left(\mu = \frac{\tau_o\,\mu_o + \sigma_{\text{corr.}}^{-2}.\bar{y}_o}{\tau_o + 1.\sigma_{\text{corr.}}^{-2}}, \sigma^2 = (\tau_o + 1.\sigma_{\text{corr.}}^{-2})^{-1}\right), && && \text{(S157)} \end{aligned}$$

where we define the hyperparameters $\mu_o$ and $\tau_o$ as follows:

$$\mu_o = \frac{1}{J}\sum_{y_{\hat{j}} \in N_J(\mathbf{x}_o)} y_{\hat{j}} \qquad \text{and} \qquad \tau_o^{-1} = \frac{1}{J-1}\sum_{y_{\hat{j}} \in N_J(\mathbf{x}_o)} (y_{\hat{j}} - \mu_o)^2.$$

Here, $N_J(\mathbf{x}_o)$ denotes a set consisting of $J$ closest neighbors to $\mathbf{x}_o$. Specifically, we compute a kernel function $k(\mathbf{x}_o, \mathcal{D} \setminus \bar{\mathcal{D}}) \in \mathbb{R}^{1 \times (n - \bar{o})}$, where $\bar{\mathcal{D}}$ represents inliers. Then, we retrieve $\{\mathbf{x}_1, \ldots, \mathbf{x}_J\}$ corresponding with the $J$ largest $k(\mathbf{x}_o, \mathcal{D} \setminus \bar{\mathcal{D}}) \in \mathbb{R}^{1 \times (n - \bar{o})}$. We set $J = 3$ for our experiments, ensuring each outlier has four neighbors. This posterior also maintains conjugacy, yielding a normally distributed posterior. For each identified outlier $y_o$, we define the prior for the latent variable $\bar{\mu}_o$ using the sample mean and variance of its neighbors. Intuitively, this prior reflects the assumption that a human expert's correction for a particularly identified outlier will be close to its neighbors. This prior resembles a probabilistic model inspired by case-based reasoning. We then obtain the expected value $\mathbb{E}_{p(\bar{\mu}_o | \bar{y}_o)}$ as

$$\mathbb{E}_{p(\bar{\mu}_o | \bar{y}_o)} = \frac{\tau_o \, \mu_o + \sigma_{\text{corr.}}^{-2} \bar{y}_o}{\tau_o + 1.\sigma_{\text{corr.}}^{-2}}, \tag{S158}$$

which equals the posterior mean. The algorithm to implement the informative mean prior is presented in Algorithm 1.

---

**Algorithm 1** Expert-guided robust mean prior algorithm

---

1: **Input**: $\mathcal{D}, \bar{\mathcal{D}}, \bar{\mathbf{o}}, \bar{\mathbf{y}}, \sigma_{\text{corr.}}^2$
2: $\bar{\mu} \leftarrow \frac{1}{|\mathcal{D}|} \sum_{j=1}^{|\mathcal{D}|} y_j$
3: $\bar{\sigma} \leftarrow \sqrt{\frac{1}{|\mathcal{D}|-1} \sum_{j=1}^{|\mathcal{D}|} (y_j - \bar{\mu})^2}$
4: **for** $o = 1, \ldots, \hat{o}$ **do**
5: $\quad \alpha_o \leftarrow |(\hat{y}_o - \bar{\mu})/\bar{\sigma}|$
6: $\quad \beta_o \leftarrow 0$
7: $\quad \mu_o \leftarrow \mu_o = \frac{1}{J} \sum_{y_{\hat{j}} \in N_J(\mathbf{x}_o)} y_{\hat{j}}$
8: $\quad \tau_o \leftarrow \frac{1}{J-1} \sum_{y_{\hat{j}} \in N_J(\mathbf{x}_o)} (y_{\hat{j}} - \mu_o)^2$
9: $\quad \mathbb{E}_{p(\bar{\delta}_o | \bar{o}_o)}[\bar{\delta}_o] \leftarrow \frac{\alpha_o + \bar{o}_o}{\alpha_o + \beta_o + 1}$
10: $\quad \mathbb{E}_{p(\bar{\mu}_o | \bar{y}_o)}[\bar{\mu}_o] \leftarrow \frac{\tau_o \mu_o + \sigma_{\text{corr.}}^{-2} \bar{y}_o}{\tau_o + \sigma_{\text{corr.}}^{-2}}$
11: **end for**
12: $m(\mathbf{x}) \leftarrow \frac{1}{\hat{o}} \sum_o \mathbb{E}_{p(\bar{\delta}_o | \bar{o}_o)}[\bar{\delta}_o] \, \mathbb{E}_{p(\bar{\mu}_o | \bar{y}_o)}[\bar{\mu}_o]$
13: **Output:** mean prior $m(\mathbf{x})$

---

# H  Benchmark details

## H.1  UCI regression datasets

**Boston**  The dataset consists of $n = 506$ observations, each representing a suburban or town area in Boston. It encompasses $d = 13$ features containing data like the average number of rooms in dwellings, pupil-teacher ratios, and per capita crime rates. We try to predict the median price of homes residents own (excluding rented properties). The dataset can be found at https://www.cs.toronto.edu/~delve/data/boston/bostonDetail.html.

**Energy**  The dataset describes the energy efficiency of buildings by correlating their heating and cooling load requirements with various building parameters. It consists of $n = 768$ data samples, each characterised by $d = 8$ distinct features, with the ultimate goal of predicting a single continuous response variable found in the last column. The dataset can be found at https://archive.ics.uci.edu/dataset/242/energy+efficiency.

**Yacht**  The dataset's main focus is on predicting the residuary resistance of sailing yachts during their initial design phase, a critical aspect in evaluating a vessel's performance and estimating the essential propulsive power required. This prediction relies on $d = 6$ primary input parameters, which include the fundamental hull dimensions and boat velocity. The dataset contains $n = 308$ observations. The dataset can be found at https://archive.ics.uci.edu/dataset/243/yacht+hydrodynamics.

**Parkinsons**  The Parkinsons Telemonitoring dataset contains $n = 197$ data samples and $d = 22$ input features derived from voice measures. The dataset comprises a collection of biomedical voice measurements from 31 individuals, including 23 diagnosed with Parkinson's disease (PD). Each row represents one of 195 voice recordings, while each column corresponds to a specific vocal feature.. The dataset can be found at https://archive.ics.uci.edu/dataset/174/parkinsons.

## H.2 High-throughput Bayesian Optimization problems

**Hartmann 6D.** The widely used Hartmann benchmark function [39].

**Lunar Lander.** The goal of this task is to find an optimal 12-dimensional control policy that allows an autonomous lunar lander to consistently land without crashing. The final objective value we optimize is the reward obtained by the policy averaged over a set of 50 random landing terrains. For this task, we use the same controller setup used by [9].

**Rover.** The rover trajectory optimization task introduced by [45] consists of finding a 60-dimensional policy that allows a rover to move along some trajectory while avoiding a set of obstacles. We use the same obstacle set up as in [30].

**Lasso DNA.** We optimize the $180-$dimensional DNA task from the LassoBench library [49] of benchmarks based on weighted LASSO regression.

## H.3 Additional outlier contamination protocols on UCI regression datasets

Our outlier contamination protocols follow the same settings as [2]. We recall the asymmetric outlier case described in the main text in Section 7.

**Asymmetric** We sample uniformly at random 10% of the training dataset input-output pairs $(\mathbf{x}_i, y_i)$, and replace the $y_i$'s by asymmetric outliers, i.e., *via* subtraction of noise sampled from a uniform distribution $\mathcal{U}(3\bar{\sigma}, 9\bar{\sigma})$, with $\bar{\sigma}$ being the standard deviation of the original observations.

Next, Section 7.4 considers two additional kind of outliers:

**Uniform** Instead of always subtracting noise as was the case for asymmetric outliers, the set of training data points selected for outlier contamination is divided in 2 subsets: half of the selected subset is contaminated by adding $z \sim U(3\sigma, 9\sigma)$, while the other half is contaminated by subtracting $z \sim U(3\sigma, 9\sigma)$.

**Focused** In this outlier generation process, we randomly select and remove a subset of data points, which will be replaced by outliers. For these outliers, we deterministically choose their values in $\mathcal{X}$. To do so, we calculate the median value for each input data dimension $j$. However, we do not place the outliers at this median position directly. Instead, we replace the removed input values by $(m_1 + \delta_1, m_2 + \delta_2 \ldots, m_d + \delta_d)^\top$, where $m_j$ is the median in the $j$-th input data dimension, and $\delta_j = \alpha_j u$, where $\alpha_j$ is the median absolute deviation of the $j$-th data dimension times 0.1, and $u \sim U(0,1)$. Simultaneously, the outlier values on $\mathcal{Y}$ are obtained by subtracting three times the standard deviation of the median of the observations $M_y$. To not have the same value for every outlier position, we also add a small perturbation $\delta_y = \alpha_y u$, where $\alpha_y$ is the median absolute deviation of $\mathbf{y}$ times 0.1, and $u \sim U(0,1)$.

# I Experimental Details

## I.1 Hardware

For the UCI regression experiments, all models—including our proposed method and the baselines—were executed on a compute cluster consisting of two machines: one equipped with dual AMD EPYC 7713 processors (64 cores each, 2.0 GHz), and another with dual Intel Xeon Gold 6148 processors (20 cores each, 2.4 GHz). The longest-running UCI task, based on the Energy dataset, required approximately 15 minutes to complete, including both our model and all baseline methods. Bayesian Optimization experiments were conducted on a cluster with four NVIDIA V100 GPUs, each with 32 GB of memory. Among these tasks, the Hartmann 6D benchmark was the fastest, completing in roughly 20 minutes, while the most computationally intensive task—DNA—required up to 20 hours of runtime.

## I.2   Hyperparameters

| UCI regression | |
| --- | --- |
| Optimizer | ADAM |
| Learning rate | 0.01 |
| Minibatch size | $n$-data |
| Number of iterations for optimizing ELBO | 50 |
| Proportion of test set | 0.2 |
| $\epsilon$ | 0.2 |
| $p$-outliers | 0.1 |
| $n$-inducing points (SVGP) | 100 |
| Projection-dim (RCaGP) | 5 |
| Mean-prior $m$ (RCaGP and RCSVGP) | $\frac{1}{n}\sum_j^n y_j$ |
| Mean-prior $m$ (SVGP) | 0 |
| $\beta$ | 1.0 |
| $\sigma^2_{\text{corr.}}$ | 1.0 |
| **High-throughput BO** | |
| $n_0$ (data initialization) | 250 |
| ADAM step size for query $\mathbf{x}$ | 0.001 |
| ADAM step size for RCaGP parameters | 0.01 |
| The number of expert corrections | $T_{\text{iterations}}/20$ |
| Minibatch size | 250 |
| $\sigma^2_{\text{corr.}}$ | 1.0 |
| $p-$outliers | 0.25 |
| Mean-prior $m$ (all models) | 0.0 |
| Projection-dim (RCaGP) | 25 |
| $n$-inducing points (SVGP) | 25 |

Table S4: Hyperparameter settings used for RCaGP. Most of them remain consistent across all the tasks.

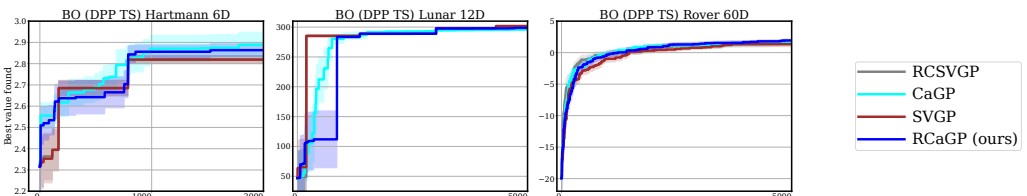

Figure S1: **High-throughput Bayesian Optimization task under asymmetric outliers using DPP-BO and the Thompson Sampling (TS) acquisition strategy.** Each panel shows the best value found each iteration found so far, averaged across 20 repetitions ± 1 std.

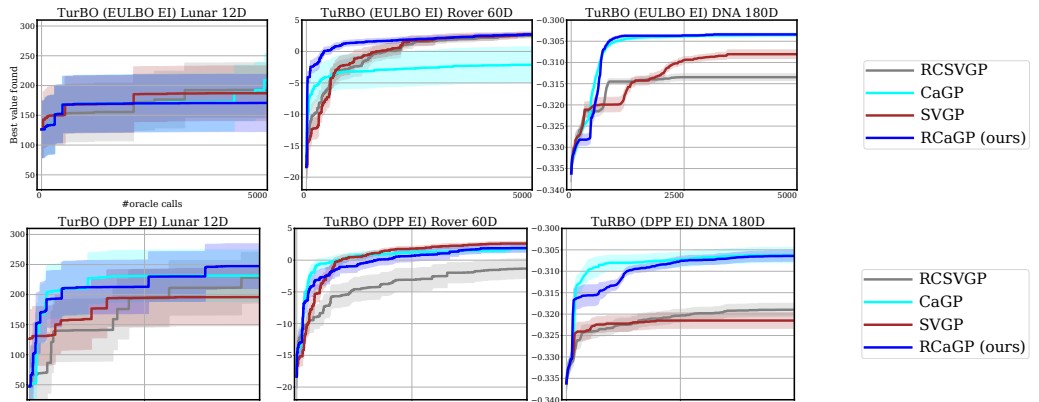

Figure S2: **High-throughput Bayesian Optimization task using TurBO under asymmetric outliers.** Each panel shows the best value found each iteration found so far, averaged across 20 repetitions ± 1 std. **Top row:** EULBO-TuRBO-EI. **Bottom row:** DPP-TuRBO-EI.

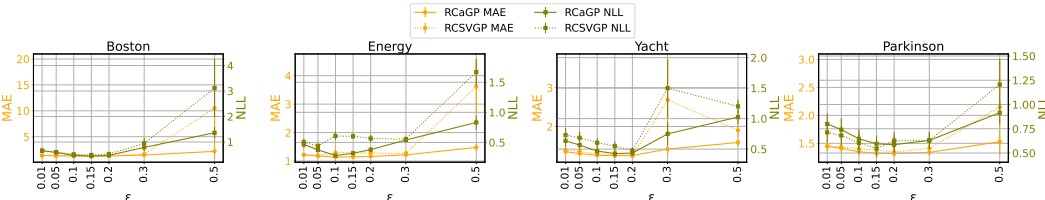

Figure S3: Dataset-specific results for the ablation study conducted in Section 7.4, varying $c$ in weight function $w$ (Equation 12).

