# OpenReview forum: "Robust and Computation-Aware Gaussian Processes"
_NeurIPS.cc/2025/Conference — NeurIPS 2025 poster_

### Official Review · Reviewer_Q5xx · 2025-06-12

**Clarity:** 3
**Significance:** 2
**Originality:** 2
**Rating:** 4
**Confidence:** 4

**Summary:**

The work proposes to combine the idea of computation-aware gaussian processes (CaGP) with robust conjugate GP (RCGP). This way the proposed method allows for conservative uncertainty estimates
and robustness to outliers.
Experiments on standard regression tasks and Bayesian optimization tasks suggest that the proposed method leads to better uncertainty estimate with and without outliers when compared to CaGP and RCGP.

**Questions:**

- It seems the weight function requires that an expert labels the outliers.
How are these labels exactly chosen in the experiments of Table 1 and Figure 3?

- How is $\beta$ of Equation (12) chosen?

- How is the projection dimensionality chosen.

- Clarification of the above questions and additional results with the method from [Ament, 2024] (see Weakness section) would affect my scoring.

**Ethical Concerns:**

["NO or VERY MINOR ethics concerns only"]

**Final Justification:**

The main reason for my decision are:
The paper is technically sound and several additional experiments including the comparison to [Ament, 2024] were presented in response to my questions, leading me to increase my score.

**Limitations:**

no, the choice of all the hyper-parameters is not discussed as a limitation.

**Quality:**

3

**Strengths And Weaknesses:**

Strength:
- Extends theoretical results from CaGP and RCGP to their proposed method.
- Empirical evaluation showing that proposed method improves uncertainty quantification compared to CaGP and RCGP

Weakness:
- The proposed method inherits from CaGP and RCGP various hyper-parameters like the projection dimension, choice of mean function, choice of weighting function (= choice of $\beta$ and $c$).
For choosing the mean function, the work assumes that (noisy) expert labels are available that flag potential outliers. This seems to be a rather strong assumption.
It is also unclear how the other hyper-parameters are chosen (except $c$ which is explained in the experiments section)
- RCGP is the only other robust method that is compared.
However, like their proposed method RCGP is known for its sensitivity to mean prior.
I would find it more convincing, if the work compared also to the method of [Ament, 2024].
- I guess the projection dimensionality directly affects the computational costs.
Since one of the main claim is computational awareness, I would like to see how the results of the proposed method change, if less computational resources are used.
In particular, in Table 1, currently the proposed method is almost always more computationally expensive than RCSVGP.
- Proposition 4.2 and Equation (22) are hard to understand. (I guess there is some type in (22)?)

References:
[Ament, 2024]
Ament, Sebastian, et al. "Robust Gaussian processes via relevance pursuit." Advances in Neural Information Processing Systems 37 (2024): 61700-61734.

---

> ### Author Rebuttal · Authors · 2025-07-31
>
> Thank you for your thoughtful feedback. We are grateful for your recognition of the extended theoretical results and the improvements in uncertainty quantification.
>
> Q1: It seems the weight function requires that an expert labels the outliers. How are these labels exactly chosen in the experiments of Table 1 and Figure 3?
>
> A1: This question is reminiscent of Q4 from 8Bte, we paste the answer here:
> >In our work, we considered two formulations for $m$:
> >- A constant mean with a learnable constant.
> >- An expert-driven prior mean, detailed in Section 5 and Appendix G.
>
> >In Table 1, Figure 3, Figure 4 (left), Table S1, Table S2, Figure S1, Figure S2, and Figure S3, we use only the constant mean for the RCSVGP and RCaGP baselines. These are labeled either simply as “RCSVGP” and “RCaGP” (Table 1, Figure 3, Figure S1–S3), or as “RCSVGP (const.)” and “RCaGP (const.)” (Table S1 and S2). As shown in Sections 7.1 and 7.2, our proposed method RCaGP (const.) yields the best overall performance.
>
> >The expert-driven prior mean is evaluated only in a dedicated ablation study (Section 7.3, paragraph two), with results shown in the right panel of Figure 4. There, the baselines are labeled “RCSVGP (exp.)” and “RCaGP (exp.)”. These results show that the expert-driven prior can improve performance in BO. However, this experiment assumes perfect expert correction of all outliers, which is unrealistic in practice. This ablation study should therefore be seen as a proof of concept. In future work, we aim to extend this strategy to scenarios with partial and noisy expert supervision.
>
> >__We apologize for any confusion caused by inconsistent labeling and will revise the manuscript for clarity.__
>
> Q2: How is $\beta$ (Equation 12) chosen?
>
> A2: We answered the question regarding $\beta$ in Q2 to reviewer jPaV.
>
> Q3: How is the projection dimensionality chosen.
>
> A3: For both UCI and BO experiments, we first tried several candidates on one task and chose the one that provided the best results. Then, we fixed this dimensionality across all tasks. We acknowledge that selecting the optimum dimensionality remains a problem for the computation-aware GP. in A6 to 8Bte, we also varied the projection dimensionality in a dedicated ablation study.
>
> Q4 (2 in Weaknesses): RCGP is the only other robust method that is compared. However, like their proposed method RCGP is known for its sensitivity to mean prior. I would find it more convincing, if the work compared also to the method of [Ament, 2024].
>
> A4: We did our best to integrate the method proposed by Ament et al. as an additional baseline. Given that all baselines employ a form of approximate inference, the most straightforward way to achieve this is to derive an SVGP model with relevance pursuit, as proposed by the following lemma:
>
> Proposition: Let $\mathcal{D}{\setminus i} = {(\mathbf{x}j, y_j): j \ne i}$ be the dataset excluding point $i$. Let $\mathbf{Z} = {z_m}$ be the inducing points, and define $\boldsymbol{\rho} = \boldsymbol{\rho}{\setminus i} + \rho_i \mathbf{e}i$, where $\boldsymbol{\rho}, \boldsymbol{\rho}{\setminus i} \in \mathbb{R}+^n$, $[\boldsymbol{\rho}_{\setminus i}]_i = 0$, and $\mathbf{e}i$ is the $i$-th canonical basis vector. Fixing $\boldsymbol{\rho}{\setminus i}$:
>
> $$\rho_i^\ast = \arg\max_{\rho_i}  \mathrm{ELBO}(\boldsymbol{\rho}_{\setminus i} + \rho_i \mathbf{e}_i)  = [(y_i - \mathbb{E}_q[y_i | \mathcal{D}{\setminus i} ])^2 - \mathbb{V}_q[y_i \vert \mathcal{D}{\setminus i}]]{+}$$
>
> where $y_i = f(x_i) + \epsilon_i$. The quantities can be expressed as a function of $\mathbf{A}^{-1} = (\mathbf{Q}_\mathbf{f} + \mathbf{D}(\sigma^2 + \boldsymbol{\rho}))^{-1}$:
>
> $$\mathbb{E}_q[y_i \vert \mathcal{D}{\setminus i}] = y_i - [\mathbf{A}^{-1} \mathbf{y}]_i / [\mathbf{A}^{-1}]{ii} \quad \text{and} \quad \mathbf{V}_q[y_i \vert \mathcal{D}{\setminus i}] = 1 / [\mathbf{A}^{-1}]{ii}
> $$
>
> where $\mathbf{D}_{\sigma^2 + \boldsymbol{\rho}}$ is a diagonal matrix whose entries are $\sigma^2 + \boldsymbol{\rho}$, and $\mathbf{Q}\mathbf{f} = \mathbf{K}{\mathbf{x} \mathbf{z}} \mathbf{K}\mathbf{z}^{-1} \mathbf{K}{\mathbf{x} \mathbf{z}}^\top$.
>
> We do not include the proof due to space limitations. Indeed, we have updated our manuscript with the full proof of this new class of SVGP. The idea is to consider ELBO by Titsias (2009), which has a closed form. The additional term trace between K and Q does not affect rho during model selection; thus, the derivation follows naturally. Experiments involving this novel baseline are currently running. We will post them as soon as possible.
>
> Q5 (4 in Weaknesses): Proposition 4.2 and Equation (22) are hard to understand. (I guess there is some type in (22)?)
>
> A5: There is indeed a typo in Equation (22): a semicolon and a comma have been wrongly inserted. Thank you for noticing that.
>
> Intuition behind Proposition 4.2: Just as a standard Gaussian Process (GP) accounts for the discrepancy between the true function and its posterior mean via posterior variance—reflecting uncertainty due to limited data—RCaGP introduces an additional layer. Specifically, RCaGP considers the discrepancy between its own posterior mean and that of the exact GP by quantifying computational uncertainty, which arises from approximations made in resource-constrained (limited computation) settings. Consequently, RCaGP captures the total discrepancy between the true function and its posterior mean through a combined uncertainty: the sum of mathematical uncertainty (from limited data) and computational uncertainty (from approximate inference).

---

> ### Author Response · Authors · 2025-08-02
> **results for relevance pursuit SVGP and ELBO term correction**
>
> We have updated our SVGP with relevance pursuit formulation. Specifically, we update the trace term of the ELBO formulation since it also depends on $\mathbf{D}_{\boldsymbol{\rho} + \sigma^2}$.
>
>  In the proof, we show that the trace difference is scaled by
>
> $\mathbf{K}_{ii} - [\mathbf{Q}_{\mathbf{f}}]_{ii}$
>
> Furthermore, the second root of $\beta_i$ will be negative or zero, depending on
>
>  $\mathbf{K}_{ii} - [\mathbf{Q}_{\mathbf{f}}]_{ii}$
>
>  In both cases, we reject this beta, leaving the positive case only. Next, we report the results for SVGP with relevance pursuit. We choose 100 inducing points, exactly the same as RCSVGP and SVGP. For the $\mathcal{K}$ schedule, we choose $\{ 0.05n, 0.05n \}$, where $n$ denotes the number of observations and there is $0.1$ fraction of outliers. For fair comparison, we optimize $\boldsymbol{\rho}$ with the Adam optimizer with 50 iterations using $0.01$ learning rate. Other settings exactly follow the main experiment on UCI dataset.
>
> $\frac{1}{\sqrt{\mathbf{K}_{ii} - [\mathbf{Q_f}]_{ii}}}$
>
> These results, to be compared with the left part of Table 1 (asymmetric outliers), show that RCaGP outperforms SVGP with relevance pursuit on all datasets, except Energy.
>
> | Method | Metric | Boston         | Yacht          | Energy         | Parkinsons      |
> |--------|--------|----------------|----------------|----------------|-----------------|
> | RRP    | MAE    | 0.550 ± 0.050  | 0.597 ± 0.112  | 0.356 ± 0.051  | 0.644 ± 0.106   |
> |        | NLL    | 1.147 ± 0.050  | 1.240 ± 0.117  | 0.896 ± 0.030  | 1.306 ± 0.121   |

---

> ### Comment · Reviewer_Q5xx · 2025-08-04
> **regarding computational costs**
>
> Thank you so much for the clarifications and the additional comparison to [Ament, 2024].
> Finally, could you please comment on the computational costs as I wrote in my previous review:
> "I guess the projection dimensionality directly affects the computational costs. Since one of the main claim is computational awareness, I would like to see how the results of the proposed method change, if less computational resources are used. In particular, in Table 1, currently the proposed method is almost always more computationally expensive than RCSVGP."

---

> > ### Author Response · Authors · 2025-08-05
> >
> > Thank you for raising this important point. While projection dimensionality $d$ might appear to influence computational cost, RCaGP's overall clock time remains largely invariant to $d$. This is because the number of projected actions scales inversely with $d$, i.e., approximately $n//d$, resulting in a total of $(n//d) * d = n$ trainable parameters—effectively constant with respect to $d$. Thus, increasing $d$ reduces the number of projected actions, balancing out the increased cost per action.

---

### Official Review · Reviewer_8Bte · 2025-06-18

**Clarity:** 4
**Significance:** 3
**Originality:** 3
**Rating:** 5
**Confidence:** 4

**Summary:**

This paper addresses the intertwined challenges of computational scalability and model robustness in Gaussian Processes when applied to large datasets corrupted by outliers. The authors introduce the RCaGP, whose core contribution is the principled integration of two key ideas: (1) leverages the mechanism of RCGPs to down-weight the influence of outliers, (2) simultaneously incorporating the approach of CaGPs to quantify and compensate for uncertainty arising from computational approximations, thereby preventing overconfident predictions. The paper also provides theoretical guarantees for RCaGP's robustness and conservative uncertainty estimates.

**Questions:**

(1) The effectiveness of the algorithm does not seem to be limited to the Matérn kernel function. It would be beneficial to include some discussion and experiments regarding other common GP kernel functions?
(2) Outliers are caused by noise. Is there any discussion regarding the outliers generated by different magnitudes of noise? It seems that GPs perform differently under varying noise levels.
(3) In the BO experiments, while using 250 points as the initial sample size ensures a fair comparison, this number is relatively large. For instance, 20 points are often sufficient for a 6-dimensional problem. This large initial set might mask the differences in sample efficiency among the models during the early exploration phase?

**Ethical Concerns:**

["NO or VERY MINOR ethics concerns only"]

**Final Justification:**

I am raising my score to 5 as I am satisfied with the authors’ response and the additional experiments conducted. After reviewing the discussion with other reviewers, I believe the paper now presents sufficient content. I encourage the authors to integrate the discussed revisions into the revised manuscript or appendix to further strengthen the work.

**Limitations:**

(1) The paper's core mechanism for achieving robustness through the weight function w(x,y) is critically dependent on a high-quality prior mean m(x). Algorithm identifies outliers by measuring the deviation of an observation y from m(x). This means that if m(x) is poorly specified, the model may erroneously down-weight normal data points or fail to identify true outliers, leading to a degradation in the performance of the entire robustness framework.

**Paper Formatting Concerns:**

No.

**Quality:**

3

**Strengths And Weaknesses:**

Strengths:
(1) The paper is generally well written and well presented.
(2) The proposed RCaGP model is not merely a simple superposition of existing methods (RCGP and CaGP), but rather a principled integration within a unified probabilistic framework. This approach of jointly solving the problem surpasses previous work that has tackled these challenges in isolation.
(3) The paper provides theoretical support for the reliability of RCaGP. The model's robustness is proven through a bounded Posterior Influence Function (PIF), which stands in sharp contrast to the unbounded PIF of CaGP.

Weaknesses:
(1) Parameters such as the soft threshold c, the low-rank approximation dimension, and the size of the approximate action matrix require manual setting or cross-validation by the user, which could otherwise affect the balance between robustness and approximation quality.
(2) The paper highlights the model's critical dependence on the prior mean m(x). The success of this mechanism hinges entirely on the quality of m(x). The use of an expert-guided prior is intended to directly address this weakness in model's robustness mechanism by providing a high-quality "initial guess" to enhance its core ability to identify and handle outliers. While the proposed expert-guided method is a good solution, it also intro a potential weakness of the model: in the complete absence of domain knowledge, how to set a reasonable prior mean remains an open question. If the domain expert's annotations are biased or insufficient, the constructed mean prior could potentially introduce misleading information.
(3) Although the paper mentions that only posterior expectations are used for computational efficiency, requiring experts to review and correct a certain proportion of potential outliers in large-scale datasets is a process that is inherently difficult to scale, which seems to conflict with the paper's highlight on High-Throughput applications.
(4) In the "Related Work" section, the paper mentions the first category of robust GP methods, which replace the Gaussian noise assumption with heavy-tailed distributions. Although the authors note that these methods often break conjugacy and require approximate inference, a direct comparison of RCaGP with a representative model from this category (e.g., an SVGP based on a Student-t likelihood) would more forcefully demonstrate whether RCaGP, while maintaining its advantage of conjugacy, still leads in terms of performance.

---

> ### Author Rebuttal · Authors · 2025-07-31
>
> Thank you for your kind and constructive review. We sincerely appreciate your recognition of the unified design of RCaGP, its theoretical grounding, and the clarity of the presentation.
>
> Q1: The effectiveness of the algorithm does not seem to be limited to the Matérn kernel function. It would be beneficial to include some discussion and experiments regarding other common GP kernel functions?
>
> A1: We generated a table similar to Table 1 but, experimenting with the RBF kernel in place of the Matern kernel. The results demonstrate the overall superiority of RCaGP both in terms of MAE and NLL, as our proposed method is only outperformed by RCSVGP on 2 out of 8 cases: Boston and Parkinsons with asymmetric outliers. The table spans 3000 characters and cannot be posted within this comment, it has been posted in the rebuttal answer associated with reviewer uEzz.
>
> Q2: Outliers are caused by noise. Is there any discussion regarding the outliers generated by different magnitudes of noise? It seems that GPs perform differently under varying noise levels.
>
> A2: We conducted an ablation study by running experiments on the Yacht dataset. We then vary the magnitude of noise by changing the parameter distribution of outliers. Specifically, we vary the upper bound of the uniform distribution and report the MAE and NLL. The settings follow the experiments reported in the main manuscript.
>
> | Method | Metric | **ub = 2**       | **ub = 5**       | **ub = 9**       | **ub = 15**      | ub = 20         |
> |--------|--------|------------------|------------------|------------------|------------------|-----------------|
> | RCaGP  | MAE    | 0.429 ± 0.051    | 0.430 ± 0.051    | 0.435 ± 0.052    | 0.440 ± 0.052    | 0.441 ± 0.056   |
> |        | NLL    | 1.257 ± 0.039    | 1.256 ± 0.042    | 1.253 ± 0.043    | 1.248 ± 0.040    | 1.244 ± 0.040   |
>
>
> The results indicate that as the magnitude of outliers increases, RCaGP experiences a gradual degradation in performance. Importantly, this degradation is relatively modest, demonstrating the method’s resilience to increasingly severe noise.
>
> Q3: In the BO experiments, while using 250 points as the initial sample size ensures a fair comparison, this number is relatively large. For instance, 20 points are often sufficient for a 6-dimensional problem. This large initial set might mask the differences in sample efficiency among the models during the early exploration phase?
>
> A3: BO outcomes results for the Hartmann6D function with 20 initializations are currently running and will be displayed soon.
>
> Q4 (Limitation): The paper's core mechanism for achieving robustness through the weight function $w(\mathbf{x},y)$ is critically dependent on a high-quality prior mean $m(\mathbf{x})$. Algorithm identifies outliers by measuring the deviation of an observation $y$ from $m(\mathbf{x})$. This means that if $m(\mathbf{x})$ is poorly specified, the model may erroneously down-weight normal data points or fail to identify true outliers, leading to a degradation in the performance of the entire robustness framework.
>
> A4: In our work, we considered two formulations for $m$:
>
> - A constant mean with a learnable constant.
>
> - An expert-driven prior mean, detailed in Section 5 and Appendix G.
>
> In Table 1, Figure 3, Figure 4 (left), Table S1, Table S2, Figure S1, Figure S2, and Figure S3, we use only the constant mean for the RCSVGP and RCaGP baselines. These are labeled either simply as “RCSVGP” and “RCaGP” (Table 1, Figure 3, Figure S1–S3), or as “RCSVGP (const.)” and “RCaGP (const.)” (Table S1 and S2). As shown in Sections 7.1 and 7.2, our proposed method RCaGP (const.) yields the best overall performance, even with a constant mean.
>
> The expert-driven prior mean is evaluated only in a dedicated ablation study (Section 7.3, paragraph two), with results shown in the right panel of Figure 4. There, the baselines are labeled “RCSVGP (exp.)” and “RCaGP (exp.)”. These results show that the expert-driven prior can improve performance in BO. However, this experiment assumes perfect expert correction of all outliers, which is unrealistic in practice. This ablation study should therefore be seen as a proof of concept. In future work, we aim to extend this strategy to scenarios with partial and noisy expert supervision.
>
> __We apologize for any confusion caused by inconsistent labeling and will revise the manuscript for clarity.__
>
> That said, we agree that using a constant prior mean can lead to degraded results, especially when the target function lies far from the prior. This limitation was discussed in [1], where the authors propose replacing the constant mean with the posterior predictive mean obtained via Generalized Bayes and filtering. Figure 4 in [1] illustrates the significant gains from this modification. We believe that integrating this idea into RCaGP is a promising and natural direction for future improvement.
>
> [1] Laplante et al. Robust and Conjugate Spatio-Temporal Gaussian Processes.
>
> Q5 (4 in Weaknesses): In the "Related Work" section, the paper mentions the first category of robust GP methods, which replace the Gaussian noise assumption with heavy-tailed distributions. Although the authors note that these methods often break conjugacy and require approximate inference, a direct comparison of RCaGP with a representative model from this category (e.g., an SVGP based on a Student-t likelihood) would more forcefully demonstrate whether RCaGP, while maintaining its advantage of conjugacy, still leads in terms of performance
>
> A5: We implemented another robust baseline: SVGP with a Student-t likelihood, using Laplace approximation for inference.
> The results for the UCI regression datasets with asymmetric outliers and without outliers are displayed in a table below. These can be compared to the results provided in the main text (Table 1). In neither the corrupted nor non-corrupted setting is the Student-t baseline competitive with the other methods.
>
> | Method     | Metric       | Boston (A)         | Energy (A)         | Yacht (A)          | Parkinsons (A)     | Boston (N)         | Energy (N)         | Yacht (N)          | Parkinsons (N)     |
> |------------|--------------|--------------------|--------------------|--------------------|--------------------|--------------------|--------------------|--------------------|--------------------|
> | Student-t  | MAE          | 0.962 ± 0.037      | 1.015 ± 0.028      | 0.907 ± 0.061      | 1.025 ± 0.079      | 0.896 ± 0.041      | 0.970 ± 0.025      | 0.815 ± 0.071      | 0.965 ± 0.074      |
> |            | NLL          | 1.619 ± 0.039      | 1.606 ± 0.024      | 1.564 ± 0.067      | 1.658 ± 0.085      | 1.593 ± 0.063      | 1.572 ± 0.026      | 1.559 ± 0.100      | 1.627 ± 0.092      |
> |            | clock-time   | 1.210 ± 0.240      | 1.800 ± 0.400      | 1.350 ± 0.260      | 0.990 ± 0.090      | 1.330 ± 0.050      | 1.330 ± 0.080      | 0.890 ± 0.150      | 0.730 ± 0.020      |
>
>
>
> Q6 (1 in Weaknesses):  Parameters such as the soft threshold $c$, the low-rank approximation dimension, and the size of the approximate action matrix require manual setting or cross-validation by the user, which could otherwise affect the balance between robustness and approximation quality.
>
> A6: We answered the question regarding hyperparameters $c$ and $\beta$ in Q2 to reviewer jPaV.
> Regarding the alternative hyperparameters of the method, such as the projection dimension, we conducted an additional ablation study to assess the impact of these hyperparameters on our results. We run the experiment on the Yacht dataset using the same settings as the UCI experiment, except now we vary the projection dimensionality for the CaGP and RCaGP models.
>
> | Method | Metric     | **d = 2**       | **d = 5**       | **d = 10**      | **d = 15**      | d = 20         | d = 25         |
> |--------|------------|------------------|------------------|------------------|------------------|------------------|------------------|
> | CaGP   | MAE        | 0.761 ± 0.089    | 0.775 ± 0.078    | 0.785 ± 0.082    | 0.795 ± 0.079    | 0.803 ± 0.080    | 0.815 ± 0.081    |
> |        | NLL        | 1.398 ± 0.073    | 1.395 ± 0.064    | 1.395 ± 0.062    | 1.398 ± 0.060    | 1.401 ± 0.059    | 1.409 ± 0.061    |
> |        | clock-time | 1.210 ± 0.240    | 0.720 ± 0.040    | 1.250 ± 0.110    | 1.430 ± 0.110    | 0.950 ± 0.180    | 1.070 ± 0.030    |
> | RCaGP  | MAE        | 0.426 ± 0.051    | 0.435 ± 0.052    | 0.438 ± 0.059    | 0.443 ± 0.055    | 0.436 ± 0.053    | 0.433 ± 0.047    |
> |        | NLL        | 1.246 ± 0.040    | 1.253 ± 0.043    | 1.250 ± 0.046    | 1.249 ± 0.046    | 1.246 ± 0.044    | 1.247 ± 0.040    |
> |        | clock-time | 1.700 ± 1.610    | 0.870 ± 0.060    | 1.190 ± 0.180    | 1.170 ± 0.060    | 1.230 ± 0.180    | 1.130 ± 0.180    |
>
> From the table, we observe that increasing the projection dimensionality does not necessarily reduce the MAE and NLL of RCaGP. However, CaGP gets higher MAE and NLL as we increase the projection dimensionality. This suggests that RCaGP is more robust to changes in projection dimension compared to CaGP.

---

> > ### Comment · Reviewer_8Bte · 2025-08-01
> > **Thank you for your detailed reply and additional experiments.**
> >
> > My concerns have been addressed. I have no more questions about this rebuttal.

---

### Official Review · Reviewer_jPaV · 2025-06-18

**Clarity:** 4
**Significance:** 2
**Originality:** 3
**Rating:** 4
**Confidence:** 4

**Summary:**

This paper proposes a generalized Gaussian process (GP) model that jointly addresses two practical limitations of GPs in real-world settings: robustness to outliers and computational tractability. The approach leverages generalized Bayesian inference with a robust loss to downweight corrupted observations, and incorporates uncertainty stemming from low-rank approximations to the posterior covariance (termed computational awareness). A prior over potentially corrupted data points is introduced. Hyperparameters are optimised using a generalised evidence lower bound. The method is supported by theoretically results on robustness and conservative uncertainty estimates. Experiments are conducted on four UCI regression tasks and four Bayesian optimization (BO) tasks under various data corruption scenarios.

**Questions:**

+ **Prior over corrupted observations:** The method relies on a prior over which points are corrupted. Can the authors clarify how this would be constructed in practical scenarios? What are real-world applications where such a prior is meaningful or available?
+ **Choice of weight function parameters:** The method depends on the parameters $\beta$ and $c$ in the weighting function $w(x,y)$. How sensitive is the performance to these values, and how should practitioners choose them?
+ **Robustness definition:** The robustness guarantee in line 192 appears weak, as it allows for any corruption inducing a non-infinite KL divergence. Can the authors clarify the intuition and implications of this result, and compare it to bounded worst-case divergence?
+ **Experimental realism:** The experimental setup, especially for BO, appears somewhat artificial. Can the authors include additional BO experiments without synthetic corruption and on higher-dimensional or larger-scale tasks where computational limitations are more pressing?
+ **Statistical significance:** It would improve interpretability if Table 1 clearly indicated which methods are not significantly worse than the best. A paired sample test could help here.
+ **Figure 1:** The predictive mean of RCaGP appears to poorly fit data at high input values. Can the authors explain this behavior? Is it due to initialization, optimization, or hyperparameters?
+ **GP hyperparameters for BO:** How were the GP lengthscales initialized for BO? This is particularly important in high-dimensional tasks like Rover and DNA [1]
+ **Additional BO results:** Could the authors report results for BO without corruption across all tasks, not just Hartmann and Lunar?

[1] Vanilla Bayesian Optimization Performs Great in High Dimensions, Hvarfner et al. 2024

**Ethical Concerns:**

["NO or VERY MINOR ethics concerns only"]

**Final Justification:**

I thank the authors for actively participating in the discussions.

**Main criticism:**
* *Limited experimental setup*: The paper only contains two experiments (regression on 4 UCI datasets and BO on 4 tasks). While the method performs well in these cases, the experiments are rather small-scale (768 samples for UCI and 10,000 for BO) and the corruptions are mostly synthetic, which does not make a strong case for the added computational uncertainty and robustness the method proposes. While the authors made a substantial effort to provide additional results to address this assessment, I believe that the empirical evaluation remains too weak.
* *Sensitivity to hyperparameters:* The method seems very sensitive to hyperparameter choices, which can be problematic in practice. This needs to be more thoroughly addressed.

**Score:** In light of the above remarks, I maintain my score of 4.

**Recommendations:** I recommend the authors extend the empirical evaluation to considerably larger datasets (such as the power dataset in Wenger et al. [45]) where computational uncertainty is more of an issue, and include cases where misspecification on real-world data is more problematic to demonstrate the usefulness of the additional robustness.

**Limitations:**

The authors acknowledge the need for a broader evaluation campaign in the conclusion. I see no obvious societal impact of this work that would have been missed by the authors.

**Paper Formatting Concerns:**

+ **Robust loss in main text:** Please include the robust loss from equation (S2) directly in the main text to aid accessibility.

**Quality:**

3

**Strengths And Weaknesses:**

### **Quality**

The paper is of high technical quality. It builds upon and synthesizes recent ideas from the GP literature on robust inference and computational uncertainty aware approximations. The methodology is sound, and the theoretical contributions are well presented. I expect this work to be of interest to the GP community.

+ **Strengths:** Sound methodology and interesting integration of existing techniques.
+ **Weaknesses:** Experimental evaluation is limited in both scope (only 4 UCI datatsets and 4 BO tasks) and realism (artifical corruption of real-world data). The proposed model is evaluated only under synthetic corruption regimes, raising concerns about the practical relevance of the results. In the BO experiments in particular, the small data regime and low input dimensionality seem insufficient to highlight the method’s strengths in realistic computational or model mismatch scenarios.

### **Clarity**

The paper is clearly written, well organized, and enjoyable to read. Background and related work are well covered, and the key contributions are clearly articulated. Figures and tables are generally helpful, although there are a few missing items that would improve clarity (e.g., what "std" refers to in Table 1, explicit definition of the robust loss in the main text).

### **Significance**

The paper tackles an important challenge by aiming to make Gaussian processes more robust to outliers and more reliable under computational constraints. These are meaningful goals, especially for real-world applications where data quality and compute resources are often limiting factors. However, I have concerns regarding the practical applicability of the proposed approach in its current form.

Methodologically, the reliance on a prior over corrupted observations assumes some knowledge about which datapoints are likely to be outliers. In many practical scenarios (especially in Bayesian optimization) such prior knowledge may be unavailable or difficult to specify. A clearer discussion of real-world use cases where this modeling assumption is justified would help assess the method’s relevance. Similarly, the need to set additional hyperparameters in the weight function (e.g., $\beta$, $c$) raises concerns: how sensitive is the method to these choices in practice and how would they be selected?

Experimentally, the current evaluation setup appears somewhat artificial. The benefits of the method are demonstrated mainly through controlled corruptions of existing datasets. While the method indeed performs well under these scenarios, such setups are closely aligned with the assumptions it was designed around, making the improvements less surprising. In contrast, the paper does not convincingly demonstrate performance gains in settings that more closely reflect real-world noise or model mismatch. The BO tasks used are relatively high-dimensional and involve small datasets ($\leq 10250$ samples), where misspecification and computational approximations of the posterior are unlikely to be a limiting factor. This undermines the motivation for computational awareness in this context.

Ultimately, the method may indeed offer substantial benefits in the presence of real-world corruptions or large-scale settings where exact inference is infeasible. However, the current experimental design does not convincingly establish such relevance. A stronger empirical case would require identifying or constructing realistic scenarios where the proposed robustness and approximation-aware inference offer clear advantages over standard GPs.

Additionally, reporting statistical significance (e.g., by bolding methods that are not significantly worse than the top performer in Table 1) would help provide a more nuanced view of the method's effectiveness.

### **Originality**

This paper offers a combination of existing ideas that have not yet been jointly explored. The integration of outlier-aware inference with posterior approximation-aware uncertainty modeling is elegant and fills a gap in the GP literature. The contribution is more incremental than novel, but still worthwhile.

---

> ### Author Rebuttal · Authors · 2025-07-31
>
> Thank you for your detailed and thoughtful review. We appreciate your recognition of the paper’s technical quality, clear presentation, and the relevance of our contributions to robust and computationally aware Gaussian Process modeling.
>
> The ``1 std'' reported in Table 1 refers to the standard deviation computed for the metrics across 20 train-test splits. The robust loss equation is presented in Equation S2. We will bring this equation to the main text from the Appendix.
>
> Q1: The method relies on a prior over which points are corrupted. Can the authors clarify how this would be constructed in practical scenarios? What are real-world applications where such a prior is meaningful or available?
>
> A1: Thank you for raising this important point. To clarify, the method does not assume a prior distribution over which data points are corrupted. Instead, it relies on a ``centering function'' $m(\mathbf{x})$, and data points with large deviations from this function—i.e., where $\lvert y - m(\mathbf{x})\rvert$ is high—are downweighted by the model. We considered two choices for $m(\mathbf{x})$: the GP prior mean, which is a constant mean with a learnable scalar, and an expert-driven prior mean, which is introduced in Section 5 and Appendix G. The latter assumes that a domain expert can flag potential outliers and provide corrections. These corrected points are then used to learn a function $m(\mathbf{x})$, which informs the weight function $w(\mathbf{x}, y)$. In this context, the notion of a “prior over corrupted observations” is appropriate, though we emphasize that such an expert-driven mean is not required for the method to be applicable. We believe this setup is relevant in settings where expert annotation is available or can be integrated into the modeling process—such as in clinical data analysis or fault detection in manufacturing. However, as demonstrated in our experiments, the method performs robustly even when using simpler, data-agnostic choices for $m(\mathbf{x})$, such as a constant or GP prior mean.
> We will revise the manuscript to include this discussion clarifying the role and interpretation of $m(\mathbf{x})$, and to better distinguish between optional expert involvement and baseline usage.
>
> Q2: The method depends on the parameters $\beta$ and $c$ in the weighting function $w(\mathbf{x},y)$. How sensitive is the performance to these values, and how should practitioners choose them?
>
> A2: In the first paragraph of Section 7.3, we conducted an ablation study on the UCI regression datasets to assess sensitivity to $c$. To recap the conclusions written in the manuscript:
>
> >We evaluate RCaGP's sensitivity to the threshold $c = Q_n(\vert \mathbf{y} - \mathbf{m} \vert, 1 - \epsilon)$, defined as the $(1 - \epsilon)$-quantile of $\lvert \mathbf{y} - \mathbf{m} \rvert$ and controlling how many values are treated as outliers, by varying $\epsilon$. Figure 4 (left) shows MAE and NLL averaged over 4 UCI datasets (normalized; per-dataset results in Figure S3. RCaGP consistently achieves lower error across all $\epsilon$, with best results at $0.15$, suggesting treating roughly 15\% of the data as potential outliers is optimal. This aligns well with the naturally present and 10\% injected asymmetric outliers.
>
> These results suggest that $c$ can be selected based on the expected proportion of outliers.
>
> We set $\beta=1$. An ablation study to explore the impact of varying $\beta$ on the yacht dataset with asymmetric outliers:
>
> | Method | Metric | **β = 0.1**       | **β = 0.5**       | **β = 1.0**       | **β = 1.5**       | β = 2.0         |
> |--------|--------|------------------|------------------|------------------|------------------|----------------|
> | RCaGP  | MAE    | 0.768 ± 0.075    | 0.421 ± 0.072    | 0.435 ± 0.052    | 0.472 ± 0.039    | 0.498 ± 0.035  |
> |        | NLL    | 1.439 ± 0.061    | 1.294 ± 0.055    | 1.253 ± 0.043    | 1.247 ± 0.031    | 1.250 ± 0.026  |
>
> Both MAE and NLL decrease rapidly as $\beta$ increases up to 0.5, indicating a significant performance improvement. After, the performance begins to degrade, but the increase in MAE and NLL is more gradual.
>
> __We will explicitly acknowledge hyperparameter tuning as a current limitation of the method.__
>
> Q3: The robustness guarantee in line 192 appears weak, as it allows for any corruption inducing a non-infinite KL divergence. Can the authors clarify the intuition and implications of this result, and compare it to bounded worst-case divergence?
>
> A3: Our theoretical result guarantees that the posterior under corrupted data remains close to the clean-data posterior whenever the corruption leads to a finite KL divergence. While this does not assume a worst-case, norm-bounded corruption, it encompasses a broad and realistic range of outliers that are not well captured by strict norm constraints.
>
> In contrast to worst-case robustness guarantees—which often rely on conservative bounds under adversarial assumptions—our result shows that as the KL divergence of the corruption decreases (regardless of whether it is norm-bounded), the posterior transitions smoothly toward the clean posterior. This provides a more flexible and practically relevant information-theoretic notion of robustness. We will add a concise summary of these points as a discussion.
>
> Q4: The experimental setup, especially for BO, appears somewhat artificial. Can the authors include additional BO experiments without synthetic corruption and on higher-dimensional or larger-scale tasks where computational limitations are more pressing?
>
> A4: We would like to clarify that the BO tasks considered in our study already fall within the high-dimensional regime. Specifically, the 60D Rover and 180D Lasso DNA benchmarks are commonly used in the evaluation of high-dimensional Bayesian optimization methods. While high dimensionality was not the central focus of our work, we believe these tasks are representative of realistic and computationally demanding BO scenarios. Regarding the scale of the experiments, we interpret "larger-scale" as referring to a greater number of optimization steps or oracle evaluations. Extending the experiments in this direction is indeed valuable, but unfortunately not feasible within the constraints of the rebuttal period. That said, we agree this is an important direction for future work and will note it as such in the revised manuscript.
> Concerning the realism of the experimental setup: we acknowledge that the use of synthetic corruption introduces a controlled setting that may differ from real-world noise patterns. However, our goal was to isolate and evaluate the robustness properties of the proposed model under well-defined, repeatable conditions. We agree that incorporating naturally noisy or real-world BO tasks would strengthen the empirical evaluation, and we will make this limitation explicit in the manuscript.
>
> Q5: It would improve interpretability if Table 1 clearly indicated which methods are not significantly worse than the best. A paired sample test could help here.
>
> A5: Thank you for this suggestion. __We will add the results of a paired sample test to the manuscript.__
>
> Q6: The predictive mean of RCaGP appears to poorly fit data at high input values. Can the authors explain this behavior? Is it due to initialization, optimization, or hyperparameters?
>
> A6: It is due to the choice of the mean function. We set the mean function relatively far below the average outliers. In a low-input region, a sufficient amount of data overrides the penalty, resulting in a fitted curve. On the other hand, the high-input region is scarce in data, allowing the mean to act more dominantly. RCSVGP exhibits a similar trend, but more pronounced—likely due to its reliance on variational approximations, which are more sensitive to prior assumptions in data-scarce regions. In contrast, the computational uncertainty in RCaGP appears to buffer the impact of the mean prior, preventing it from dominating excessively in those regions.
>
> Q7: How were the GP lengthscales initialized for BO? This is particularly important in high-dimensional tasks like Rover and DNA (Vanilla Bayesian Optimization Performs Great in High Dimensions, Hvarfner et al. 2024)
>
> A7: In our experiments, we used the default lengthscale initialization from GPyTorch, which sets a common value of approximately $\ell \approx 0.69$. The lengthscale is the same across all dimensions. While BoTorch includes strategies for dimension-aware initialization—as discussed in Hvarfner et al. (2024)—these are not part of GPyTorch’s default behavior, and we did not explicitly implement them in our setup.
> We agree that evaluating the impact of dimension-aware lengthscale initialization in the context of high-dimensional tasks such as Rover and Lasso DNA would be a valuable addition. However, given the large number of experiments already being run and the time constraints of the rebuttal period, we may not be able to include this analysis at this stage. We will note this point in the revised manuscript as a direction for future work.
>
> Q8: Could the authors report results for BO without corruption across all tasks, not just Hartmann and Lunar?
>
> A8: We provide the results for BO without corruption on the Rover 60D experiment. Lower is better.
>
> | Method   | T = 50         | **T = 100**     | **T = 200**     | **d = 500**     | **T = 1000**     | T = 2000       |
> |----------|----------------|----------------|----------------|----------------|------------------|----------------|
> | CaGP     | -14.54         | -14.24         | -11.20         | -8.95          | -1.22            | -1.04          |
> | RCaGP    | -4.95          | -4.65          | -4.65          | -4.57          | -3.45            | -1.55          |
> | SVGP     | -6.88          | -5.53          | -4.27          | -2.90          | -2.03            | 0.05           |
> | RCSVGP   | -15.21         | -8.69          | -5.14          | -2.86          | -1.37            | 1.1146         |

---

> ### Comment · Reviewer_jPaV · 2025-08-01
>
> Thank you for your answers to my questions and for all the clarifications.
>
> > "A4: [...] Concerning the realism of the experimental setup: we acknowledge that the use of synthetic corruption introduces a controlled setting that may differ from real-world noise patterns. However, our goal was to isolate and evaluate the robustness properties of the proposed model under well-defined, repeatable conditions. We agree that incorporating naturally noisy or real-world BO tasks would strengthen the empirical evaluation, and we will make this limitation explicit in the manuscript."
>
> While I understand the value of controlled experiments with synthetic corruptions, the absence of evaluation on real-world data with naturally occurring misspecification and outliers remains a critical limitation of the paper in its current form. Real-world robustness is essential for demonstrating the practical utility of the proposed method. Simply acknowledging this limitation in the manuscript is insufficient to establish the method's effectiveness in realistic scenarios where such robustness properties are most needed.
>
> > "A5: Thank you for this suggestion. **We will add the results of a paired sample test to the manuscript.**"
>
> To facilitate an informed assessment during the discussion period, could the authors provide the paired sample test results? This would enable to evaluate the statistical significance of the reported improvements and potentially update my review accordingly.
>
> > "A7: In our experiments, we used the default lengthscale initialization from GPyTorch, which sets a common value of approximately $\ell \approx 0.69$. The lengthscale is the same across all dimensions. While BoTorch includes strategies for dimension-aware initialization—as discussed in Hvarfner et al. (2024)—these are not part of GPyTorch’s default behavior, and we did not explicitly implement them in our setup. We agree that evaluating the impact of dimension-aware lengthscale initialization in the context of high-dimensional tasks such as Rover and Lasso DNA would be a valuable addition. However, given the large number of experiments already being run and the time constraints of the rebuttal period, we may not be able to include this analysis at this stage. We will note this point in the revised manuscript as a direction for future work."
>
> Thank you for the clarification. However, this reveals an experimental design issue that significantly undermines the validity of your high-dimensional results (Rover and DNA tasks). Recent literature [1,2,3] has consistently demonstrated that careful lengthscale initialization is critical for Bayesian optimization performance in high-dimensional spaces. The default GPyTorch initialization is inappropriate for high-dimensional problems because it fails to account for the curse of dimensionality, specifically the increased Euclidean distances that render stationary covariance functions ineffective at capturing meaningful correlations.
>
> In high-dimensional discrete spaces, this improper initialization typically reduces the Gaussian process surrogate to essentially random point selection, as the model cannot extrapolate meaningfully to nearby points. Given that the Rover and DNA tasks operate in 60D and 180D spaces respectively, the reported results on these benchmarks are likely unrepresentative of the true performance of the compared methods and may be dominated by this initialization artifact rather than reflecting the actual methodological differences.
>
> This is not merely a minor technical detail but a core experimental validity issue that calls into question the conclusions drawn from the high-dimensional experiments. I strongly recommend either re-running these experiments with appropriate dimension-aware lengthscale initialization (for example initializing as per [2]) or removing the high-dimensional results entirely, as they currently provide little insight into the relative merits of the proposed approach.
>
> [1] Vanilla Bayesian Optimization Performs Great in High Dimensions, Hvarfner et al. 2024.
> [2] Standard Gaussian Process is All You Need for High-Dimensional Bayesian Optimization, Xu et al. 2025.
> [3] Understanding High-Dimensional Bayesian Optimization, Papenmeier et al. 2025

---

> ### Author Response · Authors · 2025-08-02
>
> Thank you for this extensive answer.
>
> We agree on the absolute necessity to run Bayesian Optimization experiments again with a correct lengthscale initialization strategy, specifically for high-dimensional examples like Rover 60D or Lasso DNA 180D. These experiments are running at present, using the initialization proposed by Hvarfner et al. Results will be posted as soon as possible. This being said, it is worth noticing that the recent works you shared all deal with exact GPs and not custom models like SVGPs. While we expect these findings to largely transfer to approximate GP models such as SVGPs, some differences may arise due to the structural and inference-specific distinctions between exact and sparse variational GPs. Furthermore, the existing works on high-dimensional BO do not consider the presence of outliers. Finally, we consider a non-trivial EULBO acquisition function, where we need to simultaneously optimize the query, variational parameters, and the hyperparameters, which may lead to different behaviors.
>
> Next, we present in the table below the statistical significance between methods. We used Wilcoxon signed-rank test, computed across 20 train/test split folds. This table is associated with the left part of Table 1, on UCI regression datasets with asymmetric outliers.
>
> | Comparison         | Metric | Boston     | Energy     | Yacht      | Parkinsons |
> |--------------------|--------|------------|------------|------------|------------|
> | RCaGP vs CaGP      | MAE    | 9.54e-7    | 9.54e-7    | 9.54e-7    | 9.54e-7    |
> |                    | NLL    | 9.54e-7    | 9.54e-7    | 9.54e-7    | 9.54e-7    |
> | RCaGP vs RCSVGP    | MAE    | 9.54e-7    | 9.54e-7    | 0.00035    | 9.54e-7    |
> |                    | NLL    | 9.54e-7    | 9.54e-7    | 0.97577    | 9.54e-7    |
> | RCaGP vs SVGP      | MAE    | 9.54e-7    | 9.54e-7    | 9.54e-7    | 9.54e-7    |
> |                    | NLL    | 9.54e-7    | 9.54e-7    | 9.54e-7    | 9.54e-7    |
>
>
> Finally, experiments on real-world data will also follow, as soon as we can.

---

> > ### Comment · Reviewer_jPaV · 2025-08-04
> >
> > Thank you for the paired sample test, I am looking forward to see the new results. Could the authors provide the p-values for the test rather than the test statistics? This makes assessing the significance of the results easier.

---

> > > ### Author Response · Authors · 2025-08-05
> > > **Results for real-world experiment**
> > >
> > > Actually, the values reported in the Table above correspond to p-values and not the test statistics.
> > >
> > > __Real-world experiment.__ We now present results from a real-world case study: the Twitter Flash Crash task, originally introduced by Altamiro et al. [1]. This task focuses on modeling the behavior of the Dow Jones Industrial Average (DJIA) on April 17, 2013—a day when U.S. stock markets experienced a sudden, sharp decline followed by an equally rapid recovery. The event was triggered by a false tweet from the hacked Twitter account of the Associated Press, which falsely reported an explosion at the White House. We faithfully replicate the experimental setup of Altamirano et al. and adopt the same Gaussian Process (GP) model configurations as in our UCI benchmark experiments.
> > >
> > > The (unnormalized) mean and std of the data are 14698.4 and 15.84, respectively. The two abovementioned outliers have values 14599.5 and 14653. The table below reports the results of the different methods on the normalized data.
> > >
> > > | Method   | MAE   | NLL   | Weight function mean | Weight function at outliers |
> > > |----------|-------|--------|----------------------|-----------------------------|
> > > | RCaGP    | 0.320 | 0.935  | 0.770                | [0.242, 0.121]              |
> > > | CaGP     | 0.378 | 0.967  |                      |                             |
> > > | SVGP     | 0.492 | 1.074  |                      |                             |
> > > | RCSVGP   | 1.784 | 4.110  | 0.775                | [0.256, 0.125]              |
> > >
> > > As we can see, RCaGP provides the best MAE and NLL, although the difference with CaGP is thin. SVGP and RCSVGP perform poorly. Following Altamiro et al., we did not use multiple train/test folds here, so that the statistical significance of the MAE / NLL difference between baselines cannot be assessed here.
> > >
> > > Both RCaGP and RCSVGP incorporate a weight function, which provides insight into whether outliers were correctly identified and downweighted. In this context, successful identification corresponds to a low value of the weight function $w$ at outlier locations. As shown by the comparison between the mean weight and the weight values at outliers, both methods effectively downweight the outliers, with RCaGP exhibiting slightly more pronounced downweighting than RCSVGP.
> > > It is worth noticing that even if $c$ and $\beta$ in $w(x,y)$ (Eq. 12 in main text) are the same for RCaGP and RCSVGP, their centering function $m$ is a learnable constant, whose value is optimized as part of the ELBO criterion (Eq. 13). For this criterion, the optimization space for RCaGP and RCSVGP slightly differs (e.g., in RCaGP, we also optimize for the action matrix $S$), leading to different $m$, and thus, slightly different functions $w$.
> > >
> > > That said, while this real-world case remains informative, it is not ideally suited to highlight the strengths of our method, as the relatively small dataset size does not require approximate inference. However, given the limited time available and the need to prioritize more critical experiments, we still believe this additional evaluation provides relevant complementary insight.
> > >
> > > [1] Altamiro et al. Robust and Conjugate Gaussian Process Regression.

---

> > > > ### Comment · Reviewer_jPaV · 2025-08-05
> > > >
> > > > Thank you for the extra explanation and experiment.
> > > >
> > > > **Statistical significance:** These p-values look good and highlight the improved performance of your method on all configurations but for RCaGP vs RCSVGP on Yacht.
> > > >
> > > > **New experiment:** It is reassuring to see that your method also performs strongly on this real-world dataset. If the authors also have time during the rebuttal, I would be interested to see cross-validated results.

---

> > > > > ### Author Response · Authors · 2025-08-06
> > > > > **Additional results for BO with Asymmetric outliers and dimension-aware lengthscale initialization**
> > > > >
> > > > > As requested, we conducted additional experiments comparing our method to other baselines in a Bayesian optimization setting, using the **dimension-aware lengthscale initialization** introduced by Hvarfner et al. \[1]. In that work, appropriately initializing the lengthscale was shown to improve BO performance when using standard Gaussian Processes.
> > > > >
> > > > > Specifically, we follow their Equation (4) and set the initial lengthscale to
> > > > > $\ell = \mu_0 + \log\left(\frac{D}{2}\right)$,
> > > > > with $\mu_0 = \sqrt{2}$, as used in their experiments.
> > > > >
> > > > > For the experiments reported below, we follow the exact same settings as described in our main text, section 7.2 and figure 3 (the four subplots on the top left): BO with asymmetric outliers, using EULBO EI as an optimization criterion, 20 BO trials with different seeds.
> > > > >
> > > > > The results presented below are for the __Rover60D experiment__ (to be compared to those shown in Figure 3, top row, third subplot). The most relevant example would be DNA180D, but due to some cluster bottleneck, the computations crashed, and we had to launch them again. We are expecting to have them soon, before the deadline.
> > > > >
> > > > > | Method   | T = 100         | T = 5000        | T = 7500        | T = 10000       |
> > > > > |----------|------------------|------------------|------------------|------------------|
> > > > > | RCaGP    | -17.24 ± 4.96    | -1.76 ± 2.56     | -1.76 ± 2.72     | -1.64 ± 2.28     |
> > > > > | RCSVGP   | -18.09 ± 5.63    | -1.21 ± 0.44     | -1.21 ± 0.44     | -1.21 ± 0.44     |
> > > > > | CaGP     | -18.37 ± 6.09    | -4.29 ± 2.52     | -4.07 ± 2.44     | -3.87 ± 2.43     |
> > > > > | SVGP     | -18.36 ± 6.09    | -0.05 ± 2.68     |  0.35 ± 2.53     |  0.70 ± 2.49     |
> > > > >
> > > > > While RCaGP is not the top performer in this experiment, the results are broadly consistent with those reported in the main text using the default lengthscale initialization of $\ell = 0.69$
> > > > > CaGP remains clearly behind, while differences between RCaGP, RCSVGP, and SVGP are subtle. In particular, the overlapping standard deviations indicate that performance differences among these three models are not statistically significant. We therefore conclude that the results for the Rover60D task remain qualitatively similar under dimension-aware initialization.
> > > > >
> > > > > This being said, for SVGP, the average best value found at $T=10000$
> > > > > increases to $0.70$, which is clearly higher than before—suggesting that this initialization strategy may be beneficial for SVGP. For RCaGP and RCSVGP, the effect is less clear. For CaGP, performance actually worsens, indicating that this initialization may not generalize well across models.
> > > > >
> > > > > These observations may be put into perspective by the fact that, as noted earlier, existing work on dimension-aware lengthscale initialization has focused primarily on exact Gaussian Processes, rather than on approximate or custom models such as SVGP, CaGP, RCaGP, and RCSVGP. More importantly, in our BO setup, lengthscales are optimized using the EULBO objective (as described in Section 3.3 and Equation 14 of the main text), and not through MAP estimation as in [1]. This key difference in training objectives may limit the direct transferability of findings from prior work.

---

> > > > > > ### Comment · Reviewer_jPaV · 2025-08-08
> > > > > >
> > > > > > Thanks for providing those results for the Rover BO task; I agree with your assessment. I know it's a lot to ask for more experiments during this discussion period, but it would be really helpful if you could also run the DNA BO task.

---

> > > > > > > ### Author Response · Authors · 2025-08-08
> > > > > > >
> > > > > > > Given the limited time available, we are unable to provide results for DNA and instead report the Hartmann 6D task using the lengthscale initialization proposed by Hvafner et al. (2024).
> > > > > > >
> > > > > > > | Method   | T = 100         | T = 1000        | T = 1500        | T = 2000       |
> > > > > > > |----------|------------------|------------------|------------------|------------------|
> > > > > > > |RCaGP | 2.41 ± 0.35 | 2.89 ± 0.20 | 3.10 ± 0.16  | 3.19 ± 0.15 |
> > > > > > >  |   RCSVGP | 2.40 ± 0.35 | 2.78  ± 0.26 | 2.88 ± 0.18 | 2.90  ± 0.17 |
> > > > > > >   |  CaGP | 2.41 ± 0.33 | 2.76 ± 0.18 | 2.76 ± 0.17 | 2.92 ± 0.14 |
> > > > > > >    | SVGP | 2.40 ± 0.35 | 2.83  ± 0.33  | 2.97  ± 0.20  | 3.11 ± 0.15 |
> > > > > > >
> > > > > > > From the table, we observe that RCaGP outperforms all baselines. While SVGP shows a significant performance boost when using the Hvafner lengthscale initialization, the other baselines do not exhibit similar improvements. This suggests that the benefits of this initialization scheme do not generalize across all approximate GP methods.

---

### Official Review · Reviewer_uEzz · 2025-06-21

**Clarity:** 4
**Significance:** 3
**Originality:** 3
**Rating:** 5
**Confidence:** 4

**Summary:**

The authors extend the computationally aware Gaussian process (CaGP) model by integrating concepts from robust and conjugate Gaussian processes (RCGP) [Altamirano, Briol & Knoblauch, 2024]. They begin by establishing the model with an RCGP prior and then apply CaGP's variational inference approach to the representer weights of the conjugate posterior.

In addition to this core contribution, the authors:

* Propose a prior mean function based on outlier detection from an expert (Section 5).
* Demonstrate that the impact of adding contaminated data into the posterior computation, specifically the posterior influence function (PIF), is bounded in terms of the contamination (Proposition 4.1, Appendix D), an approach analogous to Altamirano, Briol & Knoblauch (2024).
* Extend results from Wenger et al. (2024), based on Kanagawa et al. (2018), to directly link RCaGP's posterior variance with the supremum of the squared difference between the predictive centered mean and all $h(x)$ with an RKHS unit norm or less. This assumes the dataset is generated by a latent function $g:\\mathcal{X}\\to\\mathbb{R}$ and its centered version $h(x) = (g(x)-m\_w(x))$ belongs to the RKHS of the data generating process (Proposition 4.2, Appendix E).
* Empirically evaluate their model on four well-known small-scale ($n<1000$) UCI regression datasets and Bayesian optimization tasks, artificially introducing outliers.

**Questions:**

Could the authors clarify the definition of $m\_w(x)$ in Proposition 4.2? While the meaning of $m\_w(\\mathbf{X})$ is not explicitly clear, we can infer that it also depends on the vector $\\mathbf{y}=g(\\mathbf{X})$. However, for the assumption that $(g(x)-m\_w(x))$ belongs to the RKHS of the data-generating kernel, are we meant to assume that $m\_w(x)$ in this context signifies $m\_w(x,g(x))=m(x)-\\sigma^2\_{\\text{noise}}2\\frac{\\mathrm{d}}{\\mathrm{d}g(x)}\\log w(x,g(x))$?

The assumption that $(g(x)-m\_w(x))$ belongs to an RKHS is similar to results from Wenger et al. (2024) based on Kanagawa et al. (2018). This assumption appears straightforward to accept in the standard Gaussian Process Regression (GPR) case, as the GP posterior mean is typically a member of the RKHS when the mean function is considered. In the current context, the shrinkage term alters this interpretation, as it effectively influences what would otherwise be considered the prior mean of the GP. I believe the authors should clarify this assumption and discuss the conditions under which it might or might not hold, thereby providing better context for Proposition 4.2.

**Possible Typo:** In the inline equation on line 95 defining $\\mathbf{C}\_i$, the inverse should be placed outside the parenthesis, rather than being attached to $\\hat{\\mathbf{K}}$.

**Ethical Concerns:**

["NO or VERY MINOR ethics concerns only"]

**Final Justification:**

I am keeping my score because I believe the authors have adequately responded to the concerns raised by myself and the other reviewers, especially with the addition of the new baselines. I recommend that the authors take our discussion into account when explaining their theoretical results. This will improve their clarity for a wider audience without compromising the precision of what is being conveyed

**Limitations:**

Yes.

**Quality:**

4

**Strengths And Weaknesses:**

I find this paper to be well-rounded in its contributions, translating expected results from its main source papers into this new setting. While some theoretical results might appear to be straightforward extensions of previous work, the proposal for the expert-guided prior mean and the comprehensive experimental evaluation provide interesting and valuable additions to the use of Gaussian Processes for robust regression, which should satisfy most readers.

Nonetheless, while the authors compare their method to the original CaGP and RCGP approaches, I believe readers might struggle to contextualize this proposal in terms of uncertainty quality and runtime when compared against non-conjugate GP regression methods, such as those employing MCMC or Sparse Variational Gaussian Processes (SVGP).

---

> ### Author Rebuttal · Authors · 2025-07-31
>
> Thank you for your thoughtful review and for appreciating the overall contributions and experimental depth of our work on robust GP regression. Regarding the typo, thank you, we will update the manuscript accordingly.
>
>
> Q: Could the authors clarify the definition of $m_w(x)$ in Proposition 4.2? While the meaning of $m_w(\mathbf{X})$ is not explicitly clear, we can infer that it also depends on the vector $\mathbf{y} = g(\mathbf{X})$. However, for the assumption that $(g(x) - m_w(x))$ belongs to the RKHS of the data-generating kernel, are we meant to assume that $m_w(x)$ in this context signifies
>
> $m_w(x, g(x)) = m(x) - \sigma^2_{noise} 2 \frac{d}{dg(x)}\log w(x, g(x))$?
>
> The assumption that $(g(x) - m_w(x))$
>  belongs to an RKHS is similar to results from Wenger et al. (2024) based on Kanagawa et al. (2018). This assumption appears straightforward to accept in the standard Gaussian Process Regression (GPR) case, as the GP posterior mean is typically a member of the RKHS when the mean function is considered. In the current context, the shrinkage term alters this interpretation, as it effectively influences what would otherwise be considered the prior mean of the GP. I believe the authors should clarify this assumption and discuss the conditions under which it might or might not hold, thereby providing better context for Proposition 4.2.
>
> A: We thank the reviewer for this insightful comment. As noted, the difference $g(\mathbf{x}) - m_w(\mathbf{x})$ belongs to the RKHS $\mathcal{H}_k$ of the data-generating kernel if
> $$\nabla_y \log(w^2) \in \mathcal{H}_k$$
>
> This holds when $\nabla_y \log(w^2)$ has a finite RKHS norm; that is, $\| \nabla_y \log(w^2) \|_{\mathcal{H}_k}^2 < \infty$.
>
> The specific smoothness and regularity conditions required for membership in an RKHS depend on the choice of kernel. For instance, the squared exponential (RBF) kernel induces an RKHS of highly smooth functions, whereas Matérn kernels admit only a finite degree of differentiability. If $\nabla_y \log(w^2)$ is too irregular—e.g., non-smooth or not square-integrable under the kernel—it will not belong to $\mathcal{H}_k$.
>
> In the case of our robust weight function, the correction term is a nonlinear, rational function of $m(\mathbf{x})$. Generally, such functions do not lie in the RKHSs associated with common kernels (e.g., RBF, Matérn), unless specific and uncommon conditions are met—such as the RKHS being closed under composition with the given nonlinearity.
>
> A practical approach to ensure $m_w(\mathbf{x})$ belongs to the RKHS is to project the correction term onto the RKHS. Alternatively, one could use or design a kernel whose RKHS explicitly accommodates this class of nonlinear transformations, although this may come at the expense of flexibility or interpretability.
>
> __We will include a discussion of this condition in the revised manuscript.__
>
>
> --------
> For space concerns, we use the space available with the rebuttal answer to post experimental results requested in Q1 by reviewer 8Bte:
>
> Q1 8Bte: The effectiveness of the algorithm does not seem to be limited to the Matérn kernel function. It would be beneficial to include some discussion and experiments regarding other common GP kernel functions?
>
> A1 8Bte: We generated a table similar to Table 1 but, experimenting with the RBF kernel in place of the Matern kernel. The results demonstrate the overall superiority of RCaGP both in terms of MAE and NLL, as our proposed method is only outperformed by RCSVGP on 2 out of 8 cases: Boston and Parkinsons with asymmetric outliers.
>
> | Method     | Metric       | Boston (A)         | Energy (A)         | Yacht (A)          | Parkinsons (A)     | Boston (N)         | Energy (N)         | Yacht (N)          | Parkinsons (N)     |
> |------------|--------------|--------------------|--------------------|--------------------|--------------------|--------------------|--------------------|--------------------|--------------------|
> | SVGP       | MAE          | 1.145 ± 0.052      | 1.138 ± 0.040      | 1.235 ± 0.044      | 1.002 ± 0.100      | 0.731 ± 0.052      | 0.908 ± 0.026      | 0.767 ± 0.063      | 0.811 ± 0.078      |
> |            | NLL          | 1.691 ± 0.043      | 1.736 ± 0.037      | 1.732 ± 0.041      | 1.590 ± 0.079      | 1.433 ± 0.053      | 1.430 ± 0.023      | 1.420 ± 0.078      | 1.448 ± 0.070      |
> |            | clock-time   | 0.930 ± 0.280      | 1.170 ± 0.070      | 0.650 ± 0.060      | 0.720 ± 0.260      | 0.940 ± 0.030      | 1.230 ± 0.150      | 0.870 ± 0.240      | 0.670 ± 0.130      |
> | CaGP       | MAE          | 1.060 ± 0.095      | 0.971 ± 0.090      | 1.133 ± 0.092      | 0.955 ± 0.110      | 0.488 ± 0.042      | 0.334 ± 0.024      | 0.512 ± 0.042      | 0.674 ± 0.077      |
> |            | NLL          | 1.611 ± 0.086      | 1.535 ± 0.071      | 1.679 ± 0.088      | 1.556 ± 0.080      | 1.278 ± 0.040      | 1.106 ± 0.011      | 1.263 ± 0.042      | 1.362 ± 0.067      |
> |            | clock-time   | 2.470 ± 0.060      | 3.540 ± 0.080      | 0.640 ± 0.100      | 0.870 ± 0.080      | 2.410 ± 0.060      | 3.540 ± 0.160      | 0.770 ± 0.060      | 0.830 ± 0.100      |
> | RCSVGP     | MAE          | 0.680 ± 0.122      | 0.456 ± 0.041      | 0.733 ± 0.114      | 0.840 ± 0.187      | 0.512 ± 0.074      | 0.558 ± 0.035      | 0.640 ± 0.074      | 0.540 ± 0.083      |
> |            | NLL          | 1.364 ± 0.080      | 1.200 ± 0.026      | 1.381 ± 0.084      | 1.518 ± 0.116      | 1.302 ± 0.063      | 1.300 ± 0.034      | 1.318 ± 0.047      | 1.417 ± 0.065      |
> |            | clock-time   | 1.500 ± 0.090      | 2.560 ± 0.100      | 0.970 ± 0.090      | 0.970 ± 0.090      | 1.530 ± 0.040      | 2.780 ± 0.120      | 1.130 ± 0.080      | 0.720 ± 0.060      |
> | RCaGP      | MAE          | 0.733 ± 0.156      | 0.353 ± 0.053      | 0.651 ± 0.095      | 0.913 ± 0.198      | 0.379 ± 0.050      | 0.326 ± 0.023      | 0.365 ± 0.035      | 0.534 ± 0.078      |
> |            | NLL          | 1.405 ± 0.118      | 1.173 ± 0.024      | 1.336 ± 0.061      | 1.531 ± 0.152      | 1.241 ± 0.046      | 1.144 ± 0.007      | 1.226 ± 0.034      | 1.307 ± 0.068      |
> |            | clock-time   | 2.550 ± 0.060      | 3.590 ± 0.090      | 0.790 ± 0.060      | 0.900 ± 0.030      | 2.470 ± 0.030      | 3.740 ± 0.170      | 0.940 ± 0.070      | 1.070 ± 0.240      |
> | Student-t  | MAE          | 0.962 ± 0.037      | 1.015 ± 0.028      | 0.907 ± 0.061      | 1.025 ± 0.079      | 0.896 ± 0.041      | 0.970 ± 0.025      | 0.815 ± 0.071      | 0.965 ± 0.074      |
> |            | NLL          | 1.619 ± 0.039      | 1.606 ± 0.024      | 1.564 ± 0.067      | 1.658 ± 0.085      | 1.593 ± 0.063      | 1.572 ± 0.026      | 1.559 ± 0.100      | 1.627 ± 0.092      |
> |            | clock-time   | 1.030 ± 0.040      | 1.350 ± 0.070      | 0.790 ± 0.060      | 0.730 ± 0.040      | 1.090 ± 0.030      | 1.460 ± 0.060      | 0.920 ± 0.050      | 0.730 ± 0.010      |

---

> ### Comment · Reviewer_uEzz · 2025-08-02
>
> Thank you for your response. It appears to me that the assumption in Proposition 4.2 is significantly stronger than the assumptions made by Wenger et al. (2024) and Kanagawa et al. (2018). In the case of the CaGP's results presented in Wenger et al. (2024), the predictive variance represents the worst-case error of the posterior mean, under the condition that the true function lies within the RKHS of the data-generating process's covariance. This seems like a reasonable assumption when the model is well-specified.
>
> However, for the RCaGP, the predictive variance represents the worst-case error when the true function, minus the shrinkage term, belongs to the RKHS of the data-generating process's covariance. I find this somewhat puzzling because RCaGP, as a generalized Bayesian method, is predicated on the idea that standard Bayesian models are misspecified. This assumption makes it difficult to contextualize these results in practice, as it implies that the RCaGP itself is somehow well-specified. Given this context, what would you consider to be the key takeaway from Proposition 4.2?

---

> > ### Author Response · Authors · 2025-08-05
> >
> > Thank you for raising this important point. We agree that, at first glance, the assumption in Proposition 4.2 may appear stronger than those made in CaGP. However, we view this assumption as a natural analogue within the robust (misspecified) setting, rather than a stricter requirement.
> >
> > **Worst-case control of the residual**: RCaGP’s predictive variance still exactly equals the supremum of squared error, but now measured against the robustified target $h(x) = g(x) - m_w(x)$, not the original $g$. This formulation reflects the influence of the shrinkage term $m_w$, which absorbs outlier effects through the robust generalized Bayesian update.
> >
> > **Interpretation in the misspecified regime**. Instead of pretending the model is fully correct, we allow a structured form of misspecification—namely, the part captured by $m_w$. Proposition 4.2 then guarantees that any remaining discrepancy is still bounded by our variance. This mirrors the CaGP assumption in spirit: just as CaGP analyzes worst-case error over functions in the RKHS, RCaGP does the same for the residual $h$, which can be viewed as the portion of the signal that the robust update leaves uncorrected.
> >
> > **Conservative uncertainty under misspecification**. Even when classical Bayesian inference may fail to offer meaningful uncertainty due to model misspecification or outliers, RCaGP maintains a rigorous worst-case interpretation of its predictive variance—so long as the residual is sufficiently regular.

---

### Note · Authors · 2025-08-12

We thank the reviewers for their feedback and summarize below the main points addressed.

__More baselines.__ We implemented two robust SVGP baselines: one with a Student-t likelihood (Laplace approximation inference) and one extending [Ament et al.] via relevance pursuit, with the derivation added to the manuscript. On UCI regression datasets with asymmetric outliers, neither matched RCaGP, except for relevance-pursuit SVGP on Energy.

__More experiments.__ We conducted sensitivity studies (weight function parameters, noise magnitude, projection dimensionality), confirming RCaGP’s robustness. With an RBF kernel on UCI data, RCaGP achieved the best overall MAE/NLL, losing only 2 of 8 asymmetric-outlier cases. We also added a real-world Twitter Flash Crash case study, characterized by extreme market anomalies. We believe it offers valuable insight, even if its small scale makes it less representative of settings where approximate inference is most relevant.

__Model clarifications.__ Reviewer queries on the expert-driven prior mean prompted clarification in the manuscript. The method does not require a prior over corrupted points: in the experiments, RCaGP/RCSVGP used only a constant mean with a learnable scalar, yet achieved the best overall results. An expert-driven mean is optional and was tested only in a dedicated ablation, where it improved BO but assumes perfect outlier correction. We also clarified that Proposition 4.2’s RKHS assumption applies to the residual (target minus shrinkage term), paralleling CaGP’s worst-case bound in a robust, misspecified setting, and elaborated on the finite-KL robustness guarantee as a flexible alternative to norm-bounded criteria.

__Lengthscale initialization in BO.__ The importance of dimension-aware lengthscale initialization over GPyTorch’s default in BO was raised. We noted key differences from prior work on the subject: (i) use of custom GPs (SVGP, RCaGP, etc.) rather than exact GPs, and (ii) lengthscales optimized via EULBO rather than MAP. These may limit direct transfer of prior findings, and we therefore do not believe the high-dimensional BO results should be dismissed solely on the basis of potentially inappropriate lengthscale initialization. We carried out experiments with the new initialization on Rover60D and Hartmann6D (DNA180D could not be completed in time). Baseline ranking remained qualitatively the same, SVGP RCSVGP and RCaGP benefited from the new initialization; CaGP saw no change or worsened.

---

### Decision · Program_Chairs · 2025-09-17

**Decision:**

Accept (poster)

**Comment:**

This paper presents a technically sound and valuable contribution that merits publication. The reviewers are in unanimous agreement regarding acceptance.

The paper's core strength is its novel and principled integration of robustness to outliers with awareness of computational approximation uncertainty in Gaussian processes, which enables scalability. Jointly addressing these two problems, intertwined in the practical application of GPs, is a significant contribution to the field.

The rebuttal was critical for acceptance, in particular the addition of comparisons of the proposed method against other robust GP approaches (such as Student-t SVGP and a relevance pursuit-based model). In the final version of the paper, the authors are encouraged incorporate these as well as other feedback from reviewers (larger datasets where computational uncertainty is more relevant, cases where misspecification on real-world data is more problematic, etc).